Prepared for submission to JHEP

# Reparametrization mode Ward Identities and chaos in higher-pt. correlators in CFT$_2$

**Arnab Kundu**[a,1]**, Ayan K. Patra**[a,2]**, Rohan R. Poojary**[b,3]

[a] *Theory Division, Saha Institute of Nuclear Physics,*
*Homi Bhaba National Institute (HBNI),*
*1/AF, Bidhannagar, Kolkata 700064, India.*

[b] *Institute for Theoretical Physics, TU Wien,*
*Wiedner Hauptstrasse 8-10, 1040 Vienna, Austria.*

*E-mail:* [1] arnab.kundu@saha.ac.in, [2] ayan.patra@saha.ac.in,
[3] rpglaznos@gmail.com

ABSTRACT: Recently introduced reparametrization mode operators in CFTs have been shown to govern stress tensor interactions *via* the shadow operator formalism and seem to govern the effective dynamics of chaotic systems. We initiate a study of Ward identities of reparametrization mode operators *i.e.* how two dimensional CFT Ward identities govern the behaviour of insertions of reparametrization modes $\epsilon$ in correlation functions: $\langle \epsilon\epsilon\phi\phi \rangle$. We find that in the semi-classical limit of large $c$ they dictate the leading $\mathcal{O}(c^{-1})$ behaviour. While for the 4pt function this reproduces the same computation as done by Heahl, Reeves & Rozali in [1], in the case of 6pt function of pair-wise equal operators this provides an alternative way of computing the Virasoro block. We are lead to propose a representation of the vacuum block projector in terms of shadow blocks in an expansion in $1/c$. This representation allows for an easier method to compute the leading answer for the vacuum block of light operators in presence of a pair of heavy operators. We compute a maximally out of time ordered correlation function in a thermal background and find the expected behaviour of an exponential growth governed by Lyapunov index $\lambda_L = 2\pi/\beta$ lasting for twice the scrambling time of the system $t^* = \frac{\beta}{2\pi} \log c$. From a bulk perspective for the out of time ordered 4pt function we find that the Casimir equation for the stress tensor block reproduces the linearised back reaction in the bulk.

# 1 Introduction

Thermodynamics provides us with a universal infra-red (IR) description of a remarkably wide range of systems: across the scale of elementary particles to the large-scale structure of the observed Universe. Chaotic dynamics underlies the dynamical process towards thermalization in generic systems with a large number of degrees of freedom. Therefore, understanding better the role of chaos and thermalization in dynamical systems, in general, is an aspect of ubiquitous importance across various disciplines of physics.

Such questions have mostly been addressed conventionally in the framework of various spin-systems, and the likes. In recent years, however, in resonance with other advances in the framework of quantum field theory and quantum gravity, such questions have found an important place in understanding salient features of quantum nature of gravity. Conformal Field Theories (CFTs), particularly in two-dimensions, provide us with a remarkable control with which such dynamical aspects can be probed and explored in a precise sense. The CFT-intuition becomes a cornerstone of understanding these aspects in both the framework of QFT (e.g., in a perturbative deformation of a CFT), as well as in quantum gravity (primarily via Holography).

Motivated with this broad perspective, we explore further the chaotic dynamics in a two-dimensional CFT, with a large central charge and a sparse spectrum. Usually, it is assumed[1] that such CFTs have a Holographic dual and are therefore related to the quantum properties of a black hole in an anti-de Sitter (AdS) space-time. The gravity answer is reproduced assuming the dominance of the identity block over blocks of other operators in the theory and maximising over all possible exchanges [3–5]. In a unitary CFT, any primary field fuses with itself via the identity operator and therefore the corresponding identity block contributes universally to any correlator where such operator fusion occurs. Particularly, the stress-tensor, which is a descendant of the identity in 2d plays an important role in the identity block. In a holographic setting these correspond to graviton exchanges in the bulk and $AdS_3$ physics can be gleaned out from studying different kinematical regimes [6–11]. These therefore exhibit the typical expected behaviour of out of time ordered correlation functions in a thermal state of an exponential growth[12–17], which in the bulk corresponds to scrambling caused by exchange of gravitons close to the horizon of the black hole [18].

In the context of $AdS_3/CFT_2$ boundary gravitons are the only degrees freedom and their effective theory when coupled with other fields can stand for a generating function of stress tensor interactions in the CFT. In this context the theory of "reparametrization modes" was expanded upon to study the contributions of identity blocks in $CFT_2$ [19–21]. This was explored in detail by Cotler and Jensen[22] by carefully considering phase space quantization [23, 24] of boundary gravitons in $AdS_3$ with relevant boundary conditions and was found to be governed by 2 copies of the Alekseev-Shatashvili action- found by path integral quantization of the co-adjoint orbit of $\text{Diff}(\mathbb{S}^1)/SL(2,\mathbb{R})$.[2] This is a theory of reparametrizations sourced

---

[1]See *e.g.* [2].
[2]See also [25] for a previous work using geometric quantization.

by the boundary gravitons. This theory also consisted of bi-local operators the 2pt functions of which encoded contributions of the Virasoro blocks in a perturbative expansion in $1/c$ about large a central charge $c$. Consequently the vacuum block contribution to 4pt operators was obtained in the LLLL & HHLL limit[3] consistent with previously known results [9, 26–28].

Global blocks corresponding to any operator exchange have been understood in terms of shadow operator formalism. It was found in [1] that the shadow operator formalism [29, 30] can be used to recast this theory at the linear level in terms of "reparametrization modes"- $\epsilon^\mu(x)$; descendant of which is the shadow of the stress-tensor. This was then used to obtain a succinct expression for the 4pt vacuum block in even dimensions; while in $d = 2$ it reproduced the single graviton exchange answer in the limit of light operator dimensions. The leading correction in $1/c$ to the 4pt functions of light operators is basically obtained from the 2pt functions of the bi-locals which are linear in $\epsilon^\mu(x)$, without having the need to perform any conformal integrals. These $\epsilon^\mu(x)$ operators therefore seem to capture the universal physics of chaotic behaviour as it is the 2pt functions of their bi-locals which grows exponentially when their temporal positions are out of time ordered. Such operators as $\epsilon^\mu$ have also made an appearance in describing the effective action of SYK like models close to criticality [31] and nearly $AdS_2$ geometries in JT theories describing near horizon dynamics of near extremal black holes [32, 33]. It is the Schwarzian action cast in terms of these $\epsilon^\mu$ operator that captures the interaction of bulk scalars with the near horizon $AdS_2$ thus exhibiting the chaotic behaviour associated with black holes. Unlike black holes in $AdS_{d>3}$ the analysis in $AdS_3$ around a BTZ geometry of such operators- although from a bulk perspective, allows one to explore dynamics of chaos far from extremality [34, 35].

In [1, 36], the role of stress-tensor exchanges and correspondingly the importance of the reparametrization modes were highlighted and the computational simplifications they render where made use of in determining the Identity block contributions to four-point and six-point functions in $CFT_2$, respectively. As usual, these computations are carried out in the Euclidean framework and then analytically continued to obtain the corresponding Lorentzian correlators. Of particular focus is the maximally braided out-of-time-order correlator (OTOC) that exhibits an exponentially growing mode in time-scales larger than the dissipation-scale [19].

As already mentioned, in [36], the reparametrization mode formalism was heavily used to unpack the physics mentioned above. Furthermore, additional assumptions were made in [36], about the structure of the six-point block, in order to explicitly compute the six-point function. In [36], the reparametrization mode formalism in $CFT_2$ was used in conjunction with a proposal for the non-linear version of the corresponding block, in a particular "star" channel. The star channel is a natural generalization of the identity block considered in the 4pt case as all internal lines are those of the stress-tensor [37–39]. This channel can be contrasted with the well studied "comb" channel where the internal operators are scalar primaries[40–48]. To obtain the full contribution from the Virasoro blocks a non-linear realization of the bi-locals of $\epsilon^\mu$ had to be used, motivated from the works of Cotler and Jensen

---

[3]Where a pair of operators have dimensions that scale as $c$ while the others are small compared to $c$.

[22] and [1, 21]. This non-linear proposal is essential in capturing the non-linear interactions in the dual gravitational description, that is ultimately responsible for the behaviour of the corresponding correlator. It is plausible that the reparametrization mode formalism, fused with standard symmetry constraints involving the stress-tensor:e.g.Ward identities, provides us with a powerful control on generic n-point correlator and its chaotic behaviour.

In this article, we initiate a study of how 2d Ward identities- of the form of $\langle T\phi\phi\rangle$, $\langle TT\phi\phi\rangle$ and so on; can be used in conjunction with the reparametrization modes and how it provides a powerful and alternative method of computing the higher point correlators. The main benefit of this approach is that the Ward identities themselves incorporate the non-linear interactions, implicitly, and avoid any further assumption. In fact, it is possible to provide evidence in support of the proposal made in [36].

In this article, we reproduce the six-point OTOC result of [36], using primarily the Ward identities on the reparametrization modes. Our method provides a further evidence in support of the proposal of [36], which is can also be generalized for subsequent higher point functions. The equivalence of these 2 methods must stem from the fact that the effect of 2d Ward identities of $\langle\phi\phi\rangle$ and the non-linear bi-locals of $\epsilon^\mu$ on CFT$_2$ correlators can be captured by the effective action for stress-tensor interactions obtained from Alekseev-Shatashvili action. This action is also related to the Polyakov action, for a discussion refer to [49]. The matter(primary) fields interact with such an action by the generating function of their 2pt correlators. From the bulk Holographic perspective, it is interesting to understand these Ward identities on the reparametrization modes. We leave such Holographic aspects for future, however, we point out that the conformal Casimir equation, satisfied by the stress-tensor conformal block, reproduces the (source-less) linearized Einstein equation from a three-dimensional bulk perspective.

This paper is sectioned as follows: In section-2 we briefly review the shadow operator formalism. In section-3 we compute the $\langle\epsilon\epsilon\phi\phi\rangle$ correlator from the stress tensor Ward identity for $\langle TT\phi\phi\rangle$. We then argue how this correlator contributes to a 6pt function of pair-wise equal operators with conformal dimensions $h \sim \mathcal{O}(c^0)$. In doing so we postulate a representation of the Virasoro vacuum block projector in terms of conformal integrals using stress-tensor $T$ and it's shadow $\tilde{T}$. This allows us to diagrammatically expand light operator correlators in a series in $1/c$. We make some observations regarding generalising this computation to higher-point correlators and give a simple proof of how 2 heavy operator insertions with conformal dimensions $H \sim c$ can be seen as a conformal transformation in the $c \to \infty$ limit for the 4pt. vacuum block. We are also able to compute leading vacuum blocks of simpler 8pt. and 9pt. generalizations of the 6pt. function algebraically. In section-4 we compute the OTOC governed by $\langle\epsilon\epsilon\phi\phi\rangle$ which contributes to the 6-pt connected "star" channel. In section-5 we make some observations which yield some insight into the bulk perspective for 4pt OTOC.

**Notation:** We make use of (lower-case)Greek alphabets to indicate $d$-dimensional vector

indices. Lower-case barred and unbarred Roman alphabets indicate $d + 2$-dimensional embedding space vector indices while $X, Y, Z$ indicate embedding space vectors. 2-dimensional coordinates are the standard (anti)holomorphic indicated by $(\bar{z})z$, $(\bar{x})x$, *etc.*. Shadow operators are indicated by an overhead $\sim$. We also use the short hand of the form $\phi(z_1) \equiv \phi_1$ or $\phi(x_1^\mu) \equiv \phi_1$, $T^{\mu\nu}(x_0) \equiv T_0^{\mu\nu}$ or $T_{\mu\nu}(x_0) \equiv T_{\mu\nu}^0$, $I^{\mu\nu}(x_1 - x_2) = I^{\mu\nu}(x_{12}) \equiv I_{12}^{\mu\nu}$ or $I_{\mu\nu}(x_{12}) \equiv I_{\mu\nu}^{12}$, bi-local dependence as $\mathcal{B}(x_1, x_2) = \mathcal{B}_{12}$ etc. for simplicity of notation along with similar use of negative numbers *i.e.* $\phi_{-1} = \phi(z_{-1})$ or $\phi_{-1} = \phi(x_{-1}^\mu)$. We also use $\int d^d x_0 = \int_0$ to indicate conformally invariant integrals[4] to save space.

## 2 Review of shadow operators and reparametrization modes.

In this section we review the shadow operator formalism and the use of reparametrization modes as described in [21]. Given a primary scalar operator $\mathcal{O}$ with dim $\Delta$ in a $\text{CFT}_d$ it's shadow is defined using the embedding space conformal integral as follows

$$\tilde{\mathcal{O}}(X) = \frac{k_{\Delta,0}}{\pi^{d/2}} \int D^d Y \, \frac{\mathcal{O}(Y)}{(-2X.Y)^{d-\Delta}}, \tag{2.1}$$

with $k(d, \Delta)$ as in (2.2) with $l = 0$. Here the embedding space coordinates $X^a = \{X^+, X^-, X^\mu\}$ on $\mathbb{R}^{d+1,1}$ wherein the $\text{CFT}_d$ is defined on an $\mathbb{R}^d$ with coordinates $x^\mu$. We refer to Simmons-Duffins[29] for defining these conformal integrals. We note certain useful results in Appendix A.

In general the shadow of the spinning operators is defined as

$$\tilde{\mathcal{O}}(y)^{\mu_1 \dots \mu_l} = \frac{k_{\Delta,l}}{\pi^{d/2}} \int D^d x \, \frac{I(x-y)^{\mu_1 \nu_1} \dots I(x-y)^{\mu_l \nu_l}}{((x-y)^2)^{d-\Delta}} \mathcal{O}_{\nu_1 \dots \nu_l}(x)$$

$$\text{with,} \quad I_{\mu\nu}(x) = \eta_{\mu\nu} - 2\frac{x_\mu x_\nu}{x^2}$$

$$\text{where} \quad k_{\Delta,l} = \frac{\Gamma(\Delta - 1) \, \Gamma(d - \Delta + l)}{\Gamma(\Delta + l - 1) \, \Gamma(\Delta - \frac{d}{2})}. \tag{2.2}$$

Here we have abused the notation and not used embedding space coordinates in defining the integrals[5]. The shadow operators; though fictitious are useful in construction of the projectors which project onto the conformal block of the operators. Projectors from shadow operators are obtained by constructing

$$|\mathcal{O}| = k'(d, \Delta, l) \int d^d x \, \tilde{\mathcal{O}}(x) \rangle \langle \mathcal{O}(x). \tag{2.3}$$

The use of these projectors in 4pt functions yields not only the block corresponding to the operator $\mathcal{O}$ but also its shadow $\tilde{\mathcal{O}}$

$$\frac{\langle \phi_1 \phi_2 | \mathcal{O} | \phi_3 \phi_4 \rangle}{\langle \phi_1 \phi_2 \rangle \langle \phi_3 \phi_4 \rangle} = C_{\phi\phi\mathcal{O}}^2 (G_\Delta^{(l)}(u, v) + G_{(d-\Delta)}^{(l)}(u, v))$$

---

[4]These are the only form of integrals we would encounter unless mentioned otherwise.

[5]We denote by $\int d^2 x$ as the 2dim conformally invariant integral for the rest of the paper.

as $u \to 0, v \to 1$
$$G_\Delta^{(l)}(u,v) \sim u^{\frac{\Delta-l}{2}}(1-v)^l, \qquad G_{d-\Delta}^{(l)}(u,v) \sim u^{\frac{d-\Delta+l}{2}}(1-v)^l \qquad (2.4)$$

where $u = \frac{x_{12}^2 x_{34}^2}{x_{13}^2 x_{24}^2}$, $v = \frac{x_{14}^2 x_{23}^2}{x_{13}^2 x_{24}^2}$ are the invariant cross ratios. We would have subtract out the the contribution coming from the shadow block by discerning its behaviour as $u \to 0$. However as was seen in [1] in the 2d case this is simply achieved by imposing holomorphic factorization in $\{z, \bar{z}\}$. In 2d, using the left-right moving Euclidean co-ordinates $\{z, \bar{z}\}$ the metric and the inversion tensor takes the form

$$\eta_{\mu\nu} = \frac{1}{2}\begin{pmatrix} 0 & 1 \\ 1 & 0 \end{pmatrix}, \qquad I^{\mu\nu} = -2\begin{pmatrix} z/\bar{z} & 0 \\ 0 & \bar{z}/z \end{pmatrix} \qquad (2.5)$$

Using which we can describe the shadow as of an operator $\mathcal{O}$ with conformal weights $\{h, \bar{h}\}$ as

$$\tilde{\mathcal{O}}(z) = k_{\frac{h+\bar{h}}{2}, \frac{h-\bar{h}}{2}} \int d^2x \, \frac{O(x)}{(x-z)^{(2-2h)}(\bar{x}-\bar{z})^{(2-2\bar{h})}} \qquad (2.6)$$

The shadow of the stress-tensor components are therefore

$$\tilde{T}(z) = \frac{2}{\pi}\int d^2x \, \frac{(z-x)^2}{(\bar{z}-\bar{x})^2}T(x), \quad \tilde{\bar{T}}(\bar{z}) = \frac{2}{\pi}\int d^2x \, \frac{(\bar{z}-\bar{x})^2}{(z-x)^2}\bar{T}(\bar{x}) \qquad (2.7)$$

The reparemetrization modes introduced in [1] are constructed from the $\tilde{T}$ shadows of the stress-tensor

$$\tilde{T}_{\mu\nu}(x) = \frac{2C_T \pi^{d/2}}{k_{0,2}}\mathbb{P}_{\mu\nu}^{\alpha\beta}\,\partial_\alpha\epsilon_\beta, \qquad \mathbb{P}_{\mu\nu}^{\alpha\beta} = \frac{1}{2}\left(\delta_\mu^\alpha\delta_\nu^\beta + \delta_\nu^\alpha\delta_\mu^\beta\right) - \frac{1}{d}\eta^{\alpha\beta}\eta_{\mu\nu} \qquad (2.8)$$

where $\mathbb{P}_{\mu\nu}^{\alpha\beta}$ is a projector onto traceless and symmetric part and $C_T = c/2$ is the coefficient of the stress-tensor 2pt function with $c$ the central charge . It is worth noting that as the shadow $\tilde{T}$ has conformal weight $\Delta = 0$ the reparametrization mode operator $\epsilon$ has a conformal weight $\Delta = -1$. In 2 dimensions the above relations take the form

$$\tilde{T}^{zz} = \tilde{T}(z) = \frac{c}{3}\bar{\partial}\epsilon, \qquad\qquad \tilde{T}^{\bar{z}\bar{z}} = \tilde{\bar{T}} = \frac{c}{3}\partial\bar{\epsilon} \qquad (2.9)$$

Using the fact that $T = \tilde{\tilde{T}}$ one can show that[1, 36]

$$T_{zz} = T = -\frac{c}{12}\partial^3\epsilon, \qquad\qquad T_{\bar{z}\bar{z}} = \bar{T} = -\frac{c}{12}\bar{\partial}^3\bar{\epsilon} \qquad (2.10)$$

Thus the stress-tensor itself can be seen as a descendent of the reparametrization mode $\epsilon$. We refer readers to section 3 of [1] for a discussion on effective action for the $\epsilon$ modes and for a d-dimensional generalization of the above relation. The above relations impose strict consistency conditions which any correlation function involving $\epsilon$s have to satisfy:

$$\langle \partial^3\epsilon_1 \ldots \partial^3\epsilon_n \phi\phi \ldots \rangle = \left(-\frac{12}{c}\right)^n \langle T_1 \ldots T_n \phi\phi \ldots \rangle,$$
$$\langle \bar{\partial}\epsilon_1 \ldots \bar{\partial}\epsilon_m \partial^3\epsilon_{m+1} \ldots \partial^3\epsilon_n \phi\phi \ldots \rangle = \left(-\frac{3}{c}\right)^n 4^{n-m}\langle \tilde{T}_1 \ldots \tilde{T}_m T_{m+1} \ldots T_n \phi\phi \ldots \rangle, \quad (2.11)$$

It is worth noting that the definition (2.9) doesn't readily imply holomorphic factorization. As we will see later in specific case of interest to us, the conditions of the form (2.11) will be used to fix the ambiguities arising from (2.9). These would give rise to a set of differential constraints. We will deal with these in the next section.

As was shown in [1] the reparametrization modes can be used directly to compute the contribution of the stress-tensor block to the 4pt function $\langle XX\phi\phi\rangle$ upto $1/c$ order .

$$
\begin{aligned}
\langle X_1 X_2 | T | \phi_3 \phi_4 \rangle &= \langle X_1 X_2 | T | T | \phi_3 \phi_4 \rangle \\
&= \mathbb{P}^{\rho\sigma}_{\alpha\beta} \mathbb{P}^{\gamma\delta}_{\mu\nu} \int d^d x \, d^d y \, \langle X_1 X_2 \, T^{\alpha\beta}(y) \rangle \, \langle \partial_\rho \epsilon_\sigma(y) \partial_\gamma \epsilon_\delta(x) \rangle \, \langle T^{\mu\nu}(x) \phi_3 \phi_4 \rangle \\
&= \Delta_\phi \Delta_X \langle \hat{\mathcal{B}}^{(1)}_{12} \hat{\mathcal{B}}^{(1)}_{34} \rangle \langle X_1 X_2 \rangle \langle \phi_3 \phi_4 \rangle
\end{aligned}
\tag{2.12}
$$

where in going from the first line to the second line we have used

$$
|T| = |\tilde{T}| = |T|^2 = \frac{k_{0,2}}{\pi^{d/2} c} \int d^d x \, \tilde{T}(x) \rangle \langle \tilde{T}(x)
\tag{2.13}
$$

In going from the second line to the third line in (2.12) we have used first the definition of $\mathbb{P}^{\alpha\beta}_{\mu\nu}$ in (2.8), then shift derivatives from $\epsilon$s to $T$s and use the conformal Ward identity for $T_{\mu\nu}$ insertions

$$
\begin{aligned}
\langle \partial_\mu T_0^{\mu\nu} \phi_1 \phi_2 \rangle &= - \left[ \delta^d(x_{01}) \partial_1^\nu + \delta^d(x_{02}) \partial_2^\nu \right] \langle \phi_1 \phi_2 \rangle + \frac{\Delta_\phi}{d} \partial_0^\nu \left[ \delta^d(x_{01}) + \delta^d(x_{02}) \right] \langle \phi_1 \phi_2 \rangle \\
\langle T_{0\mu}^\mu \phi_1 \phi_2 \rangle &= 0
\end{aligned}
\tag{2.14}
$$

to get rid of the integrals. Therefore we find the $\hat{\mathcal{B}}^{(1)}_{ij}$ to be[6]

$$
\hat{\mathcal{B}}^{(1)}_{12} = \frac{1}{d} \left( \partial_\mu \epsilon_1^\mu + \partial_\mu \epsilon_2^\mu \right) - 2 \frac{(\epsilon_1 - \epsilon_2)^\mu (x_{12})_\mu}{x_{12}^2}, \qquad x_{12}^\mu = (x_1 - x_2)^\mu
\tag{2.15}
$$

which are the bi-local bilinear operators constructed out of $\epsilon$s. These bi-locals have been used in a similar context in [50]. In 2d these take the form

$$
\mathcal{B}^{(1)}_{12} = \partial_1 \epsilon_1 + \partial_2 \epsilon_2 - 2 \frac{\epsilon_1 - \epsilon_2}{z_{12}}, \quad \bar{\mathcal{B}}^{(1)}_{12} = \bar{\partial}_1 \bar{\epsilon}_1 + \bar{\partial}_2 \bar{\epsilon}_2 - 2 \frac{\bar{\epsilon}_1 - \bar{\epsilon}_2}{\bar{z}_{12}}, \qquad z_{12} = z_1 - z_2.
\tag{2.16}
$$

The 2pt functions for $\epsilon$s can be deduced from those of $\tilde{T}$ which in turn can be readily obtained from general conformal covariance

$$
\begin{aligned}
\langle \tilde{T}_1^{\mu\nu} \tilde{T}_2^{\alpha\beta} \rangle &= 2 C_T \frac{k_{d,2}}{k_{d,0}} \mathbb{P}^{\mu\nu}_{\rho\sigma} \mathbb{P}^{\alpha\beta}_{\gamma\delta} \, I_{12}^{\rho\gamma} I_{12}^{\sigma\delta} \\
&\equiv - C_T \frac{k_{d,2}}{k_{d,0}} \mathbb{P}^{\mu\nu}_{\rho\sigma} \mathbb{P}^{\alpha\beta}_{\gamma\delta} \partial^\sigma \partial^\delta \left[ I_{12}^{\rho\gamma} x_{12}^2 \log(\mu^2 x_{12}^2) \right]
\end{aligned}
\tag{2.17}
$$

---

[6]Our definition of $\hat{\mathcal{B}}^{(1)}_{ij}$ differs from that of [36] by a factor of $\Delta \overset{2d}{=} \frac{h + \bar{h}}{2}$.

Which in 2d implies

$$\langle \tilde{T}_1 \tilde{T}_2 \rangle = \frac{2c}{3} \bar{\partial}_1 \bar{\partial}_2 \left( z_{12}^2 \log(\mu^2 z_{12} \bar{z}_{12}) \right) \tag{2.18}$$

We also note that crucially that

$$\langle T_i \tilde{T}_j \rangle = \frac{c\pi}{3} \delta^{(2)}(x_i - x_j). \tag{2.19}$$

Therefore the 2pt function of the reparametrization modes is

$$\langle \epsilon_1 \epsilon_2 \rangle = \frac{6}{c} z_{12}^2 \log(\mu^2 z_{12} \bar{z}_{12}). \tag{2.20}$$

This also satisfies the consistency conditions (2.11) for $\langle \epsilon_1 \epsilon_2 \rangle$. Here $\mu^2$ is a length parameter which does not contribute to any physical quantity that one can compute. Note that the above functional dependence is not unique upto addition of terms quatratic in distances but this would not effect computation of physically relevant quantities. The monodromy relations satisfied by the conformal blocks allows one to factor out the contribution of the shadow blocks. In 2d this turns out to simply imposing holomorphic factorization.

$$\langle \epsilon_1 \epsilon_2 \rangle_{\text{phys}} = \frac{6}{c} z_{12}^2 \log(\mu^2 z_{12}). \tag{2.21}$$

The above propagator can then be used to compute the conformal block in (2.12)

$$\langle \mathcal{B}_{12}^{(1)} \mathcal{B}_{34}^{(1)} \rangle_{\text{phys}} = \frac{2}{c} z^2 \, _2F_1(2, 2, 4, z) \tag{2.22}$$

in terms of the the invariant cross ratio $z = \frac{z_{12} z_{34}}{z_{13} z_{24}}$. This is indeed the single graviton exchange correction computed to $1/c$ order for the 4pt function of pairs of light scalars [9]. As expected the global stress-tensor block in 2d captures only the level one states generated over the vacuum being exchanged.[7]

# 3    Ward identity for $\epsilon$s and 6pt connected vacuum block

The contribution of the stress-tensor block to any correlation function can be thought of as being obtained by inserting the stress-tensor projector $|T|$ as defined in (2.12). Every such insertion can be thought of as introducing a power of $1/c$. The stress-tensor legs propagating within the diagram can all be decomposed into a web of legs connected with 3pt vertices of stress-tensor and its shadow. Consequently fusing any three of the stress-tensors introduces a power of $c$ (i.e. $\langle T_1 T_2 T_3 \rangle \sim c$).[8] Given that insertion of the stress-tensor projectors $|T|$ can be recast in terms of insertion of $\epsilon$s as in (2.12) we must be able to express the stress-tensor

---

[7]$L_{-1}^n T \rangle \sim L_{-n} \rangle$ since $T \rangle = L_{-2} \rangle$.

[8]In 2d we have $\langle T_1 T_2 T_3 \rangle = -c \frac{1}{z_{12}^2 z_{23}^2 z_{13}^2}$, $\langle \tilde{T}_1 \tilde{T}_2 \tilde{T}_3 \rangle = c \frac{8}{3} \frac{z_{12} z_{23} z_{13}}{\bar{z}_{12} \bar{z}_{23} \bar{z}_{13}}$  $\langle T_1 \tilde{T}_2 \tilde{T}_3 \rangle = c \frac{8}{3} \frac{z_{23}^4}{z_{12}^2 z_{23}^2 z_{13}^2}$
$\langle T_1 T_2 \tilde{T}_3 \rangle = c \frac{\bar{z}_{12} \bar{z}_{23} z_{13}}{z_{12}^5 \bar{z}_{23} \bar{z}_{13}}$

block contribution as expectation values involving insertions of epsilons. Insertions of stress-tensors in 2d is completely fixed in terms of the 2d Ward identity. We would therefore like to understand what this implies for the $\epsilon$ insertions.

We first begin with $\epsilon$ insertions in 2-pt. functions of scalars $\phi$ of conformal weight $\{h, h\}$ *i.e.* $\Delta = 2h$. This can be obtained from

$$\langle \tilde{T}_{-1}^{ab} \phi_1 \phi_2 \rangle = \frac{2}{\pi} \int_0 \frac{I_{-10}^{a\bar{a}} I_{-10}^{b\bar{b}} \langle T_0 \phi_1 \phi_2 \rangle_{\bar{a}\bar{b}}}{X_{-10}^0} \tag{3.1}$$

where $T_{zz} = T$, $\tilde{T}^{zz} = \tilde{T}$. Where we denote compactly the conformal integral as

$$\int_i \equiv \int d^2 X_i. \tag{3.2}$$

This can be evaluated by evaluating $\langle \tilde{B}_{(1)}^{ab} \phi_{(2)} \phi_{(3)} \rangle$ where $B_{ab}$ is spin 2 primary with weight $\delta$ and then taking the limit $\delta \to d = 2$. Alternatively, this should match the expected answer for

$$\langle \tilde{T}_{-1}^{ab} \phi_1 \phi_2 \rangle = \frac{2h}{\pi} \left. \frac{I_{-11}^{a\bar{a}} I_{\bar{a}\bar{b}}^{12} I_{-12}^{b\bar{b}}}{(X_{-11} X_{-12})^{\delta/2} X_{12}^{2h-\delta/2}} \right|_{\delta \to 0} \tag{3.3}$$

We can insert a $|T|$ in between the $\tilde{T}$ and $\phi$s in the *rhs* above as any other insertion (including) 1 would yield 0. Here we assume that $\langle \tilde{T}^{ab} \rangle = 0$.

$$\begin{aligned}
\langle \tilde{T}_{-1}^{\mu\nu} |T| \phi_1 \phi_2 \rangle &= \mathbb{P}_{\alpha\beta}^{\rho\sigma} \int d^d x \langle \tilde{T}^{\mu\nu} \partial_\rho \epsilon_\sigma(x) \rangle \langle T^{\alpha\beta}(x) \phi_1 \phi_2 \rangle \\
&= \frac{c\pi^{d/2}}{k_{0,2}} \mathbb{P}_{\alpha\beta}^{\rho\sigma} \mathbb{P}_{\gamma\delta}^{\mu\nu} \int d^d x \langle \partial^\gamma \epsilon_{-1}^\delta \partial_\rho \epsilon_\sigma(x) \rangle \langle T^{\alpha\beta}(x) \phi_1 \phi_2 \rangle \\
&= \frac{c\pi^{d/2}}{k_{0,2}} \mathbb{P}_{\alpha\beta}^{\rho\sigma} \langle \partial^\gamma \epsilon_{-1}^\sigma \hat{\mathcal{B}}_{12}^{(1)} \rangle \langle \phi_1 \phi_2 \rangle.
\end{aligned} \tag{3.4}$$

where as before we used the (2.8)&(3.34) and shift derivatives onto $T^{\alpha\beta}(x)$ above then use the Ward identity (2.14) as before. This in 2d yields[9]

$$\langle \tilde{T}_{-1} \phi_1 \phi_2 \rangle = \frac{ch}{3} \bar{\partial}_{-1} \langle \epsilon_{-1} \mathcal{B}_{12}^{(1)} \rangle \langle \phi_1 \phi_2 \rangle \tag{3.5}$$

Therefore we find

$$\langle \epsilon_{-1} \phi_1 \phi_2 \rangle = \langle \epsilon_{-1} \mathcal{B}_{12}^{(1)} \rangle \langle \phi_1 \phi_2 \rangle \tag{3.6}$$

This relation (along with its barred counter-part) had already found it's use in computing the 4pt stress-tensor block [1] as follows

$$\langle \phi_1 \phi_2 |T| \phi_3 \phi_4 \rangle = -\mathbb{P}_{\mu\nu}^{\alpha\beta} \int_{-1} \langle \phi_1 \phi_2 \, \partial_\beta \epsilon_{-1\alpha} \rangle \langle T_{-1}^{\mu\nu} \phi_3 \phi_4 \rangle$$

--------

[9]Note that in 2d the use of $|T|$ as an integral yields the projector onto only the global states of the stress-tensor, however this is enough to fix the structure of 3pt functions as they are completely determined by only global symmetries.

$$= -\Delta \mathbb{P}^{\alpha\beta}_{\mu\nu} \int_{-1} \langle \phi_1 \phi_2 \rangle \langle \hat{\mathcal{B}}^{(1)}_{12} \partial_\beta \epsilon_{-1\alpha} \rangle \langle T^{\mu\nu}_{-1} \phi_3 \phi_4 \rangle$$

$$= \Delta^2 \langle \hat{\mathcal{B}}^{(1)}_{12} \hat{\mathcal{B}}^{(1)}_{34} \rangle \langle \phi_1 \phi_2 \rangle \langle \phi_3 \phi_4 \rangle \tag{3.7}$$

where we use definition of $\epsilon$s (2.9),(2.8),(3.34) and the Ward identity in the going from the second to the third line along with $\epsilon^z = \epsilon$, $\epsilon^{\bar{z}} = \bar{\epsilon}$. Note that in the last line above $\hat{\mathcal{B}}^{(1)}_{ij}$ consists of sum of holomorphic and anti-holomorphic parts while that in (3.6) consists of only the holomorphic sector[10]. The above expression was computed in [1] by a different approach of squaring the projectors and using the stress-tensor Ward-identities.

We would next like to compute $\langle \phi_1 \phi_2 \epsilon_3 \epsilon_4 \rangle$. To this end we turn to compute $\langle \tilde{T}_{-1} \tilde{T}_0 \phi_1 \phi_2 \rangle$, here we do not have the benefit of the general structure being fixed by global conformal invariance as was in the previous case. It would be instructive to write down the full Ward identity

$$\langle T_{-1} T_0 \phi_1 \phi_2 \rangle = \frac{c/2}{z^4_{-10}} \langle \phi_1 \phi_2 \rangle + \sum_{i=0}^{2} \left( \frac{h_i}{z^2_{-1i}} + \frac{\partial_i}{z_{-1i}} \right) \frac{h z^2_{12}}{z^2_{01} z^2_{02}} \frac{1}{(z_{12} \bar{z}_{12})^{2h}} \tag{3.8}$$

where $\langle \phi_1 \phi_2 \rangle = (z_{12} \bar{z}_{12})^{-2h} \equiv X^{-2h}_{12}$, $h_0 = 2$ and $h_{1,2} = h$. we would like to evaluate the shadow of the above *rhs wrt* coordinates $X_{-1} \& X_0$. The shadow of the first term is simply the obtained by evaluating $\langle \tilde{B}^{ab}_3 \tilde{B}^{cd}_4 \rangle$ and taking the conformal dimension of $\tilde{B}$ to zero. The second term as a whole is globally conformally invariant but not in parts. It turns out one can split it into 3 parts each of which are globally conformally invariant[11]

$$\langle T_{-1} T_0 \phi_1 \phi_2 \rangle = \frac{c/2}{z^4_{-10}} \langle \phi_1 \phi_2 \rangle + \left( \frac{(h-1) z^2_{12}}{z^2_{-11} z^2_{-12}} + \frac{z^2_{01}}{z^2_{-10} z^2_{-11}} + \frac{z^2_{02}}{z^2_{-10} z^2_{-12}} \right) \frac{h z^2_{12}}{z^2_{01} z^2_{02}} \frac{1}{(z_{12} \bar{z}_{12})^{2h}}. \tag{3.9}$$

The benefit of expressing the Ward identity in conformally invariant terms is that we can make use of the expression

$$\langle T_0 \phi_1 \phi_2 \rangle = -\frac{h z^2_{12}}{z^2_{01} z^2_{02}} \langle \phi_1 \phi_2 \rangle \implies \langle \tilde{T}_4 \phi_1 \phi_2 \rangle = \frac{c}{3} \langle \bar{\partial} \epsilon_4 \phi_1 \phi_2 \rangle = \frac{2}{\pi} \int_0 \frac{z^2_{40}}{\bar{z}^2_{40}} \frac{h z^2_{12}}{z^2_{01} z^2_{02}} \langle \phi_1 \phi_2 \rangle \tag{3.10}$$

using which we can write the Ward identity as

$$\langle T_{-1} T_0 \phi_1 \phi_2 \rangle = \langle T_{-1} T_0 \rangle \langle \phi_1 \phi_2 \rangle + h(h-1) \langle T_{-1} \mathcal{B}^{(1)}_{12} \rangle \langle T_0 \mathcal{B}^{(1)}_{12} \rangle \langle \phi_1 \phi_2 \rangle$$

$$+ h \langle T_0 \mathcal{B}^{(1)}_{12} \rangle \left( \langle T_{-1} \mathcal{B}^{(1)}_{01} \rangle + \langle T_{-1} \mathcal{B}^{(1)}_{02} \rangle \right) \langle \phi_1 \phi_2 \rangle \tag{3.11}$$

Although not explicitly manifest the last term is symmetric under $0 \leftrightarrow -1$. Therefore we can write a Ward identity for the shadow stress-tensor $\tilde{T}$ as

---

[10] Any insertion of $|T|$ in 2d would consist of a holomorphic term and an anti-holomorphic term, (3.6) deals with only the holomorphic sector.

[11] We see this by counting powers of variables to be integrated *i.e.* $X_{-1} \& X_0$, and they must add up to $-d = -2$ for each of them.

$$\langle \tilde{T}_3 \tilde{T}_4 \phi_1 \phi_2 \rangle = \langle \tilde{T}_3 \tilde{T}_4 \rangle \langle \phi_1 \phi_2 \rangle + h(h-1) \langle \tilde{T}_3 \mathcal{B}_{12}^{(1)} \rangle \langle \tilde{T}_4 \mathcal{B}_{12}^{(1)} \rangle \langle \phi_1 \phi_2 \rangle$$

$$+ \frac{2h}{\pi} \int_0 \frac{z_{40}^2}{\bar{z}_{40}^2} \langle T_0 \mathcal{B}_{12}^{(1)} \rangle \left( \langle \tilde{T}_3 \mathcal{B}_{01}^{(1)} \rangle + \langle \tilde{T}_3 \mathcal{B}_{02}^{(1)} \rangle \right) \langle \phi_1 \phi_2 \rangle \quad (3.12)$$

where we have used

$$\frac{\langle T_0 \phi_1 \phi_2 \rangle}{\langle \phi_1 \phi_2 \rangle} = h \langle T_0 \mathcal{B}_{12}^{(1)} \rangle, \qquad \frac{\langle \tilde{T}_4 \phi_1 \phi_2 \rangle}{\langle \phi_1 \phi_2 \rangle} = h \langle \tilde{T}_4 \mathcal{B}_{12}^{(1)} \rangle \qquad (3.13)$$

which can be verified given the basic definitions in the previous section. Making use of the definition (2.9) we can easily write the corresponding Ward identity for the reparametrization modes, except for the last term above. Restricting further to only the physical block by taking only the holomorphic sector above yields

$$\frac{\langle \epsilon_3 \epsilon_4 \phi_1 \phi_2 \rangle_{\text{phys}}}{\langle \phi_1 \phi_2 \rangle} = \langle \epsilon_3 \epsilon_4 \rangle_{\text{phys}} + h(h-1) \langle \epsilon_3 \mathcal{B}_{12}^{(1)} \rangle_{\text{phys}} \langle \epsilon_4 \mathcal{B}_{12}^{(1)} \rangle_{\text{phys}} + h \left( \frac{12}{c} \right)^2 \mathcal{C}_{\text{phys}}^{(2)}$$

$$(3.14)$$

where the last term $\mathcal{C}^{(2)}$ needs to be determined. We note here that this term can be determined in 2 possible ways: $(i)$ by explicitly solving the integral in the last term in (3.12) by making use of the integrals listed in Appendix B, and then writing the result as total derivatives of $\bar{z}_3$ & $\bar{z}_4$. or $(ii)$ by solving consistency conditions of the type (2.11) some of which result in solving differential equations in the cross ratio. Method $(i)$ is actually insufficient for getting the right answer as there can be ambiguities in adding a term which vanish upon differentiation $wrt \ \bar{z}_{3,4}$. Upon integration (as done in Appendix C) one can write $\mathcal{C}^{(2)}$ as

$$\mathcal{C}_{\text{phys}}^{(2)} = \langle \epsilon_3 \epsilon_4 \rangle_{\text{phys}} \left[ 4 + \left( -2 + \frac{4}{z} \right) \log(1-z) \right] + z_{34}^2 \mathcal{F}(z). \qquad (3.15)$$

where $z = \frac{z_{12} z_{34}}{z_{13} z_{24}}$; we simply write the resultant integral as a total derivative of $\bar{z}_{3,4}$. Here we have added an extra term $z_{34}^2 \mathcal{F}(z)$ which would be required to make (3.14) satisfy the the constraints (2.11).

Method $(ii)$ solves for constrains due to the relation

$$T = -\frac{c}{12} \partial^3 \epsilon \qquad (3.16)$$

implying that we must get the Ward identity (3.8) (appropriately normalized) upon using the above relation on each of the $\epsilon$s in (3.14).

$$\langle \partial^3 \epsilon_3 \partial^3 \epsilon_4 \phi_1 \phi_2 \rangle = \left( \frac{12}{c} \right)^2 \langle T_3 T_4 \phi_1 \phi_2 \rangle \qquad (3.17)$$

This consistency condition is satisfied term by term and the first 2 terms in (3.14) satisfy this. Apart from the above condition we can also impose

$$\langle \partial^3 \epsilon_3 \bar{\partial} \epsilon_4 \phi_1 \phi_2 \rangle = -\frac{36}{c^2} \langle T_3 \tilde{T}_4 \phi_1 \phi_2 \rangle \tag{3.18}$$

but this condition constrains pieces that give contribution to the shadow block in $(3.14)^{12}$. To see this we note that the Ward identity (3.11) implies

$$\langle T_3 \tilde{T}_4 \phi_1 \phi_2 \rangle = \langle T_3 \tilde{T}_4 \rangle \langle \phi_1 \phi_2 \rangle + \frac{(h-1)}{h} \langle T_3 \mathcal{B}_{12}^{(1)} \rangle \langle \tilde{T}_4 \mathcal{B}_{12}^{(1)} \rangle \langle \phi_1 \phi_2 \rangle + \langle T_3 \mathcal{B}_{12}^{(1)} \rangle \left( \langle \tilde{T}_4 \mathcal{B}_{31}^{(1)} \rangle + \langle \tilde{T}_4 \mathcal{B}_{32}^{(1)} \rangle \right) \langle \phi_1 \phi_2 \rangle \tag{3.19}$$

Making use of the fact

$$\langle T_3 \phi_1 \phi_2 \rangle = -\frac{h\, z_{12}^2}{z_{23}^2 z_{13}^2} \langle \phi_1 \phi_2 \rangle, \qquad \langle \tilde{T}_4 \phi_1 \phi_2 \rangle = -4 \frac{h\, z_{41} \bar{z}_{12} z_{24}}{\bar{z}_{41} z_{12} \bar{z}_{24}} \langle \phi_1 \phi_2 \rangle = \langle \tilde{T}_4 \mathcal{B}_{12}^{(1)} \rangle \langle \phi_1 \phi_2 \rangle \tag{3.20}$$

and noting that expressing $\langle \tilde{T}_4 \phi_1 \phi_2 \rangle$ as $\bar{\partial}_4 \langle \epsilon_4 \phi_1 \phi_2 \rangle$ implies that only $\langle \epsilon_4 \phi_1 \phi_2 \rangle_{\text{shdw}}$ contributes to $\langle \tilde{T}_4 \phi_1 \phi_2 \rangle$. This is merely because $\langle \epsilon_4 \phi_1 \phi_2 \rangle_{\text{phys}} \sim \langle \epsilon_4 \mathcal{B}_{ij}^{(1)} \rangle_{\text{phys}} \langle \phi_1 \phi_2 \rangle$ does not contain inverse powers of $z_4$[13].

The terms in the *rhs* of (3.14) are directly related to the terms in the Ward identity (3.9). The constraint (3.17) is satisfied by all but the last term in (3.14) which is related to the last 2 terms in (3.9). Therefore the consistency condition satisfied by the physical part of $\mathcal{C}^{(2)}$ is

$$\partial_3^3 \partial_4^3 \mathcal{C}_{\text{phys}}^{(2)} = \frac{z^2 (2 - 2z + z^2)}{z_{34}^4 (z-1)^2}. \tag{3.21}$$

It turns out that $\mathcal{C}_{\text{phys}}^{(2)}$ can be determined to be

$$\mathcal{C}_{\text{phys}}^{(2)} = z_{34}^2 \mathcal{A}(z)$$

$$\mathcal{A}(z) = \frac{1}{16 z^2} \left[ (4z^2 + 2z - 5) \, \text{Li}_2 \left( \frac{1}{1-z} \right) + \left( z^2 + 8z - 5 \right) \text{Li}_2 (1-z) + 5 \left( z^2 - 1 \right) \text{Li}_2(z) \right.$$

$$+ 5 (2z-1) \text{Li}_2 \left( \frac{z}{z-1} \right) - 2 \left( z^2 - 6z + 6 \right) \log^2 (1-z) + 8 z^2 (\log(z-1) - 2 \log(z))$$

$$\left. + 5 \log(1-z)((2z-1)\log(z-1) + (z-2)z \log(z)) \right] \tag{3.22}$$

$\mathcal{A}(z)$ is explicitly symmetric under $(1 \leftrightarrow 2)$ & $(3 \leftrightarrow 4)$. Of course $\mathcal{A}$ is not uniquely determined but the ambiguities- which can be made explicit in the process of finding $\mathcal{A}$, do not contribute to anything physical. For example adding a function of the type

$$\frac{z_{34}^2}{z^2} \left( a_1 z^2 + a_2 z + a_3 + (b_1 z^2 + b_2 z + b_3) \log(1-z) \right) \tag{3.23}$$

---

[12]This requires having a term proportional to $\log \bar{z}$.

[13]Unlike the constraint for $\langle \epsilon_1 \epsilon_2 \rangle$ where the constraint $\langle T_1 \tilde{T}_2 \rangle = -\frac{3\pi}{c} \delta_{12}^{(2)} = \frac{3}{c} \bar{\partial}_2 \frac{1}{z_{12}}$ does constrain $\langle T_1 \epsilon_2 \rangle$.

to $\mathcal{A}(z)$ also satisfies the same consistency condition. Note one can similarly satisfy the constraint (3.18) too, but as mentioned before this would only give contributions to the shadow part in (3.14).

It must be noted that although $\langle \epsilon \epsilon \rangle$ themselves are not conformally invariant- $\epsilon$s transforming like an operator with conformal dimension $-1$; the 2pt functions of the bilinears constructed out of $\epsilon$s however are conformally invariant

$$\frac{\langle \phi_1 \phi_2 \phi_3 \phi_4 \rangle}{\langle \phi_1 \phi_2 \rangle \langle \phi_3 \phi_4 \rangle} = h_\phi^2 \langle \mathcal{B}_{12}^{(1)} \mathcal{B}_{34}^{(1)} \rangle. \tag{3.24}$$

One can also foresee that any computation of a block would involve $\mathcal{B}_{ij}^{(1)}$- as in the case of the 4pt stress-tensor block; and not $\epsilon_i$s themselves. Therefore we consider the insertions of the reparametrizations mode operator $\epsilon$s inside the bilocals $\mathcal{B}_{i,j}$ i.e.

$$\frac{\left\langle \mathcal{B}_{35}^{(1)} \mathcal{B}_{46}^{(1)} \phi_1 \phi_2 \right\rangle}{\langle \phi_1 \phi_2 \rangle}. \tag{3.25}$$

This can be readily seen as conformally invariant as it is obtained from a particular diagram contributing the global conformal block of 6-pt function of pair-wise equal operators:

$$\frac{\langle X_3 X_5 \, |T|_g \, \phi_1 \phi_2 \, |T|_g \, Y_4 Y_6 \rangle}{\langle \phi_1 \phi_2 \rangle \langle X_3 X_5 \rangle \langle Y_4 Y_6 \rangle} = \left( \frac{3}{\pi c} \right)^2 \int_{3',4'} \frac{\langle \tilde{T}_{3'} \tilde{T}_{4'} \phi_1 \phi_2 \rangle \langle T_{3'} X_3 X_5 \rangle \langle T_{4'} Y_4 Y_6 \rangle}{\langle \phi_1 \phi_2 \rangle \langle X_3 X_5 \rangle \langle Y_4 Y_6 \rangle}$$

$$= h_X h_Y \frac{\left\langle \mathcal{B}_{35}^{(1)} \mathcal{B}_{46}^{(1)} \phi_1 \phi_2 \right\rangle}{\langle \phi_1 \phi_2 \rangle} \tag{3.26}$$

where we use the sub-script $g$ in $|T|_g$ to denote projection onto global states associated with the stress-tensor. (From now on we use $|\mathcal{O}|$ to denote the full Virasoro block while $|\mathcal{O}|_g$ to denote just the global block associated with any operator $\mathcal{O}$.) This consists of a connected diagram[14] contributing to the vacuum block of the 6-pt pairwise equal operators and it only need be normalized by the 2-pt functions[15]. Using the definition of the $\epsilon$ as derivative of $\tilde{T}$ and then the Ward identity after integrating by parts as before, one is left with (3.25) upto proportionality constants which depend on the operator dimensions. Using (3.14) we can expand (3.25) as

$$\frac{\left\langle \mathcal{B}_{35}^{(1)} \mathcal{B}_{46}^{(1)} \phi_1 \phi_2 \right\rangle}{\langle \phi_1 \phi_2 \rangle} = \langle \mathcal{B}_{35}^{(1)} \mathcal{B}_{46}^{(1)} \rangle + h(h-1) \langle \mathcal{B}_{35}^{(1)} \mathcal{B}_{12}^{(1)} \rangle \langle \mathcal{B}_{12}^{(1)} \mathcal{B}_{46}^{(1)} \rangle + h \left( \frac{12}{c} \right)^2 \mathcal{K}^{(2)} \tag{3.27}$$

where $\mathcal{K}^{(2)}$ captures the contribution of the last term in (3.14). The first 2 terms are built of the familiar vacuum 4-pt conformal blocks and their contribution to the different OTOs can

---

[14]This expression does contain disconnected pieces, refer to subsection(3.1) for the discussion.

[15]We would have to explicitly remove the contribution coming from the first term in the Ward identity, this can be justified by an appropriate normalization $c.f.$ comment below (3.36)

be therefore readily discerned. $\mathcal{K}^{(2)}$ an be written as[16]

$$\mathcal{K}^{(2)} = \left[ \partial_3 \partial_4 - 2 \left( \frac{\partial_3}{z_{46}} + \frac{\partial_4}{z_{35}} \right) - \frac{2}{z_{35} z_{46}} \right] \mathcal{C}^{(2)} + (3 \leftrightarrow 5, 4 \leftrightarrow 6) \qquad (3.28)$$

We do not give the explicit expression for $\mathcal{K}^{(2)}$ as it would be too cumbersome. However we would choose to extract the behaviours of those functions which posses branch cuts in the conformally invariant cross ratios which we do in the next section. Before doing so we would like to understand what exactly would we be computing as in the 4pt case.

### 3.0.1   Non-linear contributions *via* holography

We take a small detour to note how the vacuum block of the 6pt function was computed using the reparametrization mode $\epsilon$ in [36] by Anous & Haehl. The authors made use of a non-linear generalization of $\mathcal{B}_{ij}^{(1)}$- denoted as $\mathcal{B}_{ij}$, inspired by the work of Cotler & Jensen in [22]. In [22] the authors derived an effective action for $CFT_2$ stress-tensor given in terms of $\mathcal{B}_{ij}$. In terms of which the connected contribution to the 6pt vacuum block is given as

$$\mathcal{V}_T^{(6)} = \frac{\langle \mathcal{B}_{12} \mathcal{B}_{35} \mathcal{B}_{46} \rangle \langle \mathcal{B}_{12} \rangle \langle \mathcal{B}_{46} \rangle \langle \mathcal{B}_{35} \rangle}{\langle \mathcal{B}_{12} \mathcal{B}_{46} \rangle \langle \mathcal{B}_{12} \mathcal{B}_{35} \rangle \langle \mathcal{B}_{35} \mathcal{B}_{46} \rangle} \bigg|_{\text{phys}} \qquad (3.29)$$

Their form is deduced by generalizing the conformal transformation of 2pt functions of primaries to arbitrary co-ordinate reparametrizations. $\mathcal{B}_{ij}$ is then defined as the ratio of 2pt functions of primaries in different frames.

$$z \to f(z, \bar{z}) = z + \epsilon(z, \bar{z}) + \mathcal{O}(\epsilon^2) \qquad (3.30)$$

$$\mathcal{B}_{12} \approx \mathcal{B}_{h,12} = z_{12}^{2h} \left( \frac{\partial f(z_1, \bar{z}_1) \partial f(z_2, \bar{z}_2)}{(f(z_1, \bar{z}_1) - f(z_2, \bar{z}_2))^2} \right)^h = 1 + \sum_{p \geq 1} \mathcal{B}_{12}^{(p)} \qquad (3.31)$$

$$\implies \mathcal{B}_{12}^{(1)} = h \left[ \partial \epsilon_1 + \partial \epsilon_2 - 2 \frac{(\epsilon_1 - \epsilon_2)}{z_{12}} \right] \qquad (3.32)$$

This allows the authors of [36] to have the first sub-leading correction ($\mathcal{O}(c^{-2})$ in this case) to come from truly connected 6pt diagram. It is important to note that (3.30) is not holomorphic and the effective action for stress-tensor propagation as derived in [22] is obtained from the gravitational path-integral in $AdS_3$. It is therefore plausible to expect that a formalism to compute higher point vacuum blocks must exist utilising the reparametrization modes $\epsilon$s but without recourse to holography.

We develop this formalism in the following sections along with a diagrammatic understanding of the terms involved in computing a particular $\mathcal{O}(c^{-n})$ contribution to the vacuum block. *We introduce a representation of the identity block projector built out of $T$ and $\tilde{T}$ in $CFT_2$ which allows us to see which diagrams contribute to the leading order in $1/c$ to the connected*

---

[16]The expression for $\mathcal{K}^{(2)}$ is separately symmetric under $3 \leftrightarrow 5$ and $4 \leftrightarrow 6$.

*vacuum block of an n-pt function of pairwise equal light operators.* It would also allow us to easily compute leading vacuum block contributions to operators of the form $\langle \phi\phi XXXYYY \rangle$, $\langle \phi\phi XXXYY \rangle$ and $\langle \phi\phi\phi XXXYYY \rangle$ from the building blocks already assembled for the 6pt vacuum block of $\langle \phi\phi XXYY \rangle$. We are also able to give a simple proof of how the leading order vacuum block for 4pt function of HHLL is obtained from a conformal transformation as first shown in [26].

## 3.1 Channel diagrammatics

We would next like to understand to what kind of correlation functions does the above expectation value (3.27) give an answer to. Taking a hint from the 4pt case one can easily see that it may contribute a Virasoro block of a 6pt function of 3 pairs of operators, one of the pairs being that of operator $\phi$. However since (3.27) gives a contribution to

$$\frac{\langle X_3 X_5 \, |T|_g \, \phi_1\phi_2 \, |T|_g \, Y_4 Y_6 \rangle}{\langle \phi_1\phi_2 \rangle \langle X_3 X_5 \rangle \langle Y_4 Y_6 \rangle} = \left( \frac{3}{\pi c} \right)^2 \int_{3',4'} \frac{\langle \tilde{T}_{3'} \tilde{T}_{4'} \phi_1\phi_2 \rangle \langle T_{3'} X_3 X_5 \rangle \langle T_{4'} Y_4 Y_6 \rangle}{\langle \phi_1\phi_2 \rangle \langle X_3 X_5 \rangle \langle Y_4 Y_6 \rangle}; \quad (3.33)$$

the $X_{3,5}$ & $Y_{4,6}$ external operator pairs do not fuse to exchange Virasoro states associated with the stress-tensor. This is because the conformal integral representation of the projector $|T|_g$ only allows projection onto global states: as the embedding space formalism doesn't ensure invariance under generic Virasoro transformations but only the the global conformal transformation. However we can easily show (Appendix D) that considering $X_{3,5}$ fusing to exchange Virasoro modes of the stress-tensor gives either a sub-leading contribution in the large $c$ limit or a contribution obtained from $\phi_{1,2} \leftrightarrow X_{3,5}$. Therefore the reparametrization mode Ward identity (3.27) contributes to the star channel upon symmetrization *wrt* exchange of external operator pairs. *In other words* (3.33) *gives the connected piece (star-channel) contribution of the 6-pt vacuum block for* $\langle X_3 X_5 \phi_1\phi_2 Y_4 Y_6 \rangle$ *at order* $1/c^2$ *upon symmetrization w.r.t. the three pairs* $\{X_{3,5}, \phi_{1,2}, Y_{4,6}\}$.

In this and the following subsections we develop a representation of the vacuum block projector in a power series expansion in $1/c$. This naturally allows us to use the results of the previous subsection to compute connected contribution to vacuum blocks of 6pt functions of pairwise equal operators at leading order in $1/c$. It would also- among other things, allow us to generalize this 6pt computation to simpler 8pt ($\langle X_3 X_5 X_7 \phi_1\phi_2 Y_4 Y_6 Y_8 \rangle$) and 9pt ($\langle X_3 X_5 X_7 \phi_0\phi_1\phi_2 Y_4 Y_6 Y_8 \rangle$) vacuum blocks.

Let us expand the 4pt and 6pt functions in sub-leading orders of $1/c$- where we assume that the operator dimensions are $h_\phi \sim h_X \sim h_Y \ll c$. We do this by inserting stress tensor projectors

$$|T|_g = \frac{3}{\pi c} \int_x |\tilde{T}(x)\rangle \langle T(x)| = |T^{(1)}| \tag{3.34}$$

with each projector bringing a factor of $1/c$. We indeed imagine the vacuum block projector as

$$|\mathbb{I}| = \rangle\langle + \alpha \int_0 T_0\rangle\langle \tilde{T}_0 + \cdots = I + |T^{(1)}| + \dots \qquad (3.35)$$

where $\alpha = 3/\pi c$ and the first term is simply the identity exchange while the $\dots$ are projectors which give corrections of $\mathcal{O}(1/c^{n\geq 2})$. The above projector is only useful in projecting onto states which contribute upto $\mathcal{O}(1/c)$. The validity of the second term in (3.35) is proved by showing that it squares to itself and that it indeed gives the expected vacuum block at $\mathcal{O}(1/c)$ for the simple case of 4pt fucntions of 2 pairs of light operators [1]. We would return the higher order terms in (3.35) in $1/c$ shortly.

Insertion of each such projector results in the propagation of $\epsilon$s with powers of $1/c$ as given in (2.21) and (3.14). The 4pt functions of light operators can therefore be expanded as[17]

$$\langle \phi_1\phi_2 X_3 X_5\rangle_{\text{vac}} = \langle \phi_1\phi_2|\mathbb{I}|X_3 X_5\rangle = \langle \phi_1\phi_2\rangle\langle X_3 X_5\rangle \left( 1 + h_\phi h_X \langle \mathcal{B}_{12}^{(1)} \mathcal{B}_{35}^{(1)}\rangle + \dots \right) \quad (3.36)$$

valid upto $\mathcal{O}(1/c)$ with the $\dots$ above denoting contributions at higher powers in $1/c$. A similar insertion of (3.35) in the case of 6pt function of pair wise equal light operators yields contributions from dis-connected and connected diagrams

$$\langle X_3 X_5 \phi_1\phi_2 Y_4 Y_6\rangle_{\text{vac}} = \langle X_3 X_5|\mathbb{I}|\phi_1\phi_2|\mathbb{I}|Y_4 Y_6\rangle$$

$$= \langle X_3 X_5\rangle\langle \phi_1\phi_2\rangle\langle Y_4 Y_6\rangle + \langle X_3 X_5|T^{(1)}|\phi_1\phi_2\rangle\langle Y_4 Y_6\rangle + \langle X_3 X_5\rangle\langle \phi_1\phi_2|T^{(1)}|Y_4 Y_6\rangle$$

$$+ \langle X_3 X_5|T^{(1)}|\phi_1\phi_2|T^{(1)}|Y_4 Y_6\rangle + \dots \qquad (3.37)$$

with $\dots$ implying contributions from higher order terms in (3.35), this would *not* necessarily imply that they do not contribute at order $1/c$ & $1/c^2$. Using the definition of the projectors we find

$$\langle X_3 X_5 \phi_1\phi_2 Y_4 Y_6\rangle_{\text{vac}} = \langle \phi_1\phi_2\rangle\langle X_3 X_5\rangle\langle Y_4 Y_6\rangle \left( 1 + h_\phi h_X \langle \mathcal{B}_{12}^{(1)} \mathcal{B}_{35}^{(1)}\rangle + h_\phi h_Y \langle \mathcal{B}_{12}^{(1)} \mathcal{B}_{46}^{(1)}\rangle \right.$$

$$\left. + h_X h_Y \frac{\langle \mathcal{B}_{35}^{(1)} \mathcal{B}_{46}^{(1)} \phi_1\phi_2\rangle}{\langle \phi_1\phi_2\rangle} + \dots \right), \qquad (3.38)$$

One can easily see that the disconnected pieces- in the first line above, are not symmetric *w.r.t.* the 3 pairs of operators, this is also true for the seemingly connected contribution written explicitly as $h_X h_Y \frac{\langle \mathcal{B}_{35}^{(1)} \mathcal{B}_{46}^{(1)} \phi_1\phi_2\rangle}{\langle \phi_1\phi_2\rangle}$. The symmetry in the disconnected terms is indeed restored by noting that the last term in (3.38) consists of an order $1/c$ term (*i.e.* a disconnected term) coming from the first term in (3.27) and (3.14) which makes the $\mathcal{O}(1/c)$ terms in the 6pt

---

[17]1pt functions of $\epsilon$s are assumed to vanish in flat background.

function above symmetric in terms of 3 pairs of operators. This contribution is simply the fusing of the 2 stress-tensors in the evaluation of

$$\langle X_3 X_5 | T^{(1)} | \phi_1 \phi_2 | T^{(1)} | Y_4 Y_6 \rangle = \alpha^2 \int_{0,-1} \langle X_3 X_5 \tilde{T}_0 \rangle \langle T_0 \phi_1 \phi_2 T_{-1} \rangle \langle \tilde{T}_{-1} Y_4 Y_6 \rangle \qquad (3.39)$$

in (3.37) coming from an $\mathcal{O}(c)$ term in $\langle T_0 \phi_1 \phi_2 T_{-1} \rangle$ *i.e.* $\langle T_{-1} T_0 \rangle \langle \phi_1 \phi_2 \rangle$. Put simply

$$\langle X_3 X_5 | T^{(1)} | \phi_1 \phi_2 | T^{(1)} | Y_4 Y_6 \rangle = \langle \phi_1 \phi_2 \rangle \langle X_3 X_5 | T^{(1)} | Y_4 Y_6 \rangle + \langle X_3 X_5 | T^{(1)} | \phi_1 \phi_2 | T^{(1)} | Y_4 Y_6 \rangle_{conn.}$$
$$(3.40)$$

by using $|T^{(1)}|^2 = |T^1|$.

In order to see how the symmetry in the connected component of (3.37) & (3.38) is restored we would have to expand the vacuum block projector (3.35) to higher orders. We therefore claim that the $1/c$ expansion of the vacuum block projector takes the form

$$\boxed{|\mathbb{I}| = \rangle\langle + \alpha \int_0 T_0 \rangle\langle \tilde{T}_0 + \frac{\alpha^2}{2!} \int_{0,-1} T_{-1} T_0 \rangle\langle \tilde{T}_0 \tilde{T}_{-1}, + \frac{\alpha^3}{3!} \int_{0,-1,-2} T_{-2} T_{-1} T_0 \rangle\langle \tilde{T}_0 \tilde{T}_{-1} \tilde{T}_{-2} + \dots}$$

$$|\mathbb{I}| \equiv \sum_{n=0} \frac{\alpha^n}{n!} \int_{-1,-2,\dots,n} T_{-1} T_{-2} \dots T_{-n} \rangle\langle \tilde{T}_{-n} \dots \tilde{T}_{-2} \tilde{T}_{-1} \quad \text{with } \alpha = \frac{3}{\pi c} \qquad (3.41)$$

*where one is not supposed to consider the contribution due to fusing of any 2 of the $T_i$s or $\tilde{T}_i$s.* The above expansion is in powers of $\alpha \sim 1/c$. The third term at $\mathcal{O}(c^{-2})$ is required to make the connected part of the 6pt vacuum block symmetric at $1/c^2$. For the computation of vacuum blocks of higher pt. correlators considered in this paper we do not require higher order terms written above. However their form similarly makes the connected part of the vacuum block symmetric *w.r.t.* operator pairs at all orders in the $1/c$ expansion.

We back this claim of the vacuum block projector by showing that each of the terms above square to themselves and therefore are valid projectors. However we would also have to prove that each of the above terms do not project onto a smaller subspace of states that contribute at the required order in $1/c$. This we only show for the all the terms above to leading orders in operator dimension in a 4pt vacuum block of light operators. As the projector is constructed out of the shadow formalism, one has to always take the physical part of the expression after evaluation. We prove these in a following subsection 3.3 and only concern ourselves here with how the symmetry in the disconnected component in (3.37) is restored.

We write the vacuum block projector as

$$|\mathbb{I}| = I + |T^{(1)}| + |T^{(2)}| + \dots \qquad (3.42)$$

where $|T^{(2)}| = \frac{\alpha^2}{2!} \int_{0,-1} T_{-1} T_0 \rangle\langle \tilde{T}_0 \tilde{T}_{-1}$. Plugging this in the *rhs* of (3.37) and using (3.40) we have

$$\langle X_3 X_5 \phi_1 \phi_2 Y_4 Y_6 \rangle_{\text{vac}} = \langle X_3 X_5 | \mathbb{I} | \phi_1 \phi_2 | \mathbb{I} | Y_4 Y_6 \rangle$$

$$= \langle X_3 X_5 \rangle \langle \phi_1 \phi_2 \rangle \langle Y_4 Y_6 \rangle + \langle X_3 X_5 | T^{(1)} | \phi_1 \phi_2 \rangle \langle Y_4 Y_6 \rangle + \langle X_3 X_5 \rangle \langle \phi_1 \phi_2 | T^{(1)} | Y_4 Y_6 \rangle$$

$$+ \langle \phi_1 \phi_2 \rangle \langle X_3 X_5 | T^{(1)} | Y_4 Y_6 \rangle + \mathcal{O}(c^{-2})_{disconn.}$$

$$+ \langle X_3 X_5 | T^{(1)} | \phi_1 \phi_2 | T^{(1)} | Y_4 Y_6 \rangle_{conn.}$$

$$+ \langle X_3 X_5 | T^{(2)} | \phi_1 \phi_2 | T^{(1)} | Y_4 Y_6 \rangle + \langle X_3 X_5 | T^{(1)} | \phi_1 \phi_2 | T^{(2)} | Y_4 Y_6 \rangle + \ldots \quad (3.43)$$

The last 2 terms above go to each other upon $X_{3,5} \leftrightarrow Y_{4,6}$. Lets take the first of these- $\langle X_3 X_5 | T^{(2)} | \phi_1 \phi_2 | T^{(1)} | Y_4 Y_6 \rangle$, we note that just like in the case of $\langle X_3 X_5 | T^{(1)} | \phi_1 \phi_2 | T^{(1)} | Y_4 Y_6 \rangle$ the 2 $T_i$s belonging to 2 different projector insertions can fuse together. Indeed that's the only kind of fusing of $T_i$s (or $\tilde{T}_i$s) allowed according to the construction (3.41). Further the fusing reduces the power of $c$ in the denominator as the last 2 terms above come with factors of $\alpha^3 = (3/\pi c)^3$. Let's analyse this term explicitly

$$\langle X_3 X_5 | T^{(2)} | \phi_1 \phi_2 | T^{(1)} | Y_4 Y_6 \rangle = \frac{\alpha^3}{2!} \int_{-2,-1,0} \langle X_3 X_5 \tilde{T}_{-1} \tilde{T}_0 \rangle' \langle T_0 T_{-1} \phi_1 \phi_2 T_{-2} \rangle' \langle \tilde{T}_{-2} Y_4 Y_6 \rangle \quad (3.44)$$

where the primes denotes that $T_0$ & $T_{-1}$ cannot fuse. The $\mathcal{O}(c)$ terms arise from $\langle T_0 T_{-1} \phi_1 \phi_2 T_{-2} \rangle$ when $T_{-1,0}$ fuse with $T_{-2}$ *i.e.*

$$\langle T_0 T_{-1} \phi_1 \phi_2 T_{-2} \rangle' = \langle T_{-1} T_{-2} \rangle \langle T_0 \phi_1 \phi_2 \rangle + \langle T_0 T_{-2} \rangle \langle T_{-1} \phi_1 \phi_2 \rangle + \mathcal{O}(c^0) \quad (3.45)$$

This implies that the $\mathcal{O}(c^{-2})$ component of $\langle X_3 X_5 | T^{(2)} | \phi_1 \phi_2 | T^{(1)} | Y_4 Y_6 \rangle$ is

$$\langle X_3 X_5 | T^{(2)} | \phi_1 \phi_2 | T^{(1)} | Y_4 Y_6 \rangle_{c^{-2}} = \frac{\alpha^3}{2!} \left\{ \int_{-2,-1,0} \langle X_3 X_5 \tilde{T}_{-1} \tilde{T}_0 \rangle' \langle T_{-1} T_{-2} \rangle \langle \tilde{T}_{-2} Y_4 Y_6 \rangle \langle T_0 \phi_1 \phi_2 \rangle \right.$$

$$\left. + \int_{-2,-1,0} \langle X_3 X_5 \tilde{T}_{-1} \tilde{T}_0 \rangle' \langle T_0 T_{-2} \rangle \langle \tilde{T}_{-2} Y_4 Y_6 \rangle \langle T_{-1} \phi_1 \phi_2 \rangle \right\}$$

$$= \frac{\alpha^3}{2!} \left\{ \int_{-2,-1,0} \langle X_3 X_5 \tilde{T}_{-1} \tilde{T}_0 \rangle' \langle T_{-1} \tilde{T}_{-2} \rangle \langle T_{-2} Y_4 Y_6 \rangle \langle T_0 \phi_1 \phi_2 \rangle + \int_{-2,-1,0} \langle X_3 X_5 \tilde{T}_{-1} \tilde{T}_0 \rangle' \langle T_0 \tilde{T}_{-2} \rangle \langle T_{-2} Y_4 Y_6 \rangle \langle T_{-1} \phi_1 \phi_2 \rangle \right\}$$

$$(3.46)$$

where in the second equation we $T_{-2} \leftrightarrow \tilde{T}_{-2}$. Using (2.19) the fact that $\langle T_i \tilde{T}_{-2} \rangle = \frac{c\pi}{3} \delta^2(x_i - x_{-2}) \equiv \alpha^{-1} \delta^2_{-2,i}$, we can do the conformal integral over $x_{-2}$ yielding

$$\langle X_3 X_5 | T^{(2)} | \phi_1 \phi_2 | T^{(1)} | Y_4 Y_6 \rangle_{c^{-2}} = \alpha^2 \int_{-1,0} \langle T_0 \phi_1 \phi_2 \rangle \langle X_3 X_5 \tilde{T}_{-1} \tilde{T}_0 \rangle' \langle T_{-1} Y_4 Y_6 \rangle$$

$$= \langle \phi_1 \phi_2 | T^{(1)} | X_3 X_5 | T^{(1)} | Y_4 Y_6 \rangle_{conn.} \quad (3.47)$$

Note that in the *rhs* above we end up with a connected component as the (shadow) stress-tensors $T_{-1}$ & $T_0$ are not supposed to fuse into each other. Using the above expression for $\langle X_3 X_5 \phi_1 \phi_2 Y_4 Y_6 \rangle_{\text{vac}}$ in the *rhs* of (3.43) we get

$$\langle X_3 X_5 \phi_1 \phi_2 Y_4 Y_6 \rangle_{\text{vac}} = \langle X_3 X_5 | \mathbb{I} | \phi_1 \phi_2 | \mathbb{I} | Y_4 Y_6 \rangle$$

$$= \langle X_3 X_5 \rangle \langle \phi_1 \phi_2 \rangle \langle Y_4 Y_6 \rangle + \langle X_3 X_5 | T^{(1)} | \phi_1 \phi_2 \rangle \langle Y_4 Y_6 \rangle + \langle X_3 X_5 \rangle \langle \phi_1 \phi_2 | T^{(1)} | Y_4 Y_6 \rangle$$

$$+ \langle \phi_1 \phi_2 \rangle \langle X_3 X_5 | T^{(1)} | Y_4 Y_6 \rangle + \mathcal{O}(c^{-2})_{disconn.}$$

$$+ \langle X_3 X_5 | T^{(1)} | \phi_1 \phi_2 | T^{(1)} | Y_4 Y_6 \rangle_{conn.} + \langle \phi_1 \phi_2 | T^{(1)} | X_3 X_5 | T^{(1)} | Y_4 Y_6 \rangle_{conn.}$$

$$+ \langle \phi_1 \phi_2 | T^{(1)} | Y_4 Y_6 | T^{(1)} | X_3 X_5 \rangle_{conn.} + \mathcal{O}(c^{-3})$$
$$\tag{3.48}$$

where we have suppressed the connected contributions from $\mathcal{O}(c^{-3})$ onwards and disconnected contributions from $\mathcal{O}(c^{-2})$ onwards. It is thus apparent that the first non-trivial connected contribution to the $\langle X_3 X_5 \phi_1 \phi_2 Y_4 Y_6 \rangle_{\text{vac}}$ is obtained from symmetrising

$$\langle X_3 X_5 | T^{(1)} | \phi_1 \phi_2 | T^{(1)} | Y_4 Y_6 \rangle_{conn.} \sim \mathcal{O}(1/c^2) \tag{3.49}$$

*wrt* the 3 pairs of operators. Further, it shows that every order in $1/c$ the contributions are symmetric *wrt* the 3 pairs of operators and is thus independent of how the vacuum projector $|\mathbb{I}|$ (3.41) is inserted in the evaluation of the 6pt function of pair-wise equal operators.

The contribution of (3.27) also consists of a product 4pt functions which the authors of [36] claim is cancelled by the normalization (3.29). However we study $\mathcal{O}(c^{-2})$ contribution by the last term in (3.37) recognising it to contain the truly connected component at this order as shown in (3.48). Therefore writing the connected contribution to $\langle X_3 X_5 \phi_1 \phi_2 Y_4 Y_6 \rangle_{\text{vac}}$ at $\mathcal{O}(c^{-2})$ using (3.48) & (3.26) we have

$$\boxed{V_{T,\text{conn}}^{(6)} \approx h_X h_Y \left. \frac{\langle \mathcal{B}_{35}^{(1)} \mathcal{B}_{46}^{(1)} \phi_1 \phi_2 \rangle_{c^{-2}}}{\langle \phi_1 \phi_2 \rangle} \right|_{\text{phys}} + (\phi_{1,2} \leftrightarrow X_{3,5}) + (\phi_{1,2} \leftrightarrow Y_{4,6}) + \dots} \tag{3.50}$$

where we retain only the $1/c^2$ terms in (3.27) and have normalized the *lhs* of (3.37) using the product of 2pt functions.

In what follows in section 4, we will only consider the first term of the *rhs* above for further analysis assuming that it can be easily generalized when the expressions are symmetrized for the exchange of the pairs of operators. We also believe that in order make contact with the final expression obtained by [36] for the 6pt vacuum block one needs to retain terms proportional to $h h_X h_Y$ in (3.50) above while dropping terms quadratic in any of the operator dimensions *i.e.* $h^2 h_X h_Y$, $h h_X^2 h_Y$ etc. While the authors of [36] manage to do this by employing the normalization (3.29) using $\mathcal{B}_{ij}$s, it would be interesting to figure out an equivalent normalization using expectation values of operators in the theory.

### 3.1.1 Star and Comb channels

We would like to see which channel diagrams we are computing by evaluating the above expression. We can write the quantity we wish to evaluate as (3.26)

$$
h_X h_Y \frac{\left\langle \mathcal{B}_{35}^{(1)} \mathcal{B}_{46}^{(1)} \phi_1 \phi_2 \right\rangle}{\langle \phi_1 \phi_2 \rangle} = \left( \frac{3}{\pi c} \right)^2 \int_{3',4'} \frac{\langle \tilde{T}_{3'} \tilde{T}_{4'} \phi_1 \phi_2 \rangle \langle T_{3'} X_3 X_5 \rangle \langle T_{4'} Y_4 Y_6 \rangle}{\langle \phi_1 \phi_2 \rangle \langle X_3 X_5 \rangle \langle Y_4 Y_6 \rangle}
$$
$$
= \frac{\langle X_3 X_5 | T^{(1)} | \phi_1 \phi_2 | T^{(1)} | Y_4 Y_6 \rangle}{\langle X_3 X_5 \rangle \langle \phi_1 \phi_2 \rangle \langle Y_4 Y_6 \rangle} \tag{3.51}
$$

The numerator of the above integrand in the *rhs* can be seen as equivalent to

$$
\int_{3',4'} \langle \tilde{T}_{3'} \tilde{T}_{4'} | \mathbb{I} | \phi_1 \phi_2 \rangle \langle T_{3'} X_3 X_5 \rangle \langle T_{4'} Y_4 Y_6 \rangle = \int_{3',4'} \langle \tilde{T}_{3'} \tilde{T}_{4'} \rangle \langle \phi_1 \phi_2 \rangle \langle T_{3'} X_3 X_5 \rangle \langle T_{4'} Y_4 Y_6 \rangle
$$
$$
+ \int_{3',4'} \langle \tilde{T}_{3'} \tilde{T}_{4'} | T | \phi_1 \phi_2 \rangle \langle T_{3'} X_3 X_5 \rangle \langle T_{4'} Y_4 Y_6 \rangle \tag{3.52}
$$

where $|T| = |\mathbb{I}| - I$. This yields the 6-pt "star" channel plus disconnected diagram which is $\mathcal{O}(1/c)$[18]. At this point we can make contact with the diagrams computed in [36] as the 6-pt "star"-channel diagrams. We note that using only the first term in the $1/c$ expansion of $|T|$ *i.e.* $|T^{(1)}| = \alpha \int_0 T_0 \rangle \langle \tilde{T}_0$ we get what the authors of [36] call the global part of the "star"-channel

$$
\int_{3',4'} \langle T_{3'} T_{4'} | T | \phi_1 \phi_2 \rangle \langle \tilde{T}_{3'} X_3 X_5 \rangle \langle \tilde{T}_{4'} Y_4 Y_6 \rangle \to \alpha \int_{3',4',0} \langle T_{3'} T_{4'} T_0 \rangle \langle \tilde{T}_0 \phi_1 \phi_2 \rangle \langle \tilde{T}_{3'} X_3 X_5 \rangle \langle \tilde{T}_{4'} Y_4 Y_6 \rangle. \tag{3.53}
$$

However this isn't the full contribution as the *rhs* above goes as $\sim h_\phi$ while we know that the *lhs* has terms due to $\langle T_{3'} T_{4'} \phi_1 \phi_2 \rangle$ which scale as $\sim h_\phi^2$. The authors of [36] fix this by invoking non-linear versions of $\mathcal{B}_{1,2}^{(1)}$ operators inspired by work of Cotler and Jensen [22] who obtain an effective action in terms of $\mathcal{B}_{1,2}$ from the on-shell partition function in $AdS_3$ gravity. We on the other hand proceed to evaluate (3.52) using just the Ward identity of associated with $\epsilon$ insertions *i.e.* using (3.26),(3.27).

A part of the "comb"-channel on the other hand can be described by the right diagram in Fig:1 where we analyse the channel where $X_{3,5}$ & $Y_{4,6}$ necessarily fuse into the stress-tensor. This can be called the global $T$ block of the "comb" channel [36]. This can be seen by demanding that $\phi_1 \to X_{3,5}$ while $\phi_2 \to Y_{4,6}$ in $\langle X_3 X_5 \phi_1 \phi_2 Y_4 Y_6 \rangle_{vac}$[19]. The only states that can be formed from fusing stress-tensor and $\phi$s are the descendants of $\phi$. Therefore we find the global $T$ block of the "comb" channel by inserting $|\phi|$ as

---

[18]This is precisely the first term in the Ward identity (3.27).

[19]The general comb channel does contain arbitrary primaries propagating in all internal lines but we consider this specific case, this particular comb channel was also considered in [36].

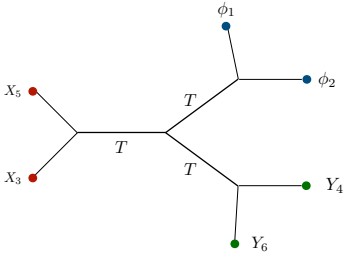

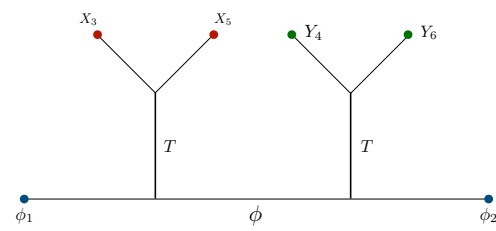

(a) Part of the star channel contribution

(b) Global stress-tensor block of the comb channel

**Figure 1**: The channels.

$$\int_{3',4'} \langle \tilde{T}_{3'} \tilde{T}_{4'} \phi_1 \phi_2 \rangle \langle T_{3'} X_3 X_5 \rangle \langle T_{4'} Y_4 Y_6 \rangle = \langle X_3 X_5 | T^{(1)} | \phi_1 \phi_2 | T^{(1)} | Y_4 Y_6 \rangle$$

$$= \langle X_3 X_5 | T^{(1)} | \phi_1 | \phi | \phi_2 | T^{(1)} | Y_4 Y_6 \rangle. \qquad (3.54)$$

where $|\phi|$ denotes the full Virasoro block projector of the primary $\phi$. Here again we have used the fact that insertion of $|\phi|$ above does not in any way restrict the states propagating between $\phi_1$ and $\phi_2$ in $\langle X_3 X_5 | T | \phi_1 \phi_2 | T | Y_4 Y_6 \rangle$. Therefore the 6-pt Virasoro "comb"-channel in Fig1 is the same as the "star" channel. It is appropriate to mention here that the global comb channel- where all projectors project onto to the global states associated with the stress tensor and internal (comb) primary operators; has been computed in [51].

## 3.2 Higher pt. functions

As we begin to analyse higher point functions the possible number of channels escalates quite fast. We would like to see what it means to analyse a possible "star"-channel when we insert the vacuum projector (3.41) in between $n$-pairs of identical operators for $n > 3$. We would be interested in the first sub-leading contribution in $1/c$ coming from a connected vacuum block of a pair-wise equal $2n_{(>3)}$-pt. function. This would particularly be useful in the holographic context wherein the dual scalars in the bulk interact with each other only via minimal coupling to gravity.

In order to make the analysis less cumbersome we make use of digrams as a shorthand for the kind of arguments given in the previous sub-section. We make use of

$$\equiv \langle X_3 X_5 | T^{(1)} | \phi_1 \phi_2 | T^{(1)} | Y_4 Y_6 \rangle, \qquad \equiv \langle X_3 X_5 | T^{(2)} | \phi_1 \phi_2 | T^{(1)} | Y_4 Y_6 \rangle$$

$$(3.55)$$

where $T^{(1)}, T^{(2)}, \ldots, T^{(n)}$ are as defined in (3.41). The first diagram above has a contribution which scales as $\mathcal{O}(1/c^{-2})$- which is the connected contribution; it also has a term which scales as $\mathcal{O}(1/c)$ obtained from the contraction of the 2 stress-tensor legs meeting at the fusion of $\phi_{1,2}$ as explained below (3.38). This is denoted in the similar vein as the first diagram in (3.56). Similarly the second diagram in (3.55) has a contribution which scales as $\mathcal{O}(1/c^3)$ while another contribution which is similarly obtained by fusing of either of the 2 stress-tensor legs coming to $\phi_{1,2}$ from $X_{3,5}$ with the one coming from $Y_{4,6}$- denoted by the second diagram in (3.56)

$$\equiv \langle \phi_1 \phi_2 \rangle \langle X_3 X_5 | T^{(1)} | Y_4 Y_6 \rangle, \qquad \equiv \langle \phi_1 \phi_2 | T^{(1)} | X_3 X_5 | T^{(1)} | Y_4 Y_6 \rangle_{conn.}$$

(3.56)

Note that the second diagram above is a connected contribution- one should not further contract the legs coming onto $X_{3,5}$ fore reasons mentioned between (3.43) and (3.47) in the previous sub-section. Further, the 2 possible ways of contracting the stress-tensor legs between $\phi_{1,2} - X_{3,5}$ with the one between $\phi_{1,2} - Y_{4,6}$ is absorbed by the $1/2!$ in the definition of $|T^{(2)}|$. Therefore it can be seen that

(3.57)

as was shown in the previous section in argument leading up to (3.48).

### 3.2.1  8pt and higher point functions

We next turn to leading connected contribution to pair-wise equal 8-pt function of light operators. Here the connected contribution occurs at $\mathcal{O}(c^{-3})$. Taking a hint from the 6pt example it is possible to guess that contributions to the leading connected vacuum block may look like

(3.58)

Let's check this claim by employing (3.41), in particular $|T| = |\mathbb{I}| - I$ as we are interested in connected contributions in $\langle X_3 X_5 \phi_1 \phi_2 Z_7 Z_9 Y_4 Y_6 \rangle_{vac.}$. Therefore insertion of $|\mathbb{I}|$ in between the pairs gives a sum

$$\langle X_3 X_5 \phi_1 \phi_2 Z_7 Z_9 Y_4 Y_6 \rangle_{vac.} = \sum_{n,m,l=0}^{\infty} \langle X_3 X_5 | T^{(n)} | \phi_1 \phi_2 | T^{(m)} | Z_7 Z_9 | T^{(l)} | Y_4 Y_6 \rangle_{vac.} \qquad (3.59)$$

containing even disconnected contributions. We would like to collect the connected terms in the above sum which contribute to the 8-pt vacuum block at $\mathcal{O}(c^{-3})$. This seems rather involved as terms in the above sum with $|T^{(i \geq 2)}|$ can allow for contractions thus contributing with lower powers in $1/c$. We do this in appendix E and show that each of the diagrams in the *rhs* of (3.58) is produced exactly once.

It is important to note that the symmetry between exchange of any pairs of operators is expected to hold at every order in the $1/c$ expansion and is independent of how the pairs are arranged. Thus one can easily speculate how an arbitrary $2n$-pt "star" channel would look like in terms of these diagrams at leading order

$$+ \quad \text{symm}_{\phi, X_2, \dots X_n} \qquad (3.60)$$

The digrams on the *rhs* are basically all possible connections such that there is unique path[20] between any of the pair of operators. All the above diagrams basically contain only $|T^{(1)}|$ and therefore contribute at $\mathcal{O}(c^{n-1})$. It would be interesting to compute their exact contributions especially in the context of maximally braided out of time ordered correlators. We leave this exercise for the near future.

### 3.2.2 Simpler 8pt and 9pt functions

Simpler higher pt functions of the form $\langle X_3 X_5 X_7 \phi_1 \phi_2 Y_4 Y_6 Y_8 \rangle$ and $\langle X_3 X_5 X_7 \phi_0 \phi_1 \phi_2 Y_4 Y_6 Y_8 \rangle$ would require only 2 insertions of the vacuum block projector $|\mathbb{I}|$ just as in the 6pt case analysed in the previous subsection. This is of course made possible as both 2-pt & 3-pt. functions are completely restricted by global conformal invariance. For example using the projector (3.41) to obtain the first sub-leading vacuum block for $\langle \phi_{1,2,3} \psi_{4,5} \rangle$ we find

$$\langle \phi_{1,2,3} | \mathbb{I} | \psi_{4,5} \rangle = \langle \phi_{1,2,3} \rangle \langle \psi_{4,5} \rangle + \alpha \int_0 \langle \phi_{1,2,3} T_0 \rangle \langle \tilde{T}_0 \psi_{4,5} \rangle$$

---

[20]The path should trace every leg only once.

$$= \langle \phi_{1,2,3} \rangle \langle \psi_{4,5} \rangle + \alpha^2 \int_{0,-1} \partial_0 \langle \phi_{1,2,3} T_0 \rangle \langle \tilde{T}_0 \tilde{T}_{-1} \rangle \langle T_{-1} \psi_{4,5} \rangle$$

$$= \langle \phi_{1,2,3} \rangle \langle \psi_{4,5} \rangle + \alpha^2 \frac{h_\phi}{2} h_\psi \langle \phi_{1,2,3} \rangle \langle \mathcal{B}^{(1)}_{1,2,3} \mathcal{B}^{(1)}_{3,4} \rangle \langle \psi_{4,5} \rangle \tag{3.61}$$

where in going from the first line to the second line we did the usual manipulation of using $(T^{(1)})^2 = T^{(1)}$, and while going from the second line to the third we simply used the Ward identity for 3pt. and 2pt functions as before. The expression for this new quantity $\mathcal{B}^{(1)}_{1,2,3}$ can be easily found from the 3pt. Ward identity for $\langle \phi_{1,2,3} T_0 \rangle$ to be

$$\langle T_0 \phi_1 \phi_2 \phi_3 \rangle = \sum_{i-1}^{3} \left( \frac{h_\phi}{z_{0i}^2} + \frac{\partial_i}{z_{0i}} \right) \langle \phi_1 \phi_2 \phi_3 \rangle$$

$$= \frac{h_\phi}{2} \left\langle (\mathcal{B}^{(1)}_{12} + \mathcal{B}^{(1)}_{23} + \mathcal{B}^{(1)}_{31}) T_0 \right\rangle \langle \phi_1 \phi_2 \phi_3 \rangle \tag{3.62}$$

where we have expressed it in terms of the Ward identity for 2pt. function in turn expressed in terms of $\mathcal{B}^{(1)}_{ij}$. The connected component of the pair-wise equal 6pt. vacuum block (3.50) requires the knowledge of $\langle \epsilon_i \epsilon_j \rangle$ along with $\langle \epsilon_i \epsilon_j \phi_{1,2} \rangle$. If one were to replace any of the pairs with a triplet then one can easily check that the whole computation of the previous section goes through and one can compute the connected component of vacuum block at the same order provided one knows $\langle \epsilon_1 \phi_{1,2,3} \rangle$ and $\langle \epsilon_i \epsilon_j \phi_{1,2,3} \rangle$. The terms obtained from the insertions of $|\mathbb{I}|$ in such cases can easily seen to be similarly represented by the digrams introduced in the previous subsection with the bold dots representing the fusion of 3 of the operators into the stress-tensor. Thus the leading order connected vacuum block contributions for $\langle X_3 X_5 X_7 \phi_1 \phi_2 Y_4 Y_6 Y_8 \rangle$ and $\langle X_3 X_5 X_7 \phi_0 \phi_1 \phi_2 Y_4 Y_6 Y_8 \rangle$ are again represented as

$$\tag{3.63}$$

The only missing ingredient for the above computations are the expectation values of the form $\langle \epsilon_{-1} \phi_0 \phi_1 \phi_2 \rangle$ and $\langle \epsilon_{-1} \epsilon_{-2} \phi_0 \phi_1 \phi_2 \rangle$ as these would be required in the computation of

$$\langle \phi_1 \phi_2 | T^{(1)} | X_3 X_5 X_7 | T^{(1)} | Y_4 Y_6 Y_8 \rangle_{conn.} = \alpha^2 \int_{-1,-2} \langle \phi_1 \phi_2 T_{-1} \rangle \langle \tilde{T}_{-1} X_3 X_5 X_7 \tilde{T}_{-2} \rangle \langle T_{-2} Y_4 Y_6 Y_8 \rangle_{conn.}$$

$$\langle X_3 X_5 X_7 | T^{(1)} | \phi_0 \phi_1 \phi_2 | T^{(1)} | Y_4 Y_6 Y_8 \rangle_{conn.} = \alpha^2 \int_{-1,-2} \langle X_3 X_5 X_7 T_{-1} \rangle \langle \tilde{T}_{-1} \phi_0 \phi_1 \phi_2 \tilde{T}_{-2} \rangle \langle T_{-2} Y_4 Y_6 Y_8 \rangle_{conn.}$$

$$\tag{3.64}$$

where the suffix *conn.* implies we do not take contributions coming from contractions of $T_i$s (or $\tilde{T}_i$s) belonging to different projector insertions. $\langle \epsilon_{-1} \phi_0 \phi_1 \phi_2 \rangle$ can easly be expressed in

terms of the bi-linear $\mathcal{B}_{ij}^{(1)}$ by first noting that

$$\langle T_{-1}\phi_0\phi_1\phi_2\rangle = \tfrac{h}{2}\left\langle\left(\mathcal{B}_{01}^{(1)}+\mathcal{B}_{12}^{(1)}+\mathcal{B}_{02}^{(1)}\right)T_{-1}\right\rangle\langle\phi_0\phi_1\phi_2\rangle$$

$$\implies \langle\epsilon_{-1}\phi_0\phi_1\phi_2\rangle = \tfrac{h}{2}\left\langle\left(\mathcal{B}_{01}^{(1)}+\mathcal{B}_{12}^{(1)}+\mathcal{B}_{02}^{(1)}\right)\epsilon_{-1}\right\rangle\langle\phi_0\phi_1\phi_2\rangle$$

$$= \tfrac{h}{2}\langle\mathcal{B}_{012}^{(1)}\,\epsilon_{-1}\rangle\langle\phi_0\phi_1\phi_2\rangle \tag{3.65}$$

Similarly noting that[21]

$$\frac{\langle T_{-1}T_{-2}\phi_0\phi_1\phi_2\rangle'}{\langle\phi_0\phi_1\phi_2\rangle} = \tfrac{1}{2}\left\{h(h-1)\langle\mathcal{B}_{01}^{(1)}T_{-1}\rangle\langle\mathcal{B}_{01}^{(1)}T_{-2}\rangle + h\langle\mathcal{B}_{01}^{(1)}T_{-1}\rangle\left(\langle\mathcal{B}_{-10}^{(1)}T_{-2}\rangle+\langle\mathcal{B}_{-11}^{(1)}T_{-2}\rangle\right)\right.$$

$$\left. + \tfrac{h^2}{2}\langle\mathcal{B}_{-10}^{(1)}T_{-1}\rangle\langle\left(\mathcal{B}_{12}^{(1)}+\mathcal{B}_{02}^{(1)}-\mathcal{B}_{01}^{(1)}\right)T_{-2}\rangle + \text{cyclic}_{\{0,1,2\}}\right\}$$

$$\implies$$

$$\frac{\langle\epsilon_{-1}\epsilon_{-2}\phi_0\phi_1\phi_2\rangle'}{\langle\phi_0\phi_1\phi_2\rangle} = \tfrac{1}{2}\left\{h(h-1)\langle\mathcal{B}_{01}^{(1)}\epsilon_{-1}\rangle\langle\mathcal{B}_{01}^{(1)}\epsilon_{-2}\rangle + h\langle\mathcal{B}_{01}^{(1)}\epsilon_{-1}\rangle\left(\langle\mathcal{B}_{-10}^{(1)}\epsilon_{-2}\rangle+\langle\mathcal{B}_{-11}^{(1)}\epsilon_{-2}\rangle\right)\right.$$

$$\left. + \tfrac{h^2}{2}\langle\mathcal{B}_{-10}^{(1)}\epsilon_{-1}\rangle\langle\left(\mathcal{B}_{12}^{(1)}+\mathcal{B}_{02}^{(1)}-\mathcal{B}_{01}^{(1)}\right)\epsilon_{-2}\rangle + \text{cyclic}_{\{0,1,2\}}\right\} \tag{3.66}$$

We note that $\langle\epsilon_{-1}\epsilon_{-2}\phi_0\phi_1\phi_2\rangle'$ can thus be written as

$$\frac{\langle\epsilon_{-1}\epsilon_{-2}\phi_0\phi_1\phi_2\rangle'}{\langle\phi_0\phi_1\phi_2\rangle} = \tfrac{1}{2}\left\{\frac{\langle\epsilon_{-1}\epsilon_{-2}\phi_0\phi_1\rangle'}{\langle\phi_0\phi_1\rangle}+\frac{1}{2}\frac{\langle\epsilon_{-1}\phi_0\phi_1\rangle}{\langle\phi_0\phi_1\rangle}\left(\frac{\langle\epsilon_{-2}\phi_0\phi_2\rangle}{\langle\phi_0\phi_2\rangle}+\frac{\langle\epsilon_{-2}\phi_1\phi_2\rangle}{\langle\phi_1\phi_2\rangle}-\frac{\langle\epsilon_{-2}\phi_0\phi_1\rangle}{\langle\phi_0\phi_1\rangle}\right)\right.$$

$$\left. +\text{cyclic}_{\{0,1,2\}}\right\} \tag{3.67}$$

Although cumbersome but it is important to note that we would require no new expectation values in order to compute $\langle\epsilon_{-1}\epsilon_{-2}\phi_0\phi_1\phi_2\rangle'$.

Thus the leading order ($\mathcal{O}(c^{-2})$) connected contribution to $\langle X_3X_5X_7\phi_1\phi_2Y_2Y_4Y_6\rangle$ is given by

$$\langle X_3X_5X_7|\mathbb{I}|\phi_1\phi_2|\mathbb{I}|Y_4Y_6Y_8\rangle = \langle X_3X_5X_7|T^{(1)}|\phi_1\phi_2|T^{(1)}|Y_4Y_6Y_8\rangle_{conn.}$$

$$+\langle\phi_1\phi_2|T^{(1)}|X_3X_5X_7|T^{(1)}|Y_4Y_6Y_8\rangle_{conn.}$$

$$+\langle X_3X_5X_7|T^{(1)}|Y_4Y_6Y_8|T^{(1)}|\phi_1\phi_2\rangle_{conn.} \tag{3.68}$$

and similarly the 9pt. function $\langle X_3X_5X_7\phi_0\phi_1\phi_2Y_2Y_4Y_6\rangle$ by

$$\langle X_3X_5X_7|\mathbb{I}|\phi_0\phi_1\phi_2|\mathbb{I}|Y_4Y_6Y_8\rangle = \langle X_3X_5X_7|T^{(1)}|\phi_0\phi_1\phi_2|T^{(1)}|Y_4Y_6Y_8\rangle_{conn.}$$

---

[21]Primes denote the subtraction of $\frac{\langle TT\rangle\langle\phi\phi\phi\rangle}{\langle\phi\phi\phi\rangle}$.

$$+\langle\phi_0\phi_1\phi_2|T^{(1)}|X_3X_5X_7|T^{(1)}|Y_4Y_6Y_8\rangle_{conn.}$$

$$+\langle X_3X_5X_7|T^{(1)}|Y_4Y_6Y_8|T^{(1)}|\phi_0\phi_1\phi_2\rangle_{conn.} \qquad (3.69)$$

. The *rhs* in the above expressions can be evaluated by using (3.65)and (3.67) to be

$$\frac{\langle X_3X_5X_7|T^{(1)}|\phi_1\phi_2|T^{(1)}|Y_4Y_6Y_8\rangle_{conn.}}{\langle X_3X_5X_7\rangle\langle\phi_1\phi_2\rangle\langle Y_4Y_6Y_8\rangle} = \frac{h_Xh_Y}{4}\frac{\langle\mathcal{B}^{(1)}_{357}\mathcal{B}^{(1)}_{468}\phi_1\phi_2\rangle'}{\langle\phi_1\phi_2\rangle}$$

$$\frac{\langle\phi_1\phi_2|T^{(1)}|X_3X_5X_7|T^{(1)}|Y_4Y_6Y_8\rangle_{conn.}}{\langle X_3X_5X_7\rangle\langle\phi_1\phi_2\rangle\langle Y_4Y_6Y_8\rangle} = \frac{h_\phi h_Y}{4}\left(\frac{\langle\mathcal{B}^{(1)}_{12}\mathcal{B}^{(1)}_{468}X_3X_5\rangle'}{\langle X_3X_5\rangle}+\right.$$

$$+\frac{h_X^2}{2}\langle\mathcal{B}^{(1)}_{12}\mathcal{B}^{(1)}_{35}\rangle\left(\langle\mathcal{B}^{(1)}_{468}\mathcal{B}^{(1)}_{37}\rangle + \langle\mathcal{B}^{(1)}_{468}\mathcal{B}^{(1)}_{57}\rangle - \langle\mathcal{B}^{(1)}_{468}\mathcal{B}^{(1)}_{35}\rangle\right)+$$

$$\left.+\mathrm{cyclic}_{(3,5,7)}\right)$$

$$\frac{\langle X_3X_5X_7|T^{(1)}|\phi_0\phi_1\phi_2|T^{(1)}|Y_4Y_6Y_8\rangle_{conn.}}{\langle X_3X_5X_7\rangle\langle\phi_0\phi_1\phi_2\rangle\langle Y_4Y_6Y_8\rangle} = \frac{h_Xh_Y}{8}\left(\frac{\langle\mathcal{B}^{(1)}_{357}\mathcal{B}^{(1)}_{468}\phi_0\phi_1\rangle'}{\langle\phi_0\phi_1\rangle}+\right.$$

$$+\frac{h_\phi^2}{2}\langle\mathcal{B}^{(1)}_{357}\mathcal{B}^{(1)}_{01}\rangle\left(\langle\mathcal{B}^{(1)}_{468}\mathcal{B}^{(1)}_{02}\rangle + \langle\mathcal{B}^{(1)}_{468}\mathcal{B}^{(1)}_{12}\rangle - \langle\mathcal{B}^{(1)}_{468}\mathcal{B}^{(1)}_{01}\rangle\right)+$$

$$\left.+\mathrm{cyclic}_{(3,5,7)}\right) \qquad (3.70)$$

It is important to note that although cumbersome the task of finding the above higher point functions has become entirely algebraic once the expression for $\langle\mathcal{B}^{(1)}_{ij}\mathcal{B}^{(1)}_{kl}\phi_1\phi_2\rangle$ or $\langle\epsilon_i\epsilon_k\phi_1\phi_2\rangle$ has ben determined. We summarize these functions in terms of cross ratios in the Appendix F.

### 3.3 Heavy operator insertions

We digress in this subsection to consider the case of heavy operators *i.e.* with conformal dimensions $H \sim c$ in the limit $c \to \infty$. This case has been well studied for Virasoro blocks of light primaries in 4pt functions with 2 heavy operators- HHLL in [26]. There it was argued that the effect of heavy operators can be absorbed into a conformal transformation, which in turn is determined in terms of the ratio $H/c$ and the position of the heavy operators. The argument used in [26] relies on the anomalous transformation of the stress-tensor involving the Schwarzian derivative under a conformal transformation to absorb terms involving $H/c$. We review this argument in the present context using the shadow operator formalism for HHLL.

We first begin with justifying the use of the vacuum block projector represented as in (3.41). This requires us to prove that $\mathbb{I}^2 = \mathbb{I}$ and further show that all the relevant $1/c$ contributions are indeed captured. To reiterate: the vacuum block projector represented as (3.41),

$$|\mathbb{I}| = \,\rangle\langle + \alpha\int_0 T_0\rangle\langle\tilde{T}_0 + \frac{\alpha^2}{2!}\int_{0,-1}T_{-1}T_0\rangle\langle\tilde{T}_0\tilde{T}_{-1}, + \frac{\alpha^3}{3!}\int_{0,-1,-2}T_{-2}T_{-1}T_0\rangle\langle\tilde{T}_0\tilde{T}_{-1}\tilde{T}_{-2} + \dots$$

$$= I + T^{(1)} + T^{(2)} + T^{(3)} + \cdots + T^{(n)} + \ldots . \tag{3.71}$$

with $\alpha = \frac{3}{\pi c}$ and where any two $T_i$s or $\tilde{T}_i$s originating form the same term are not supposed to be fused with each other. In other words the contribution coming from such contractions in any expectation value is to be subtracted. The above sum is basically a sum over projectors with increasing powers in $1/c$. We first show that the *rhs* above squares to itself. For this it suffices to show that

$$T^{(n)} T^{(m)} = \delta_{n,m} T^{(n)} \tag{3.72}$$

To show this we assume that $\langle T \rangle = 0 = \langle \tilde{T} \rangle$ *i.e.* we are in a conformal frame where the *vev* of the stress-tensor vanishes[22]. The above condition is easily seen to be satisfied for $I$ and $T^{(1)}$. We first need to show that sub-leading terms (third onwards) in (3.71) square to themselves like any projector. This can be easily seem to hold for the the second term in (3.71) *i.e.* as was shown in [1]

$$\int_0 \tilde{T}_0 \rangle \langle T_0 = \int_{0,-1} \tilde{T}_0 \rangle \langle T_0 \tilde{T}_{-1} \rangle \langle T_{-1} \tag{3.73}$$

using $\langle \tilde{T}_0 T_{-1} \rangle = \alpha^{-1} \delta^2 (z_0 - z_{-1}) \equiv \alpha^{-1} \delta^2_{0,-1}$. The square of the third term in (3.41) *i.e.* $T^{(2)}$ yields

$$\left( \frac{1}{2!} \right)^2 \int_{0,-1,1,2} T_0 T_{-1} \rangle \langle \tilde{T}_{-1} \tilde{T}_0 T_2 T_1 \rangle' \langle \tilde{T}_1 \tilde{T}_2 \tag{3.74}$$

where the prime as usual denotes that relevant $T_i$s (and $\tilde{T}_i$s) are not to be contracted. Note that the expectation value of $\langle \tilde{T}_{-1} \tilde{T}_0 T_2 T_1 \rangle$ depends on that of $\langle T_{-1} T_0 T_2 T_1 \rangle$, therefore it is enough to understand the structure of $\langle T_{-1} T_0 T_2 T_1 \rangle'$. For this we note that $\langle T_{-1} T_0 T_2 T_1 \rangle$ is given by

$$\langle T_{-1} T_0 T_2 T_1 \rangle = \sum_{i=0}^{2} \left( \frac{2}{z_{-1i}^2} + \frac{\partial_i}{z_{-1i}} \right) \sum_{j=0}^{2} \left( \frac{2}{z_{0j}^2} + \frac{\partial_i}{z_{0j}} \right) \langle T_1 T_2 \rangle$$

$$+ \langle T_{-1} T_0 \rangle \langle T_1 T_2 \rangle + \langle T_{-1} T_1 \rangle \langle T_0 T_2 \rangle + \langle T_{-1} T_2 \rangle \langle T_1 T_0 \rangle$$

$$\therefore \langle T_{-1} T_0 T_2 T_1 \rangle' = \langle T_{-1} T_1 \rangle \langle T_0 T_2 \rangle + \langle T_{-1} T_2 \rangle \langle T_1 T_0 \rangle \tag{3.75}$$

where $\langle T_{-1} T_0 T_2 T_1 \rangle'$ doesn't receive contributions from those terms in $\langle T_{-1} T_0 T_2 T_1 \rangle$ which are either proportional to $\langle T_{-1} T_0 \rangle$ or $\langle T_2 T_1 \rangle$. Therefore we can write

$$\langle \tilde{T}_{-1} \tilde{T}_0 T_2 T_1 \rangle' = \langle \tilde{T}_{-1} T_1 \rangle \langle \tilde{T}_0 T_2 \rangle + \langle \tilde{T}_{-1} T_2 \rangle \langle \tilde{T}_0 T_1 \rangle$$
$$= \delta^2 (z_{-1} - z_1) \delta^2 (z_0 - z_2) + \delta^2 (z_{-1} - z_2) \delta^2 (z_0 - z_1) \tag{3.76}$$

Using the above result in (3.74) we easily see that

$$\left( \frac{1}{2!} \right)^2 \int_{0,-1,1,2} T_0 T_{-1} \rangle \langle \tilde{T}_{-1} \tilde{T}_0 T_2 T_1 \rangle' \langle \tilde{T}_1 \tilde{T}_2 = \frac{1}{2!} \int_{0,-1} T_{-1} T_0 \rangle \langle \tilde{T}_0 \tilde{T}_{-1} \tag{3.77}$$

---

[22]Therefore the use of $\mathbb{I}$ is only justified in a frame where $\langle T \rangle = 0$.

It is not hard to see how the above argument generalizes to higher terms in (3.41) with the $1/n!$ normalization accounting for the allowed contractions of the $T$s when the projectors are squared. One can similarly show that $T^{(2)}T^{(1)} = 0$

$$T^{(1)}T^{(2)} \sim \int_{0,-1,-2} \tilde{T}_0 \rangle \langle T_0 T_{-1} T_{-2} \rangle' \langle \tilde{T}_{-2} \tilde{T}_{-1}$$

$$\text{now } \langle T_0 T_{-1} T_{-2} \rangle = \sum_{i=1}^{2} \left( \frac{2}{z_{0i}^2} + \frac{\partial_i}{z_{0i}} \right) \langle T_{-1} T_{-2} \rangle$$

$$\implies \langle T_0 T_{-1} T_{-2} \rangle' = \sum_{i=1}^{2} \left( \frac{2}{z_{0i}^2} + \frac{\partial_i}{z_{0i}} \right) \langle T_{-1} T_{-2} \rangle' = 0 \tag{3.78}$$

One can similarly show (3.72) iteratively. It suffices to consider $\langle T_{-1} \dots T_{-n} T_1 \dots T_m \rangle'$ where only $\langle T_{-j} T_i \rangle \neq 0 \, \forall j, i \in \mathbb{Z}_{>0}$. This must evaluate to

$$\langle T_{-1} \dots T_{-n} T_1 \dots T_m \rangle' = \delta_{n,m} \left( (\langle T_{-1} T_1 \rangle \dots \langle T_{-n} T_n \rangle) + \dots \right) \tag{3.79}$$

where there are $n!$ terms on the *rhs* consisting of all possible pairings. We first begin by noting that $\langle T_1 \dots T_m \rangle' = 0$ where none of the $T_i$s are allowed to contract with each other[23]. Next consider $\langle T_{-1} T_1 \dots T_m \rangle'$ where only $\langle T_{-1} T_i \rangle \neq 0 \, \forall i \in \{1.n\}$. This we find is

$$\langle T_{-1} T_1 \dots T_m \rangle' = \hat{T}_{-1} \langle T_1 \dots T_m \rangle' + \sum_{i=1}^{m} \langle T_{-1} T_i \rangle \langle T_1 \dots T_{i-1} T_{i+1} \dots T_m \rangle'$$

$$\text{where } \hat{T}_{-1} = \sum_{i=1}^{m} \left( \frac{2}{z_{1i}^2} + \frac{\partial_i}{z_{1i}} \right). \tag{3.80}$$

Here the first term vanishes and the second term vanishes conditionally unless $m = 1$, in which case the *lhs* reduces to $\langle T_{-1} T_1 \rangle$. For $m > 1$ we next consider $\langle T_{-2} T_{-1} T_1 \dots T_m \rangle'$ where as before only $\langle T_{-j} T_i \rangle \neq 0 \, \forall i, j \in \mathbb{Z}_{>0}$. This we find is

$$\langle T_{-2} T_{-1} T_1 \dots T_m \rangle' = \hat{T}_{-2} \langle T_1 \dots T_m \rangle' + \sum_{i=1}^{m} \langle T_{-1} T_i \rangle \langle T_1 \dots T_{i-1} T_{i+1} \dots T_m \rangle' \tag{3.81}$$

where again we find the first term vanishes while the second term vanishes unless $m - 1 = 1$, in which case we have

$$\langle T_{-2} T_{-1} T_1 T_2 \rangle' = \langle T_{-1} T_1 \rangle \langle T_{-2} T_2 \rangle + \langle T_{-1} T_2 \rangle \langle T_{-2} T_1 \rangle. \tag{3.82}$$

Thus proceeding in a similar manner one can show iteratively (3.79) and thus (3.72). Each of the $T^{(n)}$s in $\mathbb{I}$ can therefore be regarded as projectors orthogonal to each other and projecting onto states which contribute exactly at order $\mathcal{O}(c^{-n})$.

---

[23]This can be easily shown iteratively beginning with $\langle T_1 T_2 \rangle' = 0 \implies \langle T_1 T_2 T_3 \rangle' = 0$ and so on.

Proving each of the $T^{(n)}$s square to themselves is not enough, we would further have to show that they do not project onto a smaller set of states which contribute at $\mathcal{O}(c^{-n})$. Unfortunately one would have to compare contributions of each of the $T^{(n)}$s to the 4pt vacuum block with the expected answer. We can show that the expected answers match to leading orders in operator dimensions in the 4pt vacuum block. For example the the contribution of $T^{(2)}$ at this order is

$$\int_{0,-1} \langle\phi_1\phi_2 T_{-1}T_0\rangle\langle\tilde{T}_0\tilde{T}_{-1}\psi_3\psi_4\rangle|_{h_\phi^2 h_\psi^2} = h_\phi^2 h_\psi^2 \langle B_{12}^{(1)} B_{34}^{(1)}\rangle^2 = \langle\phi_1\phi_2|T|\psi_3\psi_4\rangle_{c^{-2},h_\phi^2 h_\psi^2} \quad (3.83)$$

where the suffixes mean *terms proportional to* and the *rhs* is computed using Virasoro operator projectors

$$\langle\phi_1\phi_2|T|\psi_3\psi_4\rangle_{c^{-2},h_\phi^2 h_\psi^2} = \sum_{m,n\geq 2} \langle\phi_1\phi_2 L_{-m}L_{-n}\rangle\mathcal{N}_{m,n}^{-1}\langle L_m L_n \psi_3\psi_4\rangle|_{h_\phi^2 h_\psi^2} \quad (3.84)$$

with the matrix inverse $\mathcal{N}_{m,n}^{-1}$ ($\mathcal{N}_{m,n} = \langle L_m L_n L_{-n}L_{-n}\rangle$) is evaluated to order $\mathcal{O}(c^{-2})$ [26]. This argument is expected to hold for later terms in (3.71). In fact one can similarly show that

$$\langle\phi_1\phi_2|T^{(n)}|\psi_3\psi_4\rangle|_{h_\phi^n h_\psi^n} = h_\phi^n h_\psi^n \langle B_{12}^{(1)} B_{34}^{(1)}\rangle^n \langle\phi_1\phi_2\rangle\langle\psi_3\psi_4\rangle = \langle\phi_1\phi_2|T|\psi_3\psi_4\rangle_{c^{-n},h_\phi^n h_\psi^n} \quad (3.85)$$

where the *rhs* can be easily evaluated as

$$\langle\phi_1\phi_2|T|\psi_3\psi_4\rangle_{c^{-n},h_\phi^n h_\psi^n} = \sum_{\{m\}=2}^{\infty} \langle\phi_1\phi_2 L_{-m_1}\dots L_{-m_n}\rangle\mathcal{N}_{\{m\}}^{-1}\langle L_{m_n}\dots L_{m_1}\psi_3\psi_4\rangle|_{c^{-n},h_\phi^n h_\psi^n}$$

using

$$\langle\phi_1(\infty)\phi_2(z)L_{-m_1}\dots L_{-m_n}\rangle|_{h_\phi^n} = \left(h_\phi^n \prod_{i=1}^n (m_i - 1)z^{-m_i} + \mathcal{O}(h_\phi^{m_i-1})\right)\langle\phi_1(\infty)\phi_2(z)\rangle$$

$$\langle L_{m_n}\dots L_{m_1}\psi_3(1)\psi_4(0)\rangle|_{h_\psi^n} = \left(h_\psi^n \prod_{i=1}^n (m_i - 1) + \mathcal{O}(h_\phi^{m_i-1})\right)\langle\psi_3(1)\psi_4(0)\rangle$$

$$\mathcal{N}_{\{m\}}^{-1}|_{c^{-n}} = c^{-n}\prod_{i=1}^n \frac{1}{m_i(m_i^2 - 1)} + \mathcal{O}(c^{-n-1}). \quad (3.86)$$

The *lhs* of (3.85) can also be easily evaluated noting that

$$\langle\phi_1\phi_2 T_1\dots T_n\rangle'|_{h_\phi^n} = h_\phi^n \prod_{i=1}^n \langle\mathcal{B}_{12}^{(1)} T_i\rangle\langle\phi_1\phi_2\rangle + \mathcal{O}(h_\phi^{n-1}) \quad (3.87)$$

and a similar expression with $T_i \leftrightarrow \tilde{T}_i$. The computation of the *lhs* of (3.85) then proceeds similar to that of $\langle\phi_1\phi_2|T^{(1)}|\psi_3\psi_4\rangle$. Therefore each of the $T^{(n)}$s can accurately capture the contribution to the 4pt vacuum block at leading order in the operator dimensions.

In what follows we will *assume* that $T^{(n)}$s as defined using (3.71) accurately capture the

contribution of states in $\langle\phi_1\phi_2\psi_3\psi_4\rangle_{vac.}$ proportional to $h_\phi^n/c^n$. We will see that this assumption correctly reproduces the HHLL vacuum block at leading order for $h_\phi \sim c$ [26] with the additional benefit of the proof being algebraic.

Let the conformal dimensions of $\Phi$ and $\psi$ be $H$ and $h$ respectively with $H \sim \mathcal{O}(c)$ s.t. $24H/c < 1$ while $h \sim \mathcal{O}(c^0)$ and finite[24]. Consider

$$\langle\Phi_1\Phi_2|\mathbb{I}|\psi_3\psi_4\rangle_w = \langle\Phi_1\Phi_2\rangle_w\langle\psi_3\psi_4\rangle_w + \alpha\int_0\langle\Phi_1\Phi_2 T_0\rangle_w\langle\tilde{T}_0\psi_3\psi_4\rangle_w$$
$$+\frac{\alpha^2}{2!}\int_{0,-1}\langle\Phi_1\Phi_2 T_{-1}T_0\rangle_w\langle\tilde{T}_0\tilde{T}_{-1}\psi_3\psi_4\rangle_w + \dots \quad (3.88)$$

in the limit $c \to \infty$, where $|\mathbb{I}|$ stands for the identity block projector and the subs-script $w$ indicates the frame in which this is being computed. Note we assume that in the $w$-frame the *vev* of stress-tensor vanishes, thus allowing us to use the projector $\mathbb{I}$ as in (3.71). Let $z(w)$ be the frame in which we would want to compute the HHLL vacuum block . Consider the $\langle\phi_1\phi_2|T^{(1)}|\psi_3\psi_4\rangle$ term above, this can be written as

$$\int_0\langle\Phi_1\Phi_2 T_0\rangle_w\langle\tilde{T}_0\phi_3\phi_4\rangle_w = J_{z/w}^\Phi\int_0\left(\frac{\partial z_0}{\partial w_0}\right)^2\left[\langle\Phi_1\Phi_2 T_0\rangle_z - \tfrac{c}{12}\{w_0,z_0\}\langle\Phi_1\Phi_2\rangle_z\right]\langle\tilde{T}_0\phi_3\phi_4\rangle_w \quad (3.89)$$

where $J_{z/w}^\Phi = \left(\frac{\partial z_i\partial z_j}{\partial w_i\partial w_j}\right)^H$ is the Jacobian related to the $\langle\Phi_i\Phi_j\rangle$. We next choose $1 - z = \left(w_1\frac{w_2-w}{w_1-w}\right)^{1/\alpha}$ with $\alpha = \sqrt{1 - \frac{24H}{c}}$ which is determined by demanding

$$\left[\langle\Phi_1\Phi_2 T_0\rangle_z - \tfrac{c}{12}\{w_0,z_0\}\langle\Phi_1\Phi_2\rangle_z\right] = 0 = \left[H\langle\mathcal{B}_{12}^{(1)}T_0\rangle - \tfrac{c}{12}\{w_0,z_0\}\right]\langle\Phi_1\Phi_2\rangle_z \quad (3.90)$$

Here $w_{1,2}$ are the positions of $\Phi$ in the $w$-frame. Therefore in the $w$-frame the corrections due to $|T^{(1)}|$ insertion contributes 0 in (3.88) in the limit $c \to \infty$. To analyse the higher-order terms in (3.88) we note

$$\langle\Phi_1\Phi_2 T_{-1}T_0\rangle_w = J_{z/w}^{\Phi,T}\left[\langle\Phi_1\Phi_2 T_{-1}T_0\rangle_z - \frac{c}{12}\{w_0,z_0\}\langle\Phi_1\Phi_2 T_{-1}\rangle_z - \frac{c}{12}\{w_{-1},z_{-1}\}\langle\Phi_1\Phi_2 T_0\rangle_z\right.$$
$$\left.+\frac{c^2}{12^2}\{w_0,z_0\}\{w_{-1},z_{-1}\}\langle\Phi_1\Phi_2\rangle_z\right]. \quad (3.91)$$

Making use of (3.11) to write $\langle\Phi_1\Phi_2 T_{-1}T_0\rangle_z$ in terms of the bilinear $\mathcal{B}_{12}^{(1)}$ and using (3.90) we find that terms proportional to $H^2$ in the above *rhs* vanish. Thus

$$\int_{0,-1}\langle\Phi_1\Phi_2 T_{-1}T_0\rangle_w\langle\tilde{T}_0\tilde{T}_{-1}\phi_3\phi_4\rangle_w \sim \mathcal{O}(H/c^2). \quad (3.92)$$

---

[24]$H < c/24$ implies a conical defect geometry in the bulk $AdS_3$ without a horizon while $H > c/24$ implies one with horizon [26] and thus mimics a thermal state at the leading order *c.f* [27]. The analysis for the later case is identical.

The analysis of higher order terms in (3.88) follows in the same vein. We note some basic properties of multiple stress-tensor insertions in $\langle \Phi_1 \Phi_2 \rangle$

$$\langle \Phi_1 \Phi_2 T_{-1} \ldots T_{-n} \rangle = H^n \prod_{i=1}^{n} \langle \mathcal{B}_{12}^{(1)} T_i \rangle + \mathcal{O}(H^{m-1})$$

$$\langle \Phi_1 \Phi_2 T_{-1} \ldots T_{-n} \rangle_w = J_{z/w}^{\Phi,T} \langle \Phi_1 \Phi_2 (T_{-1} - \tfrac{c}{12}\{w_{-1}, z_{-1}\}) \ldots (T_{-n} - \tfrac{c}{12}\{w_{-n}, z_{-n}\}) \rangle_z \quad (3.93)$$

One can then use (3.90) to show that $\langle \Phi_1 \Phi_2 T_{-1} \ldots T_{-n} \rangle_w \sim \mathcal{O}(H^{n-1}/c^n)$[25]. Therefore the higher order terms in (3.88) all give vanishing contribution in the $w$-frame in the limit $c \to \infty$. Thus

$$\langle \Phi_1 \Phi_2 | \mathbb{I} | \psi_3 \psi_4 \rangle_w = \langle \Phi_1 \Phi_2 \rangle_w \langle \psi_3 \psi_4 \rangle_w + \mathcal{O}(1/c) \tag{3.94}$$

Reading off the answer in the $z$-frame therefore yields the result of [26][26]

$$\langle \Phi_1 \Phi_2 | \mathbb{I} | \psi_3 \psi_4 \rangle_z = J_{w/z}^{\Phi} J_{w/z}^{\psi} \langle \Phi_1 \Phi_2 \rangle_w \langle \psi_3 \psi_4 \rangle_w. \tag{3.95}$$

The above set of arguments along with the analysis of the previous subsections can be readily generalized to the case of HHLLLL...L. It is worth mentioning at this point that the generalization of the HHLL analysis of [26] to HHLLL...L case was done using the monodromy method in [48] wherein the strict limit of $c \to \infty$ was taken. The above method of using $\mathbb{I}$ as in (3.71) may lend a more diagrammatic and intuitive analysis of the HHLLLL...L case.

The argument of [26] has been generalized to the case of vacuum blocks in HHLLLL...L with arbitrary but finite number light insertions and arbitrary corrections in $1/c$ as $c \to \infty$ in [44] using the monodromy method. The authors of [44] assume exponentiation of the vacuum conformal block even in the presence of 2 heavy operators and deduce that even sub-leading corrections in the $1/c$ expansion can be obtained by implementing such a conformal transformation. These sub-leading corrections in the presence of a pair of heavy operators for the 6pt case are necessary to compare with the connected star channel diagram computed in [36] in the limit the heavy operators become light. However as noted in the conclusion of [36] this does not match the 6pt function answer at $\mathcal{O}(c^{-2})$.

# 4  OTOC for $\langle \mathcal{BB}\phi\phi \rangle_{c^{-2}}$

We next turn to finding the OTOC for the $\mathcal{O}(c^{-2})$ terms in (3.27). We do not symmetrize *wrt* the pairs of operators but the final conclusions of this section (4.11),(4.8),(4.12),(4.13) hold even after symmetrization except for some numerical pre-factors in the *lhs* of (4.11) and (4.8) which are qualitatively unimportant. In order to measure the Lyapunov index in a

---

[25]We explicitly check the contribution from $T^{(3)}$ insertion in (3.88) for which even $\mathcal{O}(H^2/c^3)$ contributions vanish.

[26]The generalization to light primary block is straightforward where the same conformal transformation $z(w)$ is derived demanding cancellation of terms proportional to $H$ in $\left[ \langle \Phi_1 \Phi_2 T_0 \mathcal{O} \rangle_z - \tfrac{c}{12}\{w_0, z_0\} \langle \Phi_1 \Phi_2 \mathcal{O} \rangle_z \right]$.

thermal background one needs to first map the line coordinate to a circle *via* the exponential map

$$z_i = e^{\frac{2\pi}{\beta}(t_i + \sigma_i - i\tau_i)} \tag{4.1}$$

where we have also retained the Euclidean time $\tau$. We set the Lotrentzian times $t_i=0$ and choose a specific Euclidean time ordering for the various points in (3.27) as

$$\tau_4 < \tau_2 < \tau_6 < \tau_5 < \tau_1 < \tau_3 \tag{4.2}$$

for the out of time ordered case, and

$$\tau_4 < \tau_6 < \tau_2 < \tau_1 < \tau_5 < \tau_3 \tag{4.3}$$

for the time ordered case. We will set the spatial points to

$$\sigma_4 = \sigma_6 = \sigma_Y \quad > \quad \sigma_1 = \sigma_2 = \sigma_\phi \quad > \quad \sigma_3 = \sigma_5 = \sigma_X. \tag{4.4}$$

Having evaluated the Euclidean answer we next turn on the Loretnzian times with the following ordering

$$t_4 = t_6 = t_Y \quad < \quad t_1 = t_2 = t_\phi \quad < \quad t_3 = t_5 = t_X \tag{4.5}$$

Here we choose to work with the following invariant cross ratios

$$z = \frac{z_{12}z_{34}}{z_{13}z_{24}}, \qquad y = \frac{z_{12}z_{56}}{z_{15}z_{26}}, \qquad u = \frac{z_{12}z_{54}}{z_{15}z_{24}}. \tag{4.6}$$

the Regge limit for whom would be $z \to 0, y \to 0, u \to 0$.[27] The final expression for the various terms in (3.27) would consist of logarithms and di-logarithms($\text{Li}_2$) of functions of the above cross ratios. These (di-)logarithms have branch cuts in their arguments[28]. As the cross ratios approach the Regge limit they trace a contour in their complex plane and depending on the time ordering and their functional dependence inside the logarithm and di-logarithms they may or may not cross these branch cuts. The contours traced by the cross ratios for TO correlators is such that the they do not receive any contribution from these branch cuts. For OTO correlators however the contours traced do cross certain branch cuts and it is these contributions which give the exponential behaviour.

## 4.1 OTO of $\langle\mathcal{BB}\rangle\langle\mathcal{BB}\rangle$

We fist look that OTO behaviour of the second term in (3.27) as this seems like a square of the 4pt stress-tensor block. In the Regge limit of (4.6) the relevant cross ratios involved in this term also tend to zero *i.e.*

$$\frac{z_{12}z_{35}}{z_{13}z_{25}} = \frac{u-z}{u-1} \to 0, \qquad \frac{z_{12}z_{46}}{z_{14}z_{26}} = \frac{u-y}{u-1} \to 0 \tag{4.7}$$

---

[27]It might be useful to switch to $Z = \frac{z_{31}z_{25}}{z_{32}z_{15}} = \frac{1-u}{1-z}$, $U = \frac{z_{31}z_{24}}{z_{32}z_{14}} = \frac{1}{1-z}$, $V = \frac{z_{31}z_{26}}{z_{32}z_{16}} = \frac{1-u}{(1-y)(1-z)}$ which tend to 1 in the Regge limit. The behaviour of these cross ratios for the out of time ordering of (4.2) is plotted in [36].

[28]$\log x$ has a branch cut from $x \in (-\infty, 0]$ with a discontinuity of $2\pi i$, while $\text{Li}_2 x$ has a branch cut from $x \in [1, \infty)$ and it picks up a value of $2\pi i \log x$

The branch cuts of Logarithms involved are crossed by the above cross ratios only for out of time ordering (4.2) and not for time ordered arrangement of (4.3). The exponentially growing pieces in the second term in (3.27) are as

$$
h(h-1)\left\{ \frac{\beta^4}{4c^2\pi^2\tau_{46}\tau_{12}^2\tau_{35}}\sinh^2\left(\frac{\pi(t_{\phi Y}-\sigma_{Y\phi})}{\beta}\right)\sinh^2\left(\frac{\pi(t_{X\phi}-\sigma_{\phi X})}{\beta}\right) \right.
$$
$$
\left. +\frac{i\beta^2}{2\pi c^2\tau_{12}\tau_{35}}\sinh^2\left(\frac{\pi(t_{X\phi}-\sigma_{\phi X})}{\beta}\right)+\frac{i\beta^2}{2\pi c^2\tau_{46}\tau_{12}}\sinh^2\left(\frac{\pi(t_{\phi Y}-\sigma_{Y\phi})}{\beta}\right)\dots\right\}
$$
$$(4.8)$$

where the only the first term is relevant for the growth in the largest time interval $t_{XY}$. Above we have only retained the leading terms in $\tau_{ij}$[29] in $1/c^2$ as the Euclidean time differences $\tau_{ij} \to 0$. We clearly see the behaviour obtained in [36] where the Lyapunov index for large $t_{XY} \gg \frac{\beta}{2\pi}$ is

$$
\lambda_L = \frac{2\pi}{\beta} \tag{4.9}
$$

with the exponential growth lasting for $2t^*$ where $t^* = \frac{\beta}{2\pi}\log(c)$ is the scrambling time. The first 2 terms above showcase the characteristic growth of the 4pt OTOC for respective large intermediate times $t_{X\phi}$ & $t_{\phi Y}$ with the same $\lambda_L$ but lasting for $t^*$. It is worth noting that to deduce the exponential behaviour in all the time intervals above we did not have to solve any non-trivial differential equations. However for operator dimension $h \ll c$, the linear in $h$ terms in the above expression do combine with the exponential growth coming from OTO behaviour of the last term in (3.27) which we next turn to.

### 4.2 OTO of $\langle \mathcal{K}^{(2)} \rangle$

This term is linear in $h$ and has terms which are proportional to $\mathrm{Li}_2, \log$ & $\log^2$ of various cross ratios. The leading contribution in the Regge limit for OTO placement of operators (4.2) is of the form

$$
\langle\mathcal{K}^{(2)}\rangle_{\mathrm{oto}} \approx \frac{11\beta^4}{4\tau_{46}\tau_{12}^2\tau_{35}}\sinh^2\left(\frac{\pi(t_{\phi Y}-\sigma_{Y\phi})}{\beta}\right)\sinh^2\left(\frac{\pi(t_{X\phi}-\sigma_{\phi X})}{\beta}\right)+\mathcal{O}(\tau_{ij}^{-3}) \tag{4.10}
$$

which is the leading order behaviour for $t_{XY} \gg \frac{\beta}{2\pi}$[30]. Moreover the exponential behaviours for time interval $t_{XY}$ of terms linear in operator dimension $h$ persist upon adding the relevant terms from (4.8).

$$
\frac{\left\langle\mathcal{B}_{35}^{(1)}\mathcal{B}_{46}^{(1)}\phi_1\phi_2\right\rangle}{\langle\phi_1\phi_2\rangle} \approx \frac{1085\beta^4 h}{c^2\pi^2\tau_{46}\tau_{12}^2\tau_{35}}\sinh^2\left(\frac{\pi(t_{\phi Y}-\sigma_{Y\phi})}{\beta}\right)\sinh^2\left(\frac{\pi(t_{X\phi}-\sigma_{\phi X})}{\beta}\right)+\mathcal{O}(\tau_{ij}^{-3}) \tag{4.11}
$$

---

[29]$\tau_{ij} = \tau_i - \tau_j$

[30]Here we have suppressed terms of order $\mathcal{O}(\tau_{12}^{-3})$

Here we have retained the only the most singular terms as $\tau_{ij} \to 0$[31]. We thus see a growth in the largest time interval $t_{XY}$ governed by a Lyapunov index

$$\lambda_L = \frac{2\pi}{\beta} \tag{4.12}$$

with a growth lasting for $2t^* = \frac{\beta}{\pi} \log c$. This bodes well with the expectation that $2n$-pt function without of time ordering such that one requires $n-1$ turns in the complex time plane to faithfully describe it, grows exponentially for [19]

$$(n-1)t^* \tag{4.13}$$

However this was proved to hold [19] for a 1d theory of reparametrizations governed by the Schwarzian action. This exact behaviour was also deduced in the case of 6pt star channel vacuum block in CFT$_2$ in [36].

## 4.3 OTO for $2n$-pt vacuum block

Having studied the 6-pt case, in this sub-section we would like to gain some insights into the pairwise equal $2n$-pt out of time ordered correlator. We note a specific peculiarity of the 6-pt OTOC's behaviour in (4.11): arranging the (pairs of) correlators in increasing values of their Lorentzian times (4.5)

$$t_X > t_\phi > t_Y, \tag{4.14}$$

we find that although the operator pairs $X_{3,5}$ & $Y_{4,6}$ are not out of time ordered *wrt* each other there is an exponential growth in the time interval $t_{XY}$. In other words moving the "time-stamp" $t_X$ relative to that of $t_Y$ results in an exponential change in the OTOC in terms of $t_{XY}$. It is precisely this growth that lasts twice the scrambling time and serves as a finer probe of the chaotic behaviour.

For $2k$-pt correlation functions- with $k > 2$, there are many possible out of time ordered configurations [52, 53]. The out of time ordering in (4.2) considered here is a 6-pt generalization of what is termed as a "maximally braided" out of time ordering for pair-wise equal $2n$-pt correlators. These were studied in [19] in the context of the CFT$_1$ dual to JT-gravity in Euclidean $AdS_2$ as fine gained probes of chaos. The effective theory of reparametrizations in CFT$_1$ dual to Eucledian JT-gravity on $AdS_2$ is governed by the Schwarzian action [31–33]. Given that the $i^{\text{th}}$ pair has time stamps $\{t_i, \tau_i\}$ & $\{t_{i'}, \tau_{i'}\}$, the maximally braided OTOC can be constructed as follows: assuming that the Lorentzian times to be increasing- $t_i = t_{i'} > t_{i+1} = t_{(i+1)'}$; the corresponding Euclidean times then follow a braiding pattern

$$\tau_i > \tau_{(i-1)'} > \tau_{i+1} > \tau_{i'} \qquad \forall\, i \neq 1, n$$
$$\implies \tau_1 > \tau_2 > \tau_{1'} > \tau_3 > \tau_{2'} > \tau_4 > \tau_{3'} > \tau_5 > \tau_{4'} > \dots \tau_n > \tau_{(n-1)'} > \tau_{n'}. \tag{4.15}$$

---

[31]The sub-leading terms $1/\tau_{ij}$ in the limit $\tau_{ij} \to 0$ also exhibit the exact same behaviour.

Put simply this procedure implies out of time ordering only for operator pairs which are nearest neighbours in Lorentzian times. Therefore the 1st and the $n^{\text{th}}$ (or $i^{\text{th}}$ and the $(i + k)^{\text{th}}$, $\forall k \geq 2$) operator pairs are time ordered. For such a theory in a thermal state with temperature $2\pi/\beta$ it was shown[19] that the maximally braided OTOC for $2n$-pts grows exponentially in the largest Lorentzian time interval $t_{1n} = t_1 - t_n$, with the Lyapunov index $\lambda_L = (n-1)2\pi/\beta$. Also this exponential growth lasts for

$$\Delta t = (n-1)t^*, \qquad t^* = \frac{\beta}{2\pi} \log c \qquad (4.16)$$

where $t^*$ is the scrambling time and $c \sim G_N^{-(d-1)}$ is the central charge[32].

In the case of $CFT_2$ as discussed in subsection 3.2.1 we can see that the leading order contribution at large $c$ of the connected vacuum block of n-pairs of light operators contains the diagrams discussed in (3.60) i.e.

$$(4.17)$$

where the *rhs* consists of all possible distinct diagrams with $(n-1)$ legs connecting the $n$ pairs. To prescribe a maximally braided OTO configuration a permutation of the $n$ pairs has to be chosen so that their Lorentzian times are arranged in ascending order. Given that only the nearest neighbours in such a configuration would be out of time ordered this choice uniquely prefers one of the diagrams on the *rhs i.e.* where the same permutation is connected such that the diagram forms a line. For the OTO configuration (4.15) this is simply

$$(4.18)$$

If one were to then ask the growth in the OTOC for this OTO configuration we expect only the above diagram to exhibit the behaviour as found in [19]. Nonetheless, it would be interesting to compute the contribution of the various diagrams on the *rhs* of (4.17) and verify this claim. Multi-point $CFT_2$ correlators have been used to discern interesting physics of information retrieval from behind the horizon in wormhole geometries [54]. We leave this exercise for the near future.

---

[32]Here we assume that the Schwarzian theory is describing the near extremal dynamics of a black hole in $d+1$ dimensional space time. In general $c$ can be taken to be counting the microscopic *dof* of the system.

# 5  Bulk perspective

In this section we note certain similarities with the bulk computation demonstrating an exponential growth in OTOC as shown in [18]. We concern ourselves with a static black hole in $AdS_3$. This computation relies on the twin sided eternal Schwarzchild black hole in $AdS_3$ to compute the late time correlation between 2 pairs of operators. The pairs of operators are arranged so as to give an OTOC when suitably analytically continued for time separations much larger than the inverse temperature of the black hole $i.e.$: $t \gg \frac{\beta}{2\pi}$. This would imply that the dual large-$N$ strongly coupled system is being probed with an OTOC with the same operators for time scales much larger than the dispersion time which is of the scale of $\frac{\beta}{2\pi}$.

The particular details of this computation have since been generalized to rotating geometries [35]. The essential idea being that at times $t \gg \frac{\beta}{2\pi}$ the leading order contribution to the probe approximation can be computed from an Eikonal approximation. Here one considers a scattering of shock-waves produced in the bulk by the dual scalar fields interacting by exchanging gravitons governed by the minimal coupling of the scalars in the bulk. In the Kruskal coordinates the Schwarzchild $BTZ$ is

$$\frac{ds^2}{\ell^2} = \frac{-4dudv}{(1+uv)^2} + r_+^2 \frac{(1-uv)^2}{(1+uv)^2} dx^2 \tag{5.1}$$

In response to a in falling shock-wave with momentum $p^v$ produced by a probe scalar sourced at $\phi$ on the boundary at late times

$$T_{uu} = \frac{1}{2r_+} p^v \delta(u)\delta(x-x') \tag{5.2}$$

the above metric (5.1) produces a response [14, 55–58]

$$\frac{ds^2}{\ell^2} = \frac{-4dudv}{(1+uv)^2} + r_+^2 \frac{(1-uv)^2}{(1+uv)^2} dx^2 + h_{uu} du^2$$

$$h_{uu} = 32\pi G_N r_+ p^v \delta(u) g(x) \tag{5.3}$$

This is constrained by the differential equation arising from the terms linear in $G_N$ from the Einstein's equation with stress-tensor (5.2) as $G_N \to 0$.

$$\partial^2 g(x) - r_+^2 g(x) = -\delta(x) \tag{5.4}$$

The eikonal phase shift due to scattering an ingoing shock wave with momentum $p^v$ with an outgoing one with momentum $p^u$ is then given by computing the change in the linearised on-shell action

$$\delta S_{\text{on shell}} = \frac{1}{2} \int d^3x \sqrt{-g} h_{uu} T^{uu}$$

$$= 4\pi G_N r_+ p^v p^u g(\delta x) \tag{5.5}$$

where $\delta x$ is the difference in the location at the boundary for the sources of the 2 shock waves i.e. the location of the pair of operators which source bulk fields.

The CFT understanding of the growth of OTOC thus deduced is that it is governed by the stress-tensor block in the 4pt function. The contribution of other (heavy) operator blocks is ignored by appealing to sparseness of spectrum for holographic systems. Conformal blocks in a 2d CFT are constrained by relevant Casimir equations

$$D\mathcal{F}_{\Delta,l}(u,v) = \lambda_{\Delta,l}\mathcal{F}_{\Delta,l}(u,v) \tag{5.6}$$
$$D = \left(z^2(1-z)\partial_z^2 - z^2\partial_z\right) + \left(\bar{z}^2(1-\bar{z})\partial_{\bar{z}}^2 - \bar{z}^2\partial_{\bar{z}}\right)$$
$$\lambda_{\Delta,l} = \tfrac{1}{2}\Delta(\Delta-2) + \tfrac{l^2}{2}, \qquad \Delta = \tfrac{h+\bar{h}}{2}, l = \tfrac{h-\bar{h}}{2}$$

while perturbations of dual fields in the bulk are likewise constrained by their bulk e.o.m.. For the case at hand the bulk field dual to the CFT stress-tensor is the metric whose response to the shock-wave i.e. late time perturbation due to a scalar propagation is governed by linearised Einstein's eq. (5.4).

Assuming a late time behaviour of the stress-tensor conformal block of the form

$$\mathcal{F}(t,x) \approx \frac{e^{\frac{2\pi}{\beta}t}}{c}g(x) \tag{5.7}$$

one can expand (5.6) for late times, knowing that for late Lorentzian times where

$$t_{1,2} = t > 0, \; x_{1,2} = 0, \; t_{3,4} = 0, \; x_{3,4} = x > 0, \qquad \tau_1 > \tau_3 > \tau_2 > \tau_4 \tag{5.8}$$

the out of time ordered cross ratios behave like

$$z \approx -e^{\frac{2\pi}{\beta}(x-t)}\epsilon_{12}^*\epsilon_{34} \quad \bar{z} \approx -e^{-\frac{2\pi}{\beta}(x+t)}\epsilon_{12}^*\epsilon_{34} \tag{5.9}$$

with $\epsilon_{ij} = i\left(e^{\frac{2\pi}{\beta}\tau_i} - e^{\frac{2\pi}{\beta}\tau_j}\right)$. This late time expansion ($t \gg \frac{\beta}{2\pi}$) of the Casimir equation yields

$$\partial^2 g(x) - r_+^2 g(x) = 0 \tag{5.10}$$

which is precisely the linearised Einstein's eq. (5.4) which we were required to solve for a shock-wave but without the source delta function on the r.h.s. Note that it was crucial that we assumed a growing behaviour of the form (5.7) in $t$ for the stress-tensor which can only be assumed to hold for out of time ordering of the 4pt correlators.

Given the fact that the vacuum conformal block at $\mathcal{O}(1/c)$ is given by $\langle \mathcal{B}_{12}^{(1)}\mathcal{B}_{34}^{(1)}\rangle$ i.e. two-point functions of bi-locals constructed out of the reparametrization modes $\epsilon_i$s, there seem to be a plausible relation between them and the backreactions in the bulk of the form (5.4). It is worth asking what these reparametrization modes mean in terms of bulk fields and can they similarly furnish a effective description of chaotic degrees of freedom as they do in the CFT.

# 6 Discussion and Conclusions

In this article, we initiated a study of Ward identities for the reparametrization modes in a 2d CFT, in the context of a generic $n$-point correlator. In particular, we focussed on the four and six-point functions of pairwise identical operators and find that one can compute the vacuum block contributions to completely connected channel without having the need to presume a non-linear form of the bi-local operator(3.29),(3.31) as in [36]. The study of reparametrization mode Ward identities naturally lead us to investigate how the projector onto the identity Virasoro block can be represented in terms of conformal integrals using shadow operator formalism allowing us to

- Postulate and test a representation of the Identity block projector (3.41) in terms of conformal integrals using the shadow of the stress-tensor. The order $1/c^2$ term in this projector *i.e.* $T^{(2)}$ is required in order to correctly obtain the connected part of the 6pt. vacuum block *i.e.* the "star" channel.

- This projector representation allows us to expand the vacuum block contribution to pair wise equal 6pt function of light operators diagrammatically in powers of $1/c$. This allows us to discern what was called the "star" channel vacuum block of similar 6pt function computed in [36].

- We are able to generalize the "star" channel vacuum block to 8pt and higher point functions of pair-wise equal light operators as the leading (in $1/c$) connected diagram and give a diagrammatic expression for them. We also observe that symmetry between the operator pairs is observed in every order in its $1/c$ expansion by using the representation of the vacuum block projector (3.41).

- Having evaluated $\langle\epsilon\epsilon\phi\phi\rangle$ and using the identity block projector to order $1/c^2$ we are able to evaluate "star" channel vacuum blocks of simpler 8pt. and 9pt. generalizations to the 6pt. function algebraically.

- Using the vacuum block projector representation (3.41) we are able to give an algebraic proof of how the leading order contribution to the vacuum block for HHLL is obtained by a conformal transformation. We believe this method is easily generalizable to the case of HHLL...LL too which otherwise would require the monodromy method.

The $1/c^2$ term in the vacuum block projector $\mathbb{I}$ (3.41) is deduced by demanding that the connected part of the 6pt. function be symmetric *w.r.t.* the operator pairs. The form of the terms $T^{(n>2)}$ is then deduced from the fact that at every order in $1/c$ expansion the vacuum block remains symmetric between the pair of operators. We also see that this correctly reproduces the vacuum block at leading order in operator dimensions in all orders in $1/c$. We also see that we can correctly reproduce the leading order answer when a pair of operators are heavy *i.e.* have dimension $H \sim c$. The answer for the HHLL..LL case follows simply from

that of the HHLL case as the proof is simple and algebraic, the former earlier required the use of the monodromy method.

While this approach provides us with an alternative method of computing higher point correlators, in the stress-tensor dominated channel, it leaves open avenues for further physical aspects. As was emphasised in [36] that the physics of multiple linear graviton exchanges is more important than that of graviton self interaction [59–61], it would be interesting to understand how repeated use of the reparametrization Ward identity of the form (3.19) could help understand aspects of higher point star channel diagrams. Such a program would however require knowing the results more conformal integrals than those are currently available in literature.

The observations of section 5 indicate a plausible relationship between the reparametrization modes and the bulk backreactions to matter fields *via* Einstein's equation. From a holographic perspective it is nonetheless important to understand the bulk analogue for the reparametrization modes as these may similarly capture effective degrees of freedom which encapsulate chaotic behaviour. Such effective descriptions already exist in terms of the well studied Jackiw-Teitelboim (JT) model [32, 33] used to understand the near horizon dynamics of near extremal black holes. This model captures thermal chaotic behaviour in terms of an effective 1d theory of time reparametrizations in terms of its Schwarzian derivatives at the near horizon throat boundary. However, the phenomenon of extremal chaos as deduced by the results of [34, 35] is not captured by this model [62]. Investigations into the holographic dual description of reparametrization modes and their Ward identities would perhaps yield a more complete picture as they necessarily must reduce to the 1d reparametrization modes of the JT model in the case of near horizon dynamics of near extremal black holes.

At a technical level, it is interesting to understand in detail how this approach may work when the pairwise identical operators are relaxed to a more general configuration of operators, including spin. Spinning operators are particularly important in the understanding the physics of Kerr black holes, see *e.g.* [35, 62, 63] for discussions related to the chaos-bound in this case. An involved and physically interesting description is likely to exist in higher dimensions, for generic operators in the dual CFT [64, 65]. From a Holographic perspective, this is tied to the near horizon physics of rotating black holes, in which a complete understanding is lacking at present [62]. We hope to address some of these questions in near future.

It would also be interesting to understand how this approach can be used in the study of the stress-tensor block for 2 heavy operators ($H \sim c$) inserted along with many light operators $L \ll c$ along the lines of the argument presented in sub-section 3.3. It was shown for HHLL [26] that this is obtained from a conformal transformation of the LLLL with the transformation parameter governed by $\sqrt{1 - 24H/c}$. A similar but stronger statement was proved using the monodromy method to some extent for the case of HHLLLLL.... case in [44] where in it was argued that even sub-leading corrections in $1/c$ can be obtained by

employing a similar change of conformal frame. Although this does not match the expected answer for the 6pt vacuum block at $\mathcal{O}(c^{-2})$ in the limit the heavy operators tend to being light. A more clear understanding in this regard would shed further light on the Eigenstate Thermalization Hypothesis in $\mathrm{CFT}_2$. It would also be nice to explore a perturbative and diagramatic understanding of heavy operators contributing to the conformal blocks using 2d Ward identities along the lines of the discussion in subsection 3.3.

On a more conceptual note, recent advances in understanding the properties of thermal correlators, including that of OTOCs, makes it clear that the IR-physics encoded in these correlators implicitly know about the UV-completion, specially for systems with a Holographic dual. This statement simply follows from e.g. the chaos bound, which is inherently related to unitarity in the high energy states, that nonetheless provides a bound for an IR-quantity, *i.e.* the Lyapunov exponent. A more general understanding of this aspect is still missing, and it is a very interesting question to what extent the reparametrization modes, together with the shadow operator formalism, Ward identities and such, can shed light on such aspects.

On a related note, it is curious that the dynamics of maximal chaos, in Holography, does not necessarily require an Einstein-Hilbert dynamics. Instead, similar physics can be obtained from a Nambu-Goto dynamics[66, 67] or a Dirac-Born-Infeld dynamics[68]. In the former case, with strings propagating in an $\mathrm{AdS}_3$-background, there is a precise relation between the dual CFT and the world-sheet CFT[69]. It would be very interesting to uncover the details of how the reparametrization modes of these two CFTs are related to each other. We hope to come back to some of these issues in near future.

## Acknowledgements

The authors would like to thank Bobby Ezhuthachan for useful comments on the draft. AKP is supported by the Council of Scientific & Industrial Research (CSIR) Fellowship No. 09/489(0108)/2017-EMR-I. RP is supported by the Lise Meitner fellowship M 2882-N funded by the FWF.

## A Appendix A: Embedding space

The embedding space of $d + 2$ dimensions allows the realization of the conformal group as $SO(d + 1, 1)$ rotation. The space-time coordinates are obtained by projecting onto the null sphere in the embedding space and identifying scaling *w.r.t.* the affine parameter on the null sphere. The null sphere given by

$$X^a \cdot X_a = X^+ X^- + X^\mu X_\mu = 0 \tag{A.1}$$

can be used to set

$$X^- = -X^+ x^2, \quad X^\mu = X^+ x^\mu \text{ for } X^+ \neq 0$$
$$X^a = X^+ \{1, -x^2, x^\mu\} \tag{A.2}$$

Using this parametrization of the null sphere we see that $X^+$ is the affine parameter. Further we identify $X^+ \equiv \lambda X^+$, $\forall \lambda \in \mathbb{R}$. The projector onto the null surface and normal vectors are obtained by

$$e^a_\mu(X) = \frac{\partial X^a}{\partial x^\mu} = X^+\{0, -2x^\mu, \delta^\nu_\mu\},$$

$$k^a = \frac{\partial X^a}{\partial X^+} = \{1, -x^2, x^\mu\}$$

$$N^a = 2\delta^a_-, \tag{A.3}$$

where $N^a$ is obtained by demanding $k \cdot N = 1$ & $e_\mu \cdot N = 0$. The space-time fields are obtained from the embedding space fields by restricting them to the null sphere. Space-time primary scalar $\phi(x)$ is given by

$$\Phi(X) \equiv X^+\phi(x) \tag{A.4}$$

where $\Phi(X)$ is restricted on the null sphere. Similarly for space-time tensor primaries we have

$$V_{\mu_1\dots\mu_l}(x) \equiv e^{a_1}_{\mu_1}(X)\dots e^{a_l}_{\mu_l}(X)\, V_{a_1\dots a_l}(X). \tag{A.5}$$

where $\equiv$ is understood as having to identify (gauge fix) $X^+$ components (to 1). One can then define the inversion tensor in embedding space as

$$I_{ab}(X^a_1, X^b_2) = \eta_{ab} - \frac{X^1_b X^2_a}{X_{12}}, \quad \text{where } X_{12} = X_1 \cdot X_2 = -\tfrac{1}{2}X^+_1 X^+_2 x^2_{12},$$

$$\text{as} \quad e^a_\mu(X_1)I_{ab}\, e^b_\nu(X_2) = I_{\mu\nu} = \eta_{\mu\nu} - 2\frac{x^{12}_\mu x^{12}_\nu}{x^2_{12}}. \tag{A.6}$$

Here we have used $e_\mu(X_1) \cdot X_2 = -X^+_1 X^+_2 x^{12}_\mu$. The embedding space metric can therefore be decomposed along the infinitesimal curves (A.3) as

$$\eta_{ab} = e^a_\mu e^b_\nu \eta^{\mu\nu} + 2k^{(a}N^{b)}. \tag{A.7}$$

Tensor primaries in embedding space would also satisfy

$$V^a(X) \cdot X_a = 0 \implies V^a(X) \equiv V^a(X) + X^a s(X) \tag{A.8}$$

on the null surface $X^2 = 0$. One can then show using (A.8) and (A.7) that

$$e^a_\mu(X)I_{ab}(X, Y)V^b(Y) \equiv I_{\mu\nu}(x, y)V^\nu(y) \tag{A.9}$$

This would be useful in defining shadows of tensor primaries in terms of their space-time components.

# B  Appendix B: Conformal Integrals

We note certain useful results for conformal integrals in $d$ and $d = 2$ dimensions here [29, 30]. We indicate the $d$ dimensional conformally invariant volume in the $d + 2$ dimensional embedding space as $D^d X$ here but revert to using $d^d X$ or $d^d x$ in the main text while treating them as conformally invariant in $d$ dimensions.

$$I(Y) = \int D^d X \frac{1}{(-2X.Y)^d} = \frac{\pi^{d/2}\Gamma(d/2)}{\Gamma(d)} \frac{1}{(Y^2)^{d/2}}, \quad (\forall \, Y^2 < 0) \tag{B.1}$$

$$\int D^d X_0 \frac{1}{X_{10}^a X_{02}^b X_{03}^c} = \frac{\pi^{\frac{d}{2}}\Gamma(\frac{d}{2} - a)\Gamma(\frac{d}{2} - b)\Gamma(\frac{d}{2} - c)}{\Gamma(a)\Gamma(b)\Gamma(c)} \frac{1}{X_{12}^{\frac{d}{2}-c} X_{13}^{\frac{d}{2}-b} X_{23}^{\frac{d}{2}-a}} \tag{B.2}$$

where $a + b + c = d$ and $X_{ij} = -2X_i.X_j = (x_i - x_j)^2$ when $X_i^2 = X_j^2 = 0$.

$$\int \frac{D^d X_0}{X_{10}^{d-\Delta} X_{20}^{\Delta}} = \frac{\pi^{\frac{d}{2}}\Gamma(\Delta - \frac{d}{2})}{\Gamma(\Delta)} \frac{(X_2^2)^{\frac{d}{2}-\Delta}}{X_{12}^{d-\Delta}} \tag{B.3}$$

Using this one can show

$$\int \frac{D^d X_0 D^d X_1}{X_{10}^{d-\Delta} X_{20}^{\Delta} X_{13}^{\Delta}} = \frac{\pi^{\frac{d}{2}}\Gamma(\Delta - \frac{d}{2})\Gamma(\frac{d}{2} - \Delta)}{\Gamma(\Delta)\Gamma(d - \Delta)} \frac{1}{X_{23}^{\Delta}} \tag{B.4}$$

In the 2d case we use the integrals of he form

$$I_n = \frac{1}{\pi} \int d^2 x_0 \, f_n(z_0) \bar{f}_n(\bar{z}_0), \quad f_n(z_0) = \prod_{i=1}^{n} (z_0 - z_i)^{-h_i}, \quad \bar{f}_n(\bar{z}_0) = \prod_{i=1}^{n} (\bar{z}_0 - \bar{z}_i)^{-\bar{h}_i}$$

$$\text{where} \quad \sum_{i=1}^{n} h_i = \sum_{i=1}^{n} \bar{h}_i = d = 2, \quad h_i - \bar{h}_i \in \mathbb{Z}. \tag{B.5}$$

These integrals were solved for $n = 2, 3, 4$ cases in [30](Appendix A). We note the $n = 4$ case for our use below

$$I_4 = z_{12}^{h_3+h_4-1} z_{23}^{h-1+h_4-1} z_{31}^{h_2-1} z_{24}^{-h_4} \bar{z}_{12}^{\bar{h}_3+\bar{h}_4-1} \bar{z}_{23}^{\bar{h}-1+\bar{h}_4-1} \bar{z}_{31}^{\bar{h}_2-1} \bar{z}_{24}^{-\bar{h}_4} \mathcal{I}_4(z, \bar{z}),$$

$$\mathcal{I}_4 = K_{42} F_1(1 - h_2, h_4; h_3 + h_4, z)_2 F_1(1 - \bar{h}_2, \bar{h}_4; \bar{h}_3 + \bar{h}_4, \bar{z})$$
$$+ \bar{K}_4 (-1)^{h_1+h_4-\bar{h}_1-\bar{h}_4} z^{h_1+h_2-1} \bar{z}^{\bar{h}_1+\bar{h}_2-1} {}_2F_1(1 - h_3, h_1; h_1 + h_2, z)$$
$$\times {}_2F_1(1 - \bar{h}_3, \bar{h}_1; \bar{h}_1 + \bar{h}_2, \bar{z}),$$

$$K_4 = \frac{\Gamma(1 - h_1)\Gamma(1 - h_2)\Gamma(h_1 + h_2 - 1)}{\Gamma(\bar{h}_1)\Gamma(\bar{h}_2)\Gamma(2 - \bar{h}_1 - \bar{h}_2)}, \quad \bar{K}_4 = \frac{\Gamma(1 - h_3)\Gamma(1 - h_4)\Gamma(h_3 + h_4 - 1)}{\Gamma(\bar{h}_3)\Gamma(\bar{h}_4)\Gamma(2 - \bar{h}_3 - \bar{h}_4)}$$
$$\tag{B.6}$$

where $z = \frac{z_{12}z_{34}}{z_{13}z_{24}}, \bar{z} = \frac{\bar{z}_{12}\bar{z}_{34}}{\bar{z}_{13}\bar{z}_{24}}$

## C   Appendix C: $\epsilon$-mode Ward Identity *via* integration

In this Appendix we evaluate the $\langle \epsilon_3 \epsilon_4 \phi_1 \phi_2 \rangle$ using integrating the Ward identity namely

$$\langle T_{-1} T_0 \phi_1 \phi_2 \rangle = \frac{c/2}{z_{-10}^4} \langle \phi_1 \phi_2 \rangle + \left( \frac{(h\text{-}1)z_{12}^2}{z_{-11}^2 z_{-12}^2} + \frac{z_{01}^2}{z_{-10}^2 z_{-11}^2} + \frac{z_{02}^2}{z_{-10}^2 z_{-12}^2} \right) \frac{h z_{12}^2}{z_{01}^2 z_{02}^2} \frac{1}{(z_{12}\bar{z}_{12})^{2h}}. \tag{C.1}$$

and then expressing the result as a total derivative *wrt* $\bar{z}_{3,4}$. Consider the integral of the first term inside the brackets above[33]:

$$\int_{-1,0} \frac{z_{-13}^2 z_{40}^2}{\bar{z}_{-13}^2 \bar{z}_{40}^2} \frac{z_{12}^2}{z_{-11}^2 z_{-12}^2} \frac{z_{12}^2}{z_{01}^2 z_{02}^2} \frac{1}{(z_{12}\bar{z}_{12})^{2h}} = \int_{-1,0} \frac{z_{-13}^4 z_{40}^4 z_{12}^4 \bar{z}_{-12}^2 \bar{z}_{01}^2 \bar{z}_{02}^2}{X_{12}^{2h} X_{-13}^2 X_{40}^2 X_{01}^2 X_{02}^2 X_{-11}^2 X_{-12}^2}$$

$$= \frac{1}{X_{12}^{2h-2}} \int_0 \frac{I_{40}^{a\bar{a}} I_{40}^{b\bar{b}} I_{\bar{a}\bar{a}}^{01} I_{\bar{b}\bar{b}}^{02} I_{12}^{\tilde{a}\tilde{b}}}{X_{40}^0 X_{01} X_{02}} \int_{-1} \frac{I_{a\bar{a}}^{3\text{-}1} I_{b\bar{b}}^{3\text{-}1} I_{\bar{a}\bar{a}}^{-11} I_{\bar{b}\bar{b}}^{-12} I_{12}^{\tilde{a}\tilde{b}}}{X_{-13}^0 X_{-11} X_{-12}} \Bigg|_{zzzz} \tag{C.2}$$

where we evaluate the all $z$ component of the last expression. Note that each integral is similar to the integral used for obtaining $\langle \tilde{B}_3^{ab} \phi_1 \phi_2 \rangle$ from $\langle B_{ab}^0 \phi_1 \phi_2 \rangle$ with the dimension of $\tilde{B}_{ab} = \delta \to 0$.

$$\frac{1}{X_{12}^{2h}} \int_{-10} \frac{z_{-13}^2 z_{40}^2}{\bar{z}_{-13}^2 \bar{z}_{40}^2} \frac{z_{12}^2}{z_{-11}^2 z_{-12}^2} \frac{z_{12}^2}{z_{01}^2 z_{02}^2} = \frac{4}{X_{12}^{2h}} \frac{z_{13} z_{14} z_{23} z_{24} \bar{z}_{12}^2}{\bar{z}_{13} \bar{z}_{14} \bar{z}_{23} \bar{z}_{24} z_{12}^2}$$

$$= \frac{4}{X_{12}^{2h}} \left( I_{41}^{a\bar{a}} I_{\bar{a}\bar{b}}^{12} I_{24}^{\bar{b}b} \right) \left( I_{31}^{c\bar{c}} I_{\bar{c}\bar{d}}^{12} I_{23}^{\bar{d}d} \right) \Bigg|_{zzzz}$$

$$= \left( \frac{\pi c}{6} \right)^2 \frac{4 \bar{\partial}_4 \bar{\partial}_3}{X_{12}^{2h}} \langle \epsilon_3 \mathcal{B}_{12}^{(1)} \rangle \langle \epsilon_4 \mathcal{B}_{12}^{(1)} \rangle \tag{C.3}$$

In going from the second line to the third line we note the equivalence between the *rhs*s of (3.3) and (3.4). The integral for the second term in side the brackets in (C.1) can be written as

$$\int_{-1,0} \frac{z_{40}^2 z_{-13}^2}{\bar{z}_{40}^2 \bar{z}_{-13}^2} \frac{z_{12}^2}{z_{-10}^2 z_{-11}^2 z_{02}^2 X_{12}^{2h}} = \int_{-1,0} \frac{z_{40}^4 z_{3\text{-}1}^4 z_{12}^2 \bar{z}_{-10}^2 \bar{z}_{-11}^2 \bar{z}_{02}^2}{X_{40}^2 X_{-13}^2 X_{-10}^2 X_{-11}^2 X_{02}^2 X_{12}^{2h}}$$

$$= \int_{-1,0} \frac{I_{40}^{a\bar{a}} I_{40}^{b\bar{b}} I_{be}^{02} I_{21}^{ef} I_{\bar{c}f}^{-11} I_{\bar{d}\bar{a}}^{-10} I_{3\text{-}1}^{d\bar{d}} I_{-13}^{c\bar{c}}}{X_{40}^0 X_{-13}^0 X_{-10} X_{-11} X_{02} X_{12}^{2h-1}} \Bigg|_{zzzz}. \tag{C.4}$$

The third term inside the brackets in (C.1) is obtained by exchanging $X_1 \leftrightarrow X_2$. The above integral unlike (C.2) does not take a familiar form. However such integrals where explicitly known in 2d *c.f.* appendix A of [30]. For the case at hand we note the relevant integral in Appendix B here.

$$\int_{-1,0} \frac{z_{40}^2 z_{-13}^2}{\bar{z}_{40}^2 \bar{z}_{-13}^2} \frac{z_{12}^2}{z_{-10}^2 z_{-11}^2 z_{02}^2 X_{12}^{2h}} = \frac{\bar{\partial}_4 \bar{\partial}_3}{X_{12}^{2h}} \left\{ 6 \frac{z_{13}^2 z_{24}^2}{z_{12}^2} \left( \log(1-z) + \mathrm{Li}_2(\bar{z}) \right) - z_{34}^2 \log(\bar{z}_{34}) \right.$$

---

[33]We consider the all $z$ components of the resulting tensor.

$$+\frac{z_{13}z_{24}z_{34}}{z_{12}}\left(6\log(\bar{z}_{34})-4\log(1-z)\log(\bar{z}_{34})-4\mathrm{Li}_2(\bar{z})\right)\Big\} \quad \text{(C.5)}$$

We can write the *l.h.s* above as the all $z$ component of

$$\frac{\partial_4^a\partial_3^c}{X_{12}^{2h}}\left\{6\frac{X_{13}X_{24}}{X_{12}}I_{31}^{b\bar{b}}I_{\bar{b}\bar{d}}^{12}I_{24}^{\bar{d}d}\left(\log(1-z)+\mathrm{Li}_2(\bar{z})\right)-I_{34}^{bd}X_{34}\log(X_{34})\right.$$
$$\left.+\frac{X_{34}}{\bar{z}}I_{31}^{b\bar{b}}I_{\bar{b}\bar{d}}^{12}I_{24}^{\bar{d}d}\left(6\log(X_{34})-4\log(1-z)\log(X_{34})-4\mathrm{Li}_2(\bar{z})\right)\right\} \quad \text{(C.6)}$$

where $z=\frac{z_{12}z_{34}}{z_{13}z_{24}}$ and it's complex conjugate are related to the conformally invariant cross ratios as $u=z\bar{z}$, $v=(1-z)(1-\bar{z})$. Therefore we can now write out $\langle\epsilon\epsilon\phi\phi\rangle$ as

$$\frac{\langle\epsilon_3\epsilon_4\phi_1\phi_2\rangle}{\langle\phi_1\phi_2\rangle}=\langle\epsilon_3\epsilon_4\rangle+h(h-1)\langle\epsilon_3\mathcal{B}_{12}^{(1)}\rangle\langle\epsilon_4\mathcal{B}_{12}^{(1)}\rangle+$$
$$+h\left(\tfrac{12}{c}\right)^2\left\{6\frac{X_{13}X_{24}}{X_{12}}I_{31}^{b\bar{b}}I_{\bar{b}\bar{d}}^{12}I_{24}^{\bar{d}d}\left(\log(1-z)+\mathrm{Li}_2(\bar{z})\right)-I_{34}^{bd}X_{34}\log(X_{34})\right.$$
$$+\frac{X_{34}}{\bar{z}}I_{31}^{b\bar{b}}I_{\bar{b}\bar{d}}^{12}I_{24}^{\bar{d}d}\left(6\log(X_{34})-4\log(1-z)\log(X_{34})-4\mathrm{Li}_2(\bar{z})\right)$$
$$\left.+\left(1\leftrightarrow 2\right)\right\}_{zz} \quad \text{(C.7)}$$

Restricting to only the physical block and simplifying the result in terms of the cross ratio we find

$$\frac{\langle\epsilon_3\epsilon_4\phi_1\phi_2\rangle_{\mathrm{phys}}}{\langle\phi_1\phi_2\rangle}=\langle\epsilon_3\epsilon_4\rangle_{\mathrm{phys}}+h(h-1)\langle\epsilon_3\mathcal{B}_{12}^{(1)}\rangle_{\mathrm{phys}}\langle\epsilon_4\mathcal{B}_{12}^{(1)}\rangle_{\mathrm{phys}}+h\left(\tfrac{12}{c}\right)^2\mathcal{C}_{\mathrm{phys}}^{(2)}$$
$$\text{(C.8)}$$

where

$$\mathcal{C}_{\mathrm{phys}}^{(2)}=\langle\epsilon_3\epsilon_4\rangle_{\mathrm{phys}}\left[4+\left(-2+\frac{4}{z}\right)\log(1-z)\right]+z_{34}^2\mathcal{F}(z). \quad \text{(C.9)}$$

The extra term $\mathcal{F}(z)$ is undetermined and constraints of the form (2.11) can be further used to determine it.

# D   Appendix D: Leading vacuum block 6pt function

We show here that for the computing the leading stress-tensor Virasoro block contribution to 6pt function of pair-wise equal operators in the semi-classical limit ($c\to\infty$, $h\sim h_{X,Y}\sim\mathcal{O}(c^0)$) we need only consider an integral of the form

$$\frac{\langle X_3X_5\,|T|_g\,\phi_1\phi_2\,|T|_g\,Y_4Y_6\rangle}{\langle\phi_1\phi_2\rangle\langle X_3X_5\rangle\langle Y_4Y_6\rangle}=\left(\frac{3}{\pi c}\right)^2\int_{3',4'}\frac{\langle\tilde{T}_{3'}\tilde{T}_{4'}\phi_1\phi_2\rangle\langle T_{3'}X_3X_5\rangle\langle T_{4'}Y_4Y_6\rangle}{\langle\phi_1\phi_2\rangle\langle X_3X_5\rangle\langle Y_4Y_6\rangle}, \quad \text{(D.1)}$$

where the pairs $X_{3,5}$ and $Y_{4,6}$ fuse to exchange global states associated with the stress-tensor. Here $|T|_g \sim \int_0 T_0\rangle\langle\tilde{T}_0$ indicates projector onto global states of the stress-tensor. To see this we write the numerator of the above *lhs* as

$$\sum_{n,m=0}^{\infty} \frac{\langle X_3 X_5 L_{-1}^n L_{-2}\rangle\langle L_2 L_1^n \phi_1 \phi_2 L_{-1}^m L_{-2}\rangle\langle L_2 L_1^m Y_4 Y_6\rangle}{\langle L_2 L_1^n L_{-1}^n L_{-2}\rangle\langle L_2 L_1^m L_{-1}^m L_{-2}\rangle} \tag{D.2}$$

where we have used $T\rangle \sim |L_{-2}\rangle$ and $\langle T \sim \langle L_2|$. Although this form is computationally cumbersome it allows for an easy power counting of the central charge $c$. Each factor in the denominator grows[34] as $c$

$$\langle L_2 L_1^n L_{-1}^n L_{-2}\rangle \to \# + \#c \tag{D.3}$$

The only contribution of $c$ in the numerator comes from the middle factor, but this is clearly the disconnected piece of the Ward Identity as

$$\langle L_2 L_1^n \phi_1 \phi_2 L_{-1}^m L_{-2}\rangle \sim \mathcal{D}_{n,m}\langle T\phi_1\phi_2 T\rangle \tag{D.4}$$

where $\mathcal{D}_{n,m}$ is a differential operator generated due to the commutation relations of $[L_{\{1,-1\}}, \phi]$, $[L_1, L_{-2}]$ and $[L_{-1}, L_2]$, hence does not contribute factors of $c$. therefore

$$\langle T\phi_1\phi_2 T\rangle\Big|_{\mathcal{O}(c)} = \langle TT\rangle\langle\phi_1\phi_2\rangle \tag{D.5}$$

Therefore the disconnected part of (D.2) goes as $1/c$ while the connected part scales as $1/c^2$ as $c \to \infty$. Now lets include a single state $|L_{-p}\rangle$ where $p \geq 2$ into the projector separating the pair $X_{3,4}$ from $\phi_{1,2}$

$$\sum_{p=2}^{\infty}\sum_{n,m=0}^{\infty} \frac{\langle X_3 X_5 L_{-p} L_{-1}^n L_{-2}\rangle\langle L_2 L_1^n L_p \phi_1 \phi_2 L_{-1}^m L_{-2}\rangle\langle L_2 L_1^m Y_4 Y_6\rangle}{\langle L_2 L_1^n L_{-1}^n L_{-p} L_p L_{-2}\rangle\langle L_2 L_1^m L_{-1}^m L_{-2}\rangle}. \tag{D.6}$$

The first factor in the denominator behaves in powers of $c$ as

$$\langle L_2 L_1^n L_{-1}^n L_{-p} L_p L_{-2}\rangle \to \#(1 + \#c + \#c^2) \tag{D.7}$$

As before only the second factor in the numerator contributes powers of $c$ which are still linear at most linear in $c$. One can again show that the linear terms in $c$ arising from the second factor in the numerator $\langle L_2 L_1^n L_p \phi_1 \phi_2 L_{-1}^m L_{-2}\rangle$ contribute to the disconnected diagram, except when $p = 2$ which we deal with separately. Therefore for states generated by $p \neq 2$ the Virasoro sates formed by fusing the pair $X_{3,5}$ give sub-leading contribution of $\mathcal{O}(c^{-3})$ for the connected part.

For the case of $p = 2$ we need to understand what kind of diagrams are associated when

---

[34]We make use of $[L_m, L_n] = (m-n)L_{m+n} + \frac{c}{12}m(m-1)(m+1)\delta_{m,-n}$ to see which commutators yield a power of $c$.

the second factor of the numerator in (D.6) scales linearly with $c$. In stead of (D.6) we can consider

$$\sum_{n,m,k=0}^{\infty} \frac{\langle X_3 X_5 L_{-1}^k L_{-p} L_{-1}^n L_{-2}\rangle \langle L_2 L_1^n L_p L_1^k \phi_1 \phi_2 L_{-1}^m L_{-2}\rangle \langle L_2 L_1^m Y_4 Y_6\rangle}{\langle L_2 L_1^n L_{-1}^n L_{-p} L_{-1}^k L_1^k L_p L_{-2}\rangle \langle L_2 L_1^m L_{-1}^m L_{-2}\rangle}\Bigg|_{p=2}. \qquad (D.8)$$

where we have introduced global states generated by the action of $L_{-1}^k$ over those generated by $L_{-p} = L_{-2}$. This can be written as in terms of conformal integrals as

$$\int_{3',4',5'} \frac{\langle X_3 X_5 T_{5'} T_{3'}\rangle \langle \tilde{T}_{3'} \tilde{T}_{5'} \phi_1 \phi_2 \tilde{T}_{4'}\rangle \langle T_{4'} Y_4 Y_6\rangle}{\langle \phi_1 \phi_2\rangle \langle X_3 X_5\rangle \langle Y_4 Y_6\rangle}. \qquad (D.9)$$

Note that the denominators in (D.2),(D.6),(D.8) are just terms without any space-time dependence. Therefore we can understand how the different diagrams involved in (D.8) contribute at varying powers in $c$ by using stress tensor Ward identity on (D.9). The above integral has terms(diagrams) upto $\mathcal{O}(c^{-3})$ and we are interested in connected diagrams at $\mathcal{O}(c^{-2})$. It is obvious that the decomposition $\langle X_3 X_5\rangle \langle T_{5'} T_{3'}\rangle \subset \langle X_3 X_5 T_{5'} T_{3'}\rangle$ contributes a disconnected diagram while providing an extra power of $c$ in the numerator. The same is true of the decomposition $\langle T_3 T_5\rangle \langle \phi_1 \phi_2 T_4\rangle \subset \langle T_3 T_5 \phi_1 \phi_2 T_4\rangle$. The only connected diagram that contributes at $\mathcal{O}(c^{-2})$ emanates from the decomposition $\langle T_3 T_4\rangle \langle T_5 \phi_1 \phi_2\rangle \subset \langle T_3 T_5 \phi_1 \phi_2 T_4\rangle$. But this diagram is simply obtained from the integral in (D.1) by exchanging the pairs $\phi_{1,2} \leftrightarrow X_{3,5}$.

Therefore in the limit $c \to \infty$ it suffices to consider the integral of the form (D.1) to get the leading order contribution to the connected Virasoro block for pair-wise equal 6pt function provided one symmetrizes $wrt$ the three pairs.

# E    Appendix E: 8pt function leading order connected vacuum block

Here we determine the contributions to the connected part of the vacuum block of pairwise equal 8pt function. As discussed in subsection 3.2.1 these can be obtained from analysing the leading components in $1/c$ in

$$\langle X_3 X_5 \phi_1 \phi_2 Z_7 Z_9 Y_4 Y_6\rangle_{vac.} = \sum_{n,m,l=0}^{\infty} \langle X_3 X_5 | T^{(n)} | \phi_1 \phi_2 | T^{(m)} | Z_7 Z_9 | T^{(l)} | Y_4 Y_6\rangle_{vac.} \qquad (E.1)$$

where the vacuum block projector (3.71) is used $i.e.$ $\mathbb{I} = I + T^{(1)} + T^{(2)} + \cdots + T^{(n)} + \ldots$. As evident above there are 3 positions where the projector can be inserted with each $T^{(i)}$ accompanied with a factor of $\alpha^i/i! \sim 1/c^i i!$. Terms involving $|T^{(i>2)}|$ would contribute to the leading connected diagram at order $\mathcal{O}(c^{-3})$ as contractions between the stress-tensors belonging to different places of insertions of $|\mathbb{I}|$ are allowed and would lower the powers of $c$ in the denominator. To understand this let's consider stress-tensor contractions on a generic

term in this sum which we diagrammatically denote as

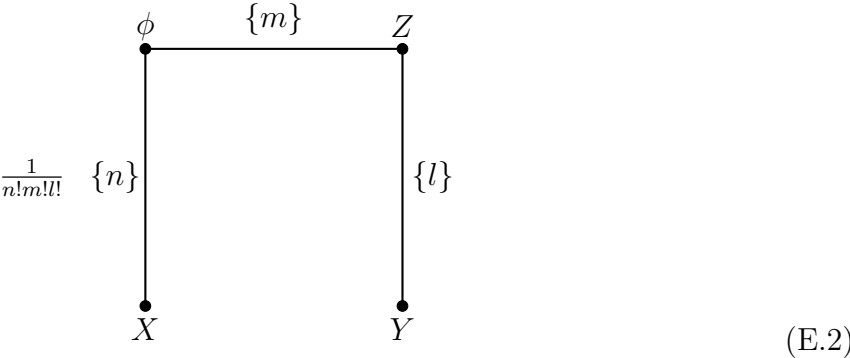

$$(\text{E.2})$$

with $\{n\}$ denoting a bunch of $n$- legs; we also denote the normalization coming from each set of legs. Also as per the rules of contraction we can only contract the (stress-tensor) legs fusing into $\phi_{1,2}$ and $Z_{7,9}$ from 2 different points (in this case from points $X_{3,5}$, and $Y_{4,6}$.). Let's take one such contraction- fusing $a$ legs between $X_{3,5} - \phi_{1,2}$ with $a$ legs between $\phi_{1,2} - Z_{7,9}$. There are $\frac{n!m!}{(n-a)!(m-a)!}$ ways of doing this, thus giving

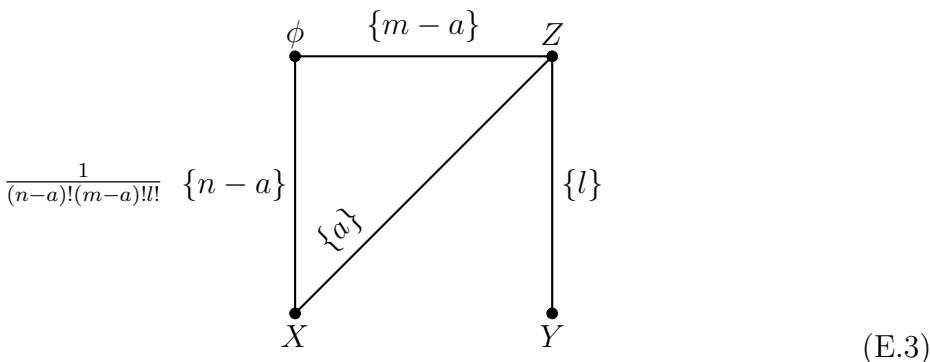

$$(\text{E.3})$$

Similarly contracting $f$ legs between $X_{3,5} - Z_{7,9}$ with those between $Z_{7,9} - Y_{4,6}$[35]. We thus get

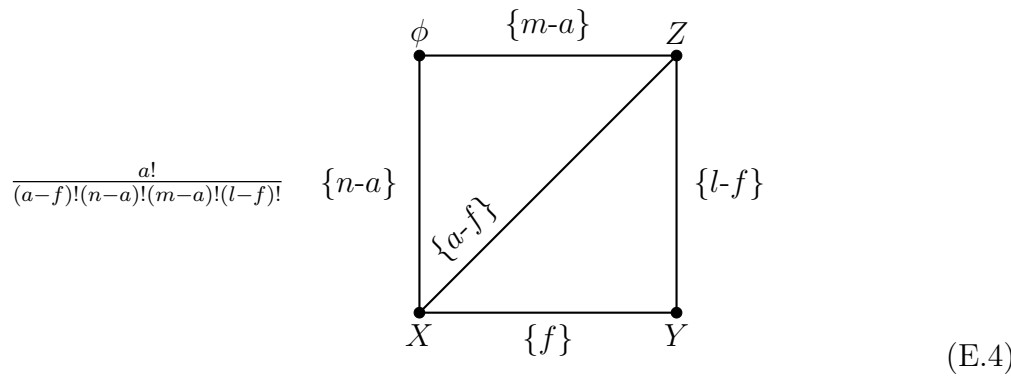

$$(\text{E.4})$$

---

[35]The $\{a\}$ legs between $X_{3,5} - Z_{7,9}$ cannot now contract with the legs between $X_{3,5} - \phi_{1,2}$ or with $Z_{7,9} - \phi_{1,2}$ as the former were formed from the latter, thus are prohibited by the rules of contraction of (3.41).

Similarly contracting $b$ legs between $Y_{4,6} - Z_{7,9}$ and $Z_{7,9} - \phi_{1,2}$, followed by subsequently contracting $e$ legs between $Y_{4,6} - \phi_{1,2}$ and $\phi_{1,2} - X_{3,5}$ we get

$$\frac{b!a!}{(b-e)!(a-f)!(n-a-e)!(m-a-b)!(l-f-b)!}$$ 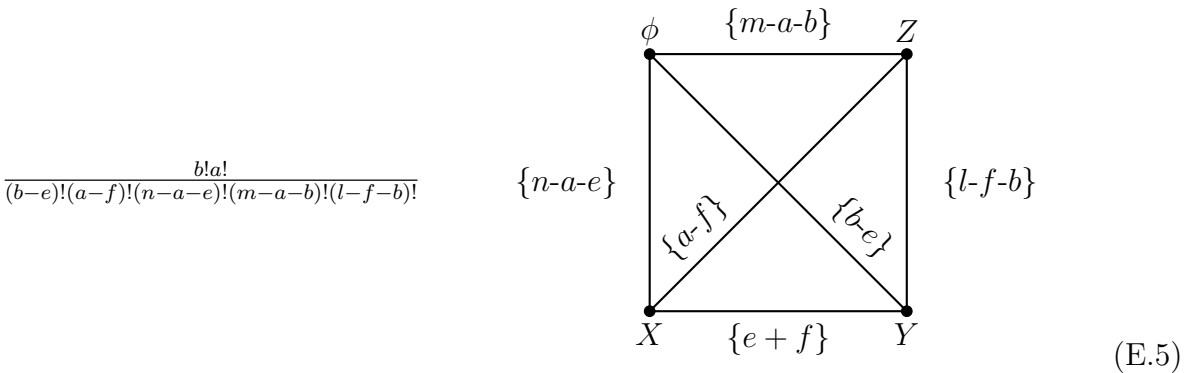

(E.5)

Using the above diagram we can construct any diagram we want starting from (E.2) and figure out its normalization. The power of $c$ in the denominator is simply equal to the total number of legs in the final diagram. Also note that given a particular value of $\{n, m, l\}$ in (E.2), it does not matter how many different ways one can construct a desired diagram by contractions of stress-tensor. For example, one of the diagrams that would contribute to the connected 8pt vacuum block at $\mathcal{O}(c^{-3})$ is

$$\phi \qquad Z$$
$$X \qquad Y$$
$$conn.$$

(E.6)

This can be reached in 2 possible ways

$$\phi \quad Z \rightarrow \phi \quad Z \rightarrow \phi \quad Z$$
$$X \quad Y \qquad X \quad Y \qquad X \quad Y$$

(E.7)

or

$$\phi \quad Z \rightarrow \phi \quad Z \rightarrow \phi \quad Z$$
$$X \quad Y \qquad X \quad Y \qquad X \quad Y$$

(E.8)

We can choose any one of them and this degeneracy doesn't contribute a numerical factor of 2. The reason why this is so can simply be traced back to the algebraic expression

$\langle X_3 X_5 | T^{(2)} | \phi_1 \phi_2 | T^{(1)} | Z_7 Z_9 | T^{(2)} | Y_4 Y_6 \rangle$ and its contributions at $\mathcal{O}(c^{-3})$ order. We therefore see that making use of the normalization for different contractions leading up to (E.5) that only the diagrams in the *rhs* of (3.58) contribute at $\mathcal{O}(c^{-3})$ with the combinatorial normalization of (E.5) as 1.

# F    Appendix F: Leading vacuum block for simple 8pt and 9pt functions.

Here we explicitly state the results for the connected vacuum blocks of simpler higher point generalizations of pair-wise equal 6pt function of light operators *i.e.* $\langle X_3 X_5 X_7 \, \phi_1 \phi_2 Y_4 Y_6 Y_8 \rangle$ and $\langle X_3 X_5 X_7 \, \phi_0 \phi_1 \phi_2 Y_4 Y_6 Y_8 \rangle$. These also occur at $\mathcal{O}(1/c^2)$. The only expressions we need are the 4pt vacuum block $\sim \langle \mathcal{B}^{(1)}_{ij} \mathcal{B}^{(1)}_{kl} \rangle$ and $\mathcal{K}^{(2)}$ computed from (3.28) using $\langle \epsilon \epsilon \phi \phi \rangle$. The cross ratios we use are

$$z = \frac{z_{12} z_{34}}{z_{13} z_{24}}, \ \xi = \frac{z_{15} z_{34}}{z_{13} z_{54}}, \ \eta = \frac{z_{16} z_{34}}{z_{13} z_{64}}, \ \sigma = \frac{z_{17} z_{34}}{z_{13} z_{74}}, \ \chi = \frac{z_{18} z_{34}}{z_{13} z_{84}}, \ \zeta = \frac{z_{10} z_{34}}{z_{13} z_{04}} \quad \text{(F.1)}$$

In the main text we gave expression for 8 point vacuum block namely,

$$\frac{\langle X_3 X_5 X_7 \phi_1 \phi_2 Y_4 Y_6 Y_8 \rangle_{\text{vac}}}{\langle X_3 X_5 X_7 \rangle \langle \phi_1 \phi_2 \rangle \langle Y_4 Y_6 Y_8 \rangle} = \left( 1 + \frac{\langle \phi_1 \phi_2 T^{(1)} | X_3 X_5 X_7 | T^{(1)} \rangle \langle Y_4 Y_6 Y_8 \rangle_{\text{conn.}}}{\langle X_3 X_5 X_7 \rangle \langle \phi_1 \phi_2 \rangle \langle Y_4 Y_6 Y_8 \rangle} + \right.$$
$$+ \frac{\langle \phi_1 \phi_2 | T^{(1)} | Y_4 Y_6 Y_8 | T^{(1)} | X_3 X_5 X_7 \rangle_{\text{conn.}}}{\langle X_3 X_5 X_7 \rangle \langle \phi_1 \phi_2 \rangle \langle Y_4 Y_6 Y_8 \rangle} +$$
$$\left. + \frac{\langle X_3 X_5 X_7 | T^{(1)} | \phi_1 \phi_2 | T^{(1)} | Y_4 Y_6 Y_8 \rangle_{\text{conn.}}}{\langle X_3 X_5 X_7 \rangle \langle \phi_1 \phi_2 \rangle \langle Y_4 Y_6 Y_8 \rangle} + \text{disconn.} \right) \quad \text{(F.2)}$$

Each terms in the rhs of (F.2) can be expressed in terms of bilocals,

$$\frac{\langle X_3 X_5 X_7 | T | \phi_1 \phi_2 | T | Y_4 Y_6 Y_8 \rangle_{\text{conn.}}}{\langle \phi_1 \phi_2 \rangle \langle X_3 X_5 X_7 \rangle \langle Y_4 Y_6 Y_8 \rangle} = \frac{1}{4} h_X h_Y \frac{\langle \mathcal{B}^{(1)}_{357} \mathcal{B}^{(1)}_{468} \phi_1 \phi_2 \rangle'}{\langle \phi_1 \phi_2 \rangle}$$

$$\frac{\langle \phi_1 \phi_2 | T^{(1)} | X_3 X_5 X_7 | T^{(1)} | Y_4 Y_6 Y_8 \rangle_{\text{conn.}}}{\langle X_3 X_5 X_7 \rangle \langle \phi_1 \phi_2 \rangle \langle Y_4 Y_6 Y_8 \rangle} = \frac{h_\phi h_Y}{4} \left[ \frac{\langle \mathcal{B}^{(1)}_{12} \mathcal{B}^{(1)}_{468} X_3 X_5 \rangle'}{\langle X_3 X_5} + \right.$$
$$\left. + \frac{h_X^2}{2} \langle \mathcal{B}^{(1)}_{12} \mathcal{B}^{(1)}_{35} \rangle \langle \mathcal{B}^{(1)}_{468} (\mathcal{B}^{(1)}_{37} + \mathcal{B}^{(1)}_{57} - \mathcal{B}^{(1)}_{35}) \rangle + cyclic_{(3,5,7)} \right]$$
$$\text{(F.3)}$$

where $\mathcal{B}^{(1)}_{ijk}$ is defined as a cyclic sum of bilocals,

$$\mathcal{B}^{(1)}_{ijk} = \mathcal{B}^{(1)}_{ij} + \mathcal{B}^{(1)}_{jk} + \mathcal{B}^{(1)}_{ki} \quad \text{(F.4)}$$

It is enough to compute connected piece of $\frac{\langle \mathcal{B}^{(1)}_{35} \mathcal{B}^{(1)}_{46} \phi_1 \phi_2 \rangle}{\langle \phi_1 \phi_2 \rangle}$ for the computation of 8 and 9 point functions. To see this we determine first line of equation (F.3),

$$\frac{1}{4} h_X h_Y \frac{\langle \mathcal{B}^{(1)}_{357} \mathcal{B}^{(1)}_{468} \phi_1 \phi_2 \rangle'}{\langle \phi_1 \phi_2 \rangle} = \frac{h_X h_Y}{4} \{ \mathcal{A}(z, \xi, \eta, h_\phi) + \mathcal{A}(z, \xi, \chi, h_\phi) + \mathcal{A}(z, \sigma, \eta, h_\phi) + \mathcal{A}(z, \sigma, \chi, h_\phi) +$$

$$+\mathcal{A}\left(\frac{z}{\xi}, \frac{\sigma}{\xi}, \frac{\chi}{\xi}, h_\phi\right) + \mathcal{A}\left(\frac{z}{\xi}, \frac{\sigma}{\xi}, \frac{\eta}{\xi}, h_\phi\right) + \mathcal{A}\left(-\frac{(\eta-1)z}{z-\eta}, -\frac{(\eta-1)\xi}{\xi-\eta}, \frac{(\eta-1)\chi}{\eta-\chi}, h_\phi\right)$$
$$+\mathcal{A}\left(-\frac{(\eta-1)z}{z-\eta}, -\frac{(\eta-1)\sigma}{\sigma-\eta}, \frac{(\eta-1)\chi}{\eta-\chi}, h_\phi\right) + \mathcal{A}\left(\frac{z(\xi-\eta)}{\xi(z-\eta)}, \frac{\sigma(\xi-\eta)}{\xi(\sigma-\eta)}, \frac{\chi(\xi-\eta)}{\xi(\chi-\eta)}, h_\phi\right)\Big\}\text{(F.5)}$$

where $\mathcal{A}(z, \xi, \eta)$ is given by,

$$\mathcal{A}(z, \xi, \eta, h_\phi) = \frac{\langle \mathcal{B}_{35}^{(1)} \mathcal{B}_{46}^{(1)} \phi_1\phi_2\rangle'}{\langle \phi_1\phi_2\rangle} = h_\phi(h_\phi - 1)\langle \mathcal{B}_{35}^{(1)} \mathcal{B}_{12}^{(1)}\rangle\langle \mathcal{B}_{12}^{(1)} \mathcal{B}_{46}^{(1)}\rangle + h_\phi\left(\frac{12}{c}\right)^2 \mathcal{K}^{(2)} \qquad \text{(F.6)}$$

By knowing $\mathcal{A}(z, \xi, \eta, h)$ we can also easily compute other terms in the 8 point vacuum block. We find,

$$\frac{\langle \phi_1\phi_2|T^{(1)}|X_3 X_5 X_7|T^{(1)} Y_4 Y_6 Y_8\rangle_{\text{conn.}}}{\langle X_3 X_5 X_7\rangle\langle \phi_1\phi_2\rangle\langle Y_4 Y_6 Y_8\rangle} = \frac{h_\phi h_Y}{4}\Big\{ \mathcal{I}(z, \xi, \eta, \sigma, \chi, h_X) + \mathcal{I}\left(\frac{z}{\xi}, \frac{\sigma}{\xi}, \frac{\eta}{\xi}, \frac{1}{\xi}, \frac{\chi}{\xi}, h_X\right)$$
$$+\mathcal{I}\left(\frac{z}{\sigma}, \frac{1}{\sigma}, \frac{\eta}{\sigma}, \frac{\xi}{\sigma}, \frac{\chi}{\sigma}, h_X\right)\Big\} \qquad \text{(F.7)}$$

$$\frac{\langle \phi_1\phi_2|T^{(1)}|Y_4 Y_6 Y_8|T^{(1)} X_3 X_5 X_7\rangle_{\text{conn.}}}{\langle X_3 X_5 X_7\rangle\langle \phi_1\phi_2\rangle\langle Y_4 Y_6 Y_8\rangle} = \frac{h_\phi h_X}{4}\Big\{ \mathcal{I}\left(\frac{z}{-1+z}, \frac{\eta}{-1+\eta}, \frac{\xi}{-1+\xi}, \frac{\chi}{-1+\chi}, \frac{\sigma}{-1+\sigma}, h_Y\right) +$$
$$+\mathcal{I}\left(\frac{(\eta-1)z}{\eta(z-1)}, \frac{(\eta-1)\chi}{\eta(\chi-1)}, \frac{(\eta-1)\xi}{\eta(\xi-1)}, \frac{\eta-1}{\eta}, \frac{(\eta-1)\sigma}{\eta(\sigma-1)}, h_Y\right) +$$
$$+\mathcal{I}\left(\frac{(\chi-1)z}{\chi(z-1)}, \frac{\chi-1}{\chi}, \frac{\xi(\chi-1)}{(\xi-1)\chi}, \frac{\chi-1}{\chi}, \frac{\sigma(\chi-1)}{(\sigma-1)\chi}, h_Y\right)\Big\} \qquad \text{(F.8)}$$

where $\mathcal{I}(z, \xi, \eta, \sigma, \chi, h_X)$ is given by,

$$\mathcal{I}(z, \xi, \eta, \sigma, \chi, h_X) = \mathcal{D}(z, \xi, \eta, h_X) + \mathcal{D}(z, \xi, \chi, h_X) + \mathcal{D}\left(-\frac{(\eta-1)z}{z-\eta}, -\frac{(\eta-1)\xi}{\xi-\eta}, \frac{(\eta-1)\chi}{\eta-\chi}, h_X\right)$$
$$+\frac{12}{c}\left(\frac{(\xi(z-2)+z)\log\left(\frac{\xi(z-1)}{z-\xi}\right)}{(\xi-1)z} - 2\right)\Big[\tilde{\mathcal{D}}(\xi, \eta, \sigma, h_X) + \tilde{\mathcal{D}}(\xi, \chi, \sigma, h_X) +$$
$$+\tilde{\mathcal{D}}\left(-\frac{(\eta-1)\xi}{\xi-\eta}, \frac{(\eta-1)\chi}{\eta-\chi}, \frac{(\eta-1)\sigma}{\eta-\sigma}, h_X\right)\Big] \qquad \text{(F.9)}$$

$$\mathcal{D}(z, \xi, \eta, h_X) = \mathcal{A}(1-\xi, 1-z, 1-\eta, h_X)$$
$$\tilde{\mathcal{D}}(\xi, \eta, \sigma, h_X) = \frac{6h_X^2}{c}\left(\frac{(-2\eta+\xi+\sigma)\log\left(\frac{\eta-\xi}{\eta-\sigma}\right)}{\xi-\sigma} + \frac{(-2\eta+\xi+1)\log\left(\frac{\eta-1}{\eta-\xi}\right)}{\xi-1} - \frac{(-2\eta+\sigma+1)\log\left(\frac{\eta-1}{\eta-\sigma}\right)}{\sigma-1} - 2\right)$$
$$\text{(F.10)}$$

Using similar method we can also compute 9 point vacuum block,

$$\frac{\langle X_3 X_5 X_7 \phi_0\phi_1\phi_2 Y_4 Y_6 Y_8\rangle_{\text{vac}}}{\langle X_3 X_5 X_7\rangle\langle \phi_0\phi_1\phi_2\rangle\langle Y_4 Y_6 Y_8\rangle} = \Big\{ 1 + \frac{\langle X_3 X_5 X_7|T^{(1)}|\phi_0\phi_1\phi_2|T^{(1)}|Y_4 Y_6 Y_8\rangle_{\text{conn.}}}{\langle X_3 X_5 X_7\rangle\langle \phi_0\phi_1\phi_2\rangle\langle Y_4 Y_6 Y_8\rangle} + \text{symm}_{\{X,\phi,Y\}}\Big\}$$
$$\text{(F.11)}$$

We find each term in the *rhs* as following,

$$\frac{\langle X_3 X_5 X_7|T^{(1)}|\phi_0\phi_1\phi_2|T^{(1)}|Y_4 Y_6 Y_8\rangle_{\text{conn.}}}{\langle X_3 X_5 X_7\langle \phi_0\phi_1\phi_2\langle Y_4 Y_6 Y_8} = \frac{h_X h_Y}{8}\Big[\mathcal{R}\left(\frac{\sigma-\xi}{\zeta-\xi}, \frac{\xi}{\xi-\zeta}, \frac{\eta-\xi}{\zeta-\xi}, \frac{z-\xi}{\zeta-\xi}, \frac{\chi-\xi}{\zeta-\xi}, \frac{\xi-1}{\xi-\zeta}, h_\phi\right)$$

$$+\mathcal{R}\left(1-\tfrac{\sigma}{\xi},1-\tfrac{z}{\xi},1-\tfrac{\eta}{\xi},1-\tfrac{\zeta}{\xi},1-\tfrac{\chi}{\xi},\tfrac{\xi-1}{\xi},h_\phi\right)+$$
$$+\mathcal{R}\left(\tfrac{\sigma-\xi}{z-\xi},\tfrac{\zeta-\xi}{z-\xi},\tfrac{\eta-\xi}{z-\xi},\tfrac{\xi}{\xi-z},\tfrac{\chi-\xi}{z-\xi},\tfrac{\xi-1}{\xi-z},h_\phi\right)\Big] \tag{F.12}$$

while the rest are found by implementing the cordinate exchanges in terms of the cross ratios as

$$X_{\{3,5,7\}}\leftrightarrow\phi_{\{0,1,2\}}\implies\{z,\xi,\eta,\sigma,\chi,\zeta\}\to\left\{\tfrac{\sigma-\xi}{\zeta-\xi},\tfrac{\xi}{\xi-\zeta},\tfrac{\eta-\xi}{\zeta-\xi},\tfrac{z-\xi}{\zeta-\xi},\tfrac{\chi-\xi}{\zeta-\xi},\tfrac{\xi-1}{\xi-\zeta}\right\}$$
$$Y_{\{4,6,8\}}\leftrightarrow\phi_{\{0,1,2\}}\implies\{z,\xi,\eta,\sigma,\chi,\zeta\}\to\left\{\tfrac{(\zeta-1)(\eta-\chi)}{(\eta-1)(\zeta-\chi)},\tfrac{(\zeta-1)(\eta-\xi)}{(\eta-1)(\zeta-\xi)},\tfrac{(\zeta-1)\eta}{\zeta(\eta-1)},\tfrac{(\zeta-1)(\eta-\sigma)}{(\eta-1)(\zeta-\sigma)},\tfrac{(\zeta-1)(z-\eta)}{(\eta-1)(z-\zeta)},\tfrac{\zeta-1}{\eta-1}\right\}$$
$$\tag{F.13}$$

where

$$\mathcal{R}(z,\xi,\eta,\sigma,\chi,\zeta,h_X)=\mathcal{I}(z,\xi,\eta,\sigma,\chi,h_X)+\mathcal{I}\left(\tfrac{z-\zeta}{z-1},\tfrac{z-\xi}{z-1},\tfrac{z-\eta}{z-1},\tfrac{z-\sigma}{z-1},\tfrac{z-\chi}{z-1},\tfrac{z}{z-1},h_X\right)+$$
$$+\mathcal{I}\left(\tfrac{\zeta}{\zeta-1},\tfrac{\zeta-\xi}{\zeta-1},\tfrac{\zeta-\eta}{\zeta-1},\tfrac{\zeta-\sigma}{\zeta-1},\tfrac{\zeta-\chi}{\zeta-1},\tfrac{\zeta-z}{\zeta-1},h_X\right)\tag{F.14}$$

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
