# Peer review of "Reparametrization mode Ward Identities and chaos in higher-pt. correlators in CFT$_2$"

_SciPost Physics_

## Round 1 · Referee Report · Anonymous (Referee 1) · 2022-3-18

Strengths

1- Well motivated study and generalization of recent technology for conformal blocks in 2d; 2- Novel proposal for Virasoro identity block projection operator in large central charge expansion, which could facilitate future investigations into large $c$ Virasoro blocks; 3- Demonstrates utility of approach by tackling technical, non-trivial problems of interest and making contact with known results obtained via different methods; 4- Referencing is good;

Weaknesses

1- Content is interesting and has potential to be useful, if a bit technical and possibly limited in scope; 2- Some key proposals, evidence, and results rendered ambiguous due to grammar mistakes, unclear explanations, and typos; 3- While commendable effort has been expended to be self-contained, some definitions and notation used without introduction;

Report

This paper further develops the use of the shadow operator formalism, and in particular it's implementation via "reparameterization" modes, in two dimensional CFT. The authors use this formalism to then study 6pt functions of light operators at large central charge, while also touching on generalizations to higher point functions as well as the behavior of correlation functions with two heavy operators (i.e. with dimensions $\cal O (c)$). The authors then briefly use their results to probe chaos, via out-of-time order correlation functions, and reproduce expected behavior. Finally, in a holographic context, the authors remark on a curious relation between the equations governing the bulk geometry and the conformal Casimir equation satisfied by the stress tensor.

While the motivation, methods utilized, and results claimed of this work are interesting and could prove useful for subsequent work in the field, unfortunately the presentation makes it difficult to assess the validity and applicability of the results. In particular, while the authors claim to reproduce existing results, the language in the text suggests that additional work is required to verify that their 6pt result matches that in the literature (c.f. the discussion shortly after eq. (3.50)). As this 6pt function provides non-trivial evidence in support of the authors' proposal for the Virasoro identity projector, which is used heavily in the subsequent formalism, this ambiguity in the validity of the result merits further discussion.

Furthermore, the work is in need of substantial editing and proofreading, as the grammar and organization hinder readability. The work additionally suffers from some typos and notational inconsistency. While the authors' demonstrate adequate awareness of existing literature, at times the manuscript fails to be self-contained and uses notation/terminology without prior introduction.

In all, I believe that this work would be worth publishing if the above points are clarified, and the grammar and presentation are substantially improved. Most importantly, how conclusive is the matching of the 6pt example with existing results? The work would also benefit from a discussion of other avenues to verify their proposal; for example, can their proposal address $1/c$ corrections to the Virasoro 4pt vacuum block, beyond leading order in external operator dimensions $h$ (in comparison with e.g. hep-th/1512.03052)?

Requested changes

1- Clarify relation of 6pt result with previous work, in particular the 6pt conformal block from ref [36]; does your result, eq. 3.50, agree with that of [36] and if not, why? Similarly, the authors mention that "part of the comb channel" is accessible via their results; what does "part" mean, and are the results compatible with the global comb blocks discussed in [51]?; 2- Clarify proposal for Virasoro identity projector and explicitly outline steps for using it to compute blocks at a given order in $1/c$; e.g., the proposal is claimed to be a $1/c$ expansion but the authors then demonstrate how, in the case of a 6pt fn, components at naive $\cal{O}(c^{-3})$ are enhanced to $\cal{O} (c^{-2})$; is this type of behavior expected generically and do the authors have any speculations on what (if anything) changes using the proposal for correlators beyond pairwise (or triple-wise) identical operators? 3- Much of the work focuses on the vacuum block contributions, due to its presumed dominance; the authors should explicitly comment on this assumption of vacuum block dominance in the channel under consideration (especially when analytically continuing for OTOCs); 4- Typos, inconsistent/unclear notation: (i) inconsistent relation between $\Delta$ and $h$, $\bar h$ (i.e. in sections 2 and 3) and unclear notation with $k(d, \Delta)$ and $k'(d, \Delta, l)$ in section 2; additionally, including the conformal weights ${h, \bar h}$ for $\tilde T$ would be useful to the reader; (ii) explicitly introduce holomorphic/antiholomorphic representation of reparameterization modes $\epsilon^\mu$ in $\S$2 (i.e. $\epsilon^z =\epsilon$ and $\epsilon^{\bar z} = \bar \epsilon$); (iii) notation for vacuum state projector $I$, in e.g. eq (3.35), (3.42) etc, a bit misleading (it looks like the identity operator) and worth a comment; additional projector notation is inconsistent ($|T|$ vs. $|T^{(1)}|$ vs. $|T^1|$, etc); (iv) what do the dots signify in eq (3.50)? Also, the notation for the left hand side (i.e. $V^{(6)}_T$) should be defined somewhere (is it meant to be the same as ${\cal V}^{(6)}_T$ in eq (3.29)?); (v) sum over j in (3.75) should be over 1,2 not 0,1,2; (vi) the coordinate transformation $z \to w$ for HHLL discussion (after equation (3.89)) is not quite right (it doesn't send $z_i$ to $w_i$ as claimed); (vii) how is the Regge limit defined (in terms of the Lorentzian time separations) for the 6pt function studied in $\S$4?; (viii) include relation between $r_+$ and $T$ in $\S$5; (ix) at times, authors' distinction between correlator, global block, and Virasoro block unclear;

  • validity: low
  • significance: good
  • originality: good
  • clarity: low
  • formatting: acceptable
  • grammar: below threshold

Author:  Rohan R. Poojary  on 2022-05-12  [id 2460]

(in reply to Report 1 on 2022-03-18)

We thank the referee for their valuable suggestions and find the critique of the manuscript well warranted. We have performed the necessary checks and report its
findings here in advance. We also thank the referee for pointing out the lack of clarity in the manuscript and would be making appropriate changes along the lines
suggested.

The proposition for the vacuum block projector (eq(3.41)) is wrong!. This we find after comparing the terms proportional to $h_\phi^2$ in the $1/c^2$ contribution of the vacuum block for the
4pt. light operators $\langle \phi_1\phi_2\psi_3\psi_4\rangle$. Here the form of the proposed vacuum block projector at $1/c^2$ order allows us to explicitly compute terms
quadratic in any of the light operator dimensions. However this very contribution can be computed from the leading (in $1/c$) term in HHLL correlator (where $\phi$ is taken
to be heavy). If one uses the proposed (wrong) projector to compute the leading order contribution to HHLL correlator as is done in sec.(3.3) then one seems to get the right
expected answer at $1/c^2$. Nonetheless this projector is wrong and we would omit the relevant parts from our paper.

We further find that we indeed compute the 6pt. Virasoro Comb Channel vacuum block and not the Star Channel vacuum block as claimed in the paper.
This is given by the first term in eq(3.50) (the other two terms in this equation are not present for the Comb channel).
The $\langle \epsilon \epsilon \phi \phi\rangle $ correlator (eq(3.14)) indeed does allow us to compute the Virasoro Comb Channel as depicted in Figure 1(b) of the paper.
We also find that this has not been computed in the literature before and that the Ward identity of the reparametrization modes considered in the paper allows us
to ascertain this. This method of computing also readily allows us to promote any of the pairs to triplets. We also find that such simpler generalizations to
higher (7,8,and 9)pt. functions does not affect the scrambling time $i.e.$ the scrambling time is still $2t_*$ with $t_*$ being the scrambling time for 4pt OTOCs.

We also find a crucial physical difference highlighted by the Virasoro comb channel vacuum block as compared to just the Global answer: The global comb channel does not show an
exponential growth if the operator pairs $X_3,X_5$ and $Y_4,Y_6$ are Out of Time Ordered w.r.t. each other while being Time Ordered w.r.t. the pair $\phi_1,\phi_2$ ( Fig.1(b)).
However the same comb channel grows exponentially when considering the full Virasoro contribution. Physically this seems to bode well with the picture of Virasoro
modes propagating in the internal leg of the comb channel (Fig.1(b)) corresponding to gravitons being exchanged in the analogous bulk setting. We also note that
the maximally braided OTOC considered for the global comb channel in ref[36] required the $X_{3,5}$ pair to be Out of Time Ordered with the $\phi_1$ while
$Y_{4,6}$ to be Out of Time Ordered with $\phi_2$ $i.e.$ each of the 'Out of Time Ordering' involves Virasoro states being exchanged while only global states are
exchanged between the $X_{3,5}$ pair and $Y_{4,6}$ pair along the internal line.

We would hence be resubmitting the manuscript after incorporating the above mentioned edits accordingly and also address all the other non-technical issues raised in the reports.

---

## Round 1 · Referee Report · Anonymous (Referee 2) · 2022-3-23

Strengths

1- Provides a formula that, if true, would be extremely useful for algorithmically constructing the Virasoro identity block for arbitrary numbers of operators.

Weaknesses

1- Does not provide sufficient evidence for the important formula (3.41)

2- The presentation is obscure, even for experts

Report

In this work, the authors engage in the theory of reparametrization modes in 2d CFT--denoted $\epsilon(z,\bar{z})$--which encode (multi-)stress-tensor exchanges in the Virasoro identity block of arbitrary numbers of external operators. The paper attempts to understand how Virasoro ward identities govern the form of multiple $\epsilon$-insertions, which is an important task.

Perhaps the most interesting equation in the whole paper is (3.41) which claims to represent, in the shadow operator formalism, the projector onto the Virasoro identity block. If this formula were shown to be correct, this would prove to be an incredible advance, as it provides an algorithmic procedure for computing the Virasoro vacuum block order by order in a $1/c$ expansion, for arbitrary numbers of external operators!

However, I do not think that the authors have given enough evidence to show that (3.41) is actually correct. In an odd choice of presentation, the authors decide to first demonstrate (3.41)'s validity for the 6pt funtion, rather than,e.g. the 4pt function, leading to equation (3.50) for the $O(c^{-2})$ contribution to the 6pt Virasoro vacuum block. Evaluating this expression requires the reader to unpackage (3.27) from (2.22) and (3.22) and to symmetrize this, after which it is still not clear if the answer matches the result from [36].

It is perhaps even more important to establish the validity of the projector for the HHLL 4pt function. In this case the expression for the $1/c$ corrections to the leading term are known and given in equation (1.4) of 1512.03052. This has also been reproduced using the reparametrization mode theory in equation (6.50) of 1808.03263. For this paper to be accepted, I would like to see evidence that (3.41) is capable of reproducing these corrections both in the HHLL four point function case and in the 6pt case matching with [36].

I would also like to add that this paper is mired with other odd presentation choices. For example, the entire study is crucially done in 2d, however, the review in section 2 applies for arbitrary d, requiring the addition Lorentz indices and projectors that have no use in the analysis of the paper, at the expense of readability. Section 3 is devoted to the use of Virasoro Ward identities (only applicable in 2d), but has expressions that seemingly look valid in arbitrary d (e.g. (3.4)). Section 3.1 is called "Channel Diagrammatics" but is almost entirely devoid of diagrams, and includes extremely cumbersome equations such as (3.43) or (3.48), which are presented but never massaged into usable expressions, leaving the work for the reader. The proof that (3.41) is a projector only comes in section 3.3 during the discussion about external operator dimensions scaling with $c$.

The paper also claims to derive nice expressions for 8- and 9-point functions, but these are also given in a compressed format, requiring hefty amounts of decoding, such that it is not clear that they can be used. For example, it is claimed by Haehl and Rozali that the OTO configurations of 2k-point function in a chaotic CFT has the same Lyapunov growth for arbitrary k, and only the scrambling time grows with k. Is it possible to see this from the analysis in section 3 for e.g. the 8-pt function?

Thus, I find myself in the position of requesting major changes. The paper first and foremost needs to establish that equation (3.41) gives results consistent with the literature (both [36] and 1512.03052) . And I suggest the authors do so in a way that readers would find manageable to reproduce. I also recommend a significant streamlining of the presentation. Often formulas are presented without further use, obfuscating what could be a simple point. I would avoid this wherever possible.

I think it would also be important to explain why the use of the ward identities in the theory of reparametrization modes is necessarily different from the coadjoint orbit story.

Requested changes

1- Give evidence for the validity of (3.41) by matching with known results from the literature

2-Package formulas in a way that would make them usable to readers, e.g. equation 3.50.

3-Streamline the presentation significantly.

  • validity: -
  • significance: high
  • originality: ok
  • clarity: poor
  • formatting: below threshold
  • grammar: good

Author:  Rohan R. Poojary  on 2022-05-12  [id 2459]

(in reply to Report 2 on 2022-03-23)
Category:
correction
validation or rederivation

We thank the referee for their valuable suggestions and find the critique of the manuscript well warranted. We have performed the necessary checks and report its
findings here in advance. We also thank the referee for pointing out the lack of clarity in the manuscript and would be making appropriate changes along the lines
suggested.

The proposition for the vacuum block projector (eq(3.41)) is wrong!. This we find after comparing the terms proportional to $h_\phi^2$ in the $1/c^2$ contribution of the vacuum block for the
4pt. light operators $\langle \phi_1\phi_2\psi_3\psi_4\rangle$. Here the form of the proposed vacuum block projector at $1/c^2$ order allows us to explicitly compute terms
quadratic in any of the light operator dimensions. However this very contribution can be computed from the leading (in $1/c$) term in HHLL correlator (where $\phi$ is taken
to be heavy). If one uses the proposed (wrong) projector to compute the leading order contribution to HHLL correlator as is done in sec.(3.3) then one seems to get the right
expected answer at $1/c^2$. Nonetheless this projector is wrong and we would omit the relevant parts from our paper.

We further find that we indeed compute the 6pt. Virasoro Comb Channel vacuum block and not the Star Channel vacuum block as claimed in the paper.
This is given by the first term in eq(3.50) (the other two terms in this equation are not present for the Comb channel).
The $\langle \epsilon \epsilon \phi \phi\rangle $ correlator (eq(3.14)) indeed does allow us to compute the Virasoro Comb Channel as depicted in Figure 1(b) of the paper.
We also find that this has not been computed in the literature before and that the Ward identity of the reparametrization modes considered in the paper allows us
to ascertain this. This method of computing also readily allows us to promote any of the pairs to triplets. We also find that such simpler generalizations to
higher (7,8,and 9)pt. functions does not affect the scrambling time $i.e.$ the scrambling time is still $2t_*$ with $t_*$ being the scrambling time for 4pt OTOCs.

We also find a crucial physical difference highlighted by the Virasoro comb channel vacuum block as compared to just the Global answer: The global comb channel does not show an
exponential growth if the operator pairs $X_3,X_5$ and $Y_4,Y_6$ are Out of Time Ordered w.r.t. each other while being Time Ordered w.r.t. the pair $\phi_1,\phi_2$ ( Fig.1(b)).
However the same comb channel grows exponentially when considering the full Virasoro contribution. Physically this seems to bode well with the picture of Virasoro
modes propagating in the internal leg of the comb channel (Fig.1(b)) corresponding to gravitons being exchanged in the analogous bulk setting. We also note that
the maximally braided OTOC considered for the global comb channel in ref[36] required the $X_{3,5}$ pair to be Out of Time Ordered with the $\phi_1$ while
$Y_{4,6}$ to be Out of Time Ordered with $\phi_2$ $i.e.$ each of the 'Out of Time Ordering' involves Virasoro states being exchanged while only global states are
exchanged between the $X_{3,5}$ pair and $Y_{4,6}$ pair along the internal line.

We would hence be resubmitting the manuscript after incorporating the above mentioned edits accordingly and also address all the other non-technical issues raised in the reports.

---

## Editorial Decision

resubmitted